# MathVista: Evaluating Mathematical Reasoning of Foundation Models in Visual Contexts

**Pan Lu**[1,3]**, Hritik Bansal**[1]**, Tony Xia**[1]**, Jiacheng Liu**[2]**, Chunyuan Li**[3]**,**
**Hannaneh Hajishirzi**[2]**, Hao Cheng**[3]**, Kai-Wei Chang**[1]**, Michel Galley**[3]**, Jianfeng Gao**[3]
[1]UCLA, [2]University of Washington, [3]Microsoft Research, Redmond
**https://mathvista.github.io**

## Abstract

Large Language Models (LLMs) and Large Multimodal Models (LMMs) exhibit impressive problem-solving skills in many tasks and domains, but their ability in mathematical reasoning in visual contexts has not been systematically studied. To bridge this gap, we present MathVista, a benchmark designed to combine challenges from diverse mathematical and visual tasks. It consists of 6,141 examples, derived from 28 existing multimodal datasets involving mathematics and 3 newly created datasets (*i.e.*, IQTest, FunctionQA, and PaperQA). Completing these tasks requires fine-grained, deep visual understanding and compositional reasoning, which all state-of-the-art foundation models find challenging.

With MathVista, we have conducted a comprehensive, quantitative evaluation of 12 prominent foundation models. The best-performing GPT-4V model achieves an overall accuracy of 49.9%, substantially outperforming Bard, the second-best performer, by 15.1%. Our in-depth analysis reveals that the superiority of GPT-4V is mainly attributed to its enhanced visual perception and mathematical reasoning. However, GPT-4V still falls short of human performance by 10.4%, as it often struggles to understand complex figures and perform rigorous reasoning. This significant gap underscores the critical role that MathVista will play in the development of general-purpose AI agents capable of tackling mathematically intensive and visually rich real-world tasks. We further explore the new ability of *self-verification*, the application of *self-consistency*, and the interactive chatbot capabilities of GPT-4V, highlighting its promising potential for future research.

## 1 Introduction

Mathematical reasoning stands as a testament to the intricacies of human intelligence (Kahneman, 2011). It requires rigorous logical thinking, domain-specific knowledge, and the ability to engage in multistep reasoning processes (Lightman et al., 2023). This complexity is observed not only in textual scenarios but also significantly in visual contexts. When assessing a child's mathematical reasoning capabilities, problems are often designed to encompass visual contexts (Stipek & Iver, 1989; Pollitt et al., 2020). At the same time, AI agents with strong mathematical reasoning capabilities in visual contexts have a wide range of real-world applications, such as solving complex problems (Seo et al., 2015; Wang et al., 2017), addressing logical queries about statistical data (Wu et al., 2023; Yang et al., 2023a; Liu et al., 2024b), and assisting in theorem proving and scientific discovery in advanced research (Taylor et al., 2022; Dong et al., 2023; Trinh et al., 2024).

Numerous datasets have been curated to assess the mathematical reasoning abilities of AI systems, with most presented purely in text form. Some datasets such as ChartQA (Lu et al., 2021a; Dahlgren Lindström & Abraham, 2022; Masry et al., 2022) have explored mathematical reasoning in vision-language settings. However, these datasets tend to either focus on specific tasks, like math word problems, or particular visual contexts, such as geometry problems or bar charts. General-purpose visual question answering (VQA) datasets on natural scenes contain only a small portion of questions necessitating mathematical reasoning, leaving a comprehensive investigation of vision-language reasoning within a mathematical framework largely unexplored.

On the other hand, Large Language Models (LLMs) (OpenAI, 2022; 2023a) and Large Multimodal Models (LMMs) (Google, 2023; OpenAI, 2023b; Team et al., 2023) have exhibited impressive

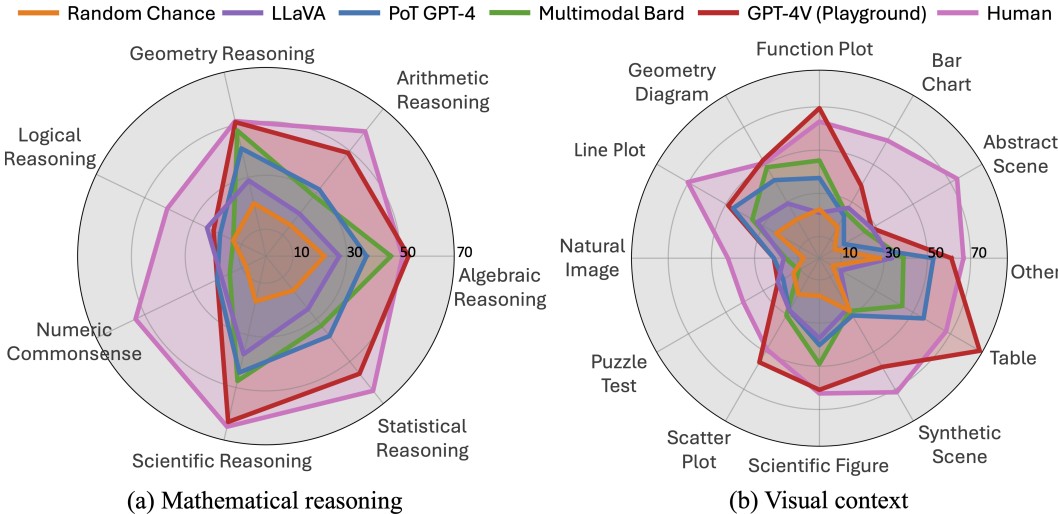

Figure 1: Accuracies of one leading LLM (*i.e.*, PoT GPT-4), four prominent LMMs, random chance, and human performance on our proposed MATHVISTA across mathematical reasoning and visual context types. PoT GPT-4 is a textual, program-aided LLM augmented with the Bard caption and OCR text. GPT-4V is manually evaluated via the playground chatbot.

problem-solving skills in many tasks and domains. Recently, some studies have aimed to augment existing LLMs with mathematical and scientific reasoning capabilities using external tools (Lu et al., 2023a; Wang et al., 2023b). However, the ability of these foundation models to perform mathematical reasoning in visual contexts has not been systematically examined. Therefore, it is essential to develop a new benchmark to (1) facilitate the development of mathematical reasoning systems in visually intensive scenarios, and (2) evaluate the research progress of LLMs and LMMs, especially their capabilities in solving rigorous reasoning tasks.

In this paper, we present MATHVISTA, a consolidated **Math**ematical reasoning benchmark in **Vis**ual contexts. We propose a task taxonomy to guide the development of MATHVISTA: (1) we identify seven mathematical reasoning types: *algebraic reasoning*, *arithmetic reasoning*, *geometry reasoning*, *logical reasoning*, *numeric common sense*, *scientific reasoning*, and *statistical reasoning*; (2) we focus on five primary tasks: *figure question answering* (FQA), *geometry problem solving* (GPS), *math word problem* (MWP), *textbook question answering* (TQA), and *visual question answering* (VQA); and (3) we encompass a diverse array of visual contexts, including natural images, geometry diagrams, abstract scenes, synthetic scenes, as well as various figures, charts, and plots. MATHVISTA incorporates 28 existing multimodal datasets, including 9 math-targeted question answering (MathQA) datasets and 19 VQA datasets. In addition, we have created three new datasets (*i.e.*, IQTest, FunctionQA, PaperQA) which are tailored to evaluating logical reasoning on puzzle test figures, algebraic reasoning over functional plots, and scientific reasoning with academic paper figures, respectively. Overall, MATHVISTA consists of 6,141 examples, with 736 of them being newly curated (Table 1). To facilitate fine-grained evaluation, examples are annotated with metadata, including question type, answer type, task category, grade level, visual context, and required reasoning skills. Detailed descriptions of data collection can be found in §2, §C, and §D.

We conduct extensive experiments on MATHVISTA to evaluate the reasoning abilities of 12 foundation models known for their leading performance in mathematical and multimodal reasoning. This ensemble includes three LLMs (*i.e*, ChatGPT, GPT-4, Claude-2), two proprietary LMMs (*i.e.*, GPT-4V, Bard), and seven open-source LMMs. For LLMs, we examine zero-shot and few-shot settings using two prompting strategies: chain-of-thought (CoT) (Wei et al., 2022b) and program-of-thought (PoT) (Chen et al., 2022b). These LLMs can also be augmented with off-the-shelf visual models for image captioning and OCR. We establish a human performance baseline by engaging qualified human annotators with a high school diploma or higher. We show that MATHVISTA, featuring advanced topics such as college curricula and scientific reasoning, is a very challenging benchmark, with human performance reaching only 60.3% accuracy.

Our results indicate that CoT GPT-4, the best-performing LLM without visual tool augmentations, achieves an overall accuracy of 29.2%. Multimodal Bard, the best-performing LMM, achieves

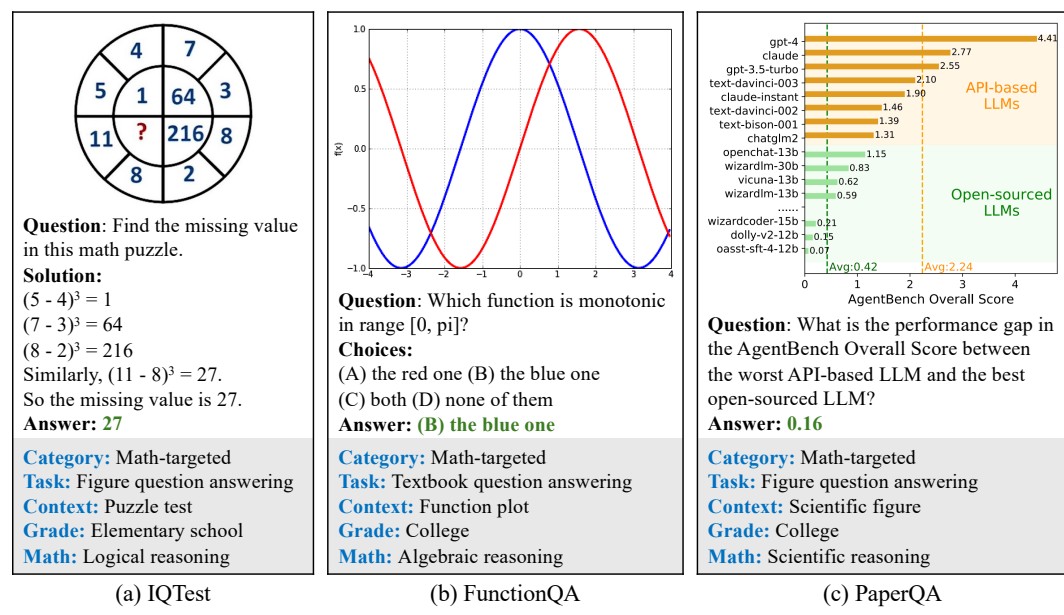

Figure 2: Examples of our newly annotated datasets: IQTest, FunctionQA, and PaperQA.

34.8% (§3.3), which attains only 58% of human performance (34.8% vs 60.3%). When augmented with Bard captions and OCR text, PoT GPT-4 obtains 33.9%, closely matching Multimodal Bard (§3.4). Further analysis indicates that the Multimodal Bard model failures arise from incorrect calculations and hallucinations caused by visual perception and textual reasoning (§3.5).

With MATHVISTA, we report, for the first time, a comprehensive quantitative and qualitative evaluation of GPT-4V (OpenAI, 2023b), the latest multimodal version of GPT-4. Remarkably, GPT-4V achieves a state-of-the-art accuracy of 49.9%, a significant improvement of 15.1% over Multimodal Bard. As illustrated in Figure 1, GPT-4V even surpasses human performance on a set of tasks involving algebraic reasoning and complex visual contexts, which include tables and function plots. Nevertheless, a 10.4% gap in overall accuracy remains when compared to the human baseline, leaving plenty of room for model improvement. Our in-depth analysis (§H) reveals that the superiority of GPT-4V is mainly attributed to its strong capabilities in visual perception and mathematical reasoning. We further highlight its emergent ability for *self-verification* (§H.5), the use of *self-consistency* (§H.6), and its ability to drive goal-directed multi-turn human-AI dialogues (§H.7).

## 2 THE MATHVISTA DATASET

### 2.1 COLLECTION GUIDELINES

Our benchmark, MATHVISTA, is motivated to bridge the notable gap in existing benchmarks, which primarily evaluate mathematical reasoning in textual contexts. It adheres to the following collection guidelines: (1) it covers multiple tasks and topics to mirror real-world applications; (2) it incorporates diverse visual contexts and mathematical skills to foster a well-rounded evaluation; (3) it offers varying levels of challenge to effectively uncover the potential limitations of current models; and (4) it provides robust settings for deterministic evaluations.

The taxonomy for this work is introduced as follows: We identify seven types of mathematical reasoning: *algebraic reasoning*, *arithmetic reasoning*, *geometry reasoning*, *logical reasoning*, *numeric common sense*, *scientific reasoning*, and *statistical reasoning*, with detailed definitions provided in §C.1 and examples shown in §C.2. We focus on five primary tasks: *figure question answering* (FQA), which centers around statistical reasoning over multiple charts and plots; *geometry problem solving* (GPS), which deals with geometrical topics; *math word problem* (MWP), which involves arithmetic reasoning in everyday scenarios; *textbook question answering* (TQA), which usually entails knowledge-intensive reasoning on scientific topics and figures; and *visual question answering* (VQA). Furthermore, our objective is to account for a diverse array of visual contexts, including

natural images, geometry diagrams, abstract scenes, synthetic scenes, multiple charts and plots, scientific figures, tables, function plots, puzzle test figures, and more, with examples shown in §C.3.

## 2.2 DATA COLLECTION

**Collection of MathQA datasets.** We collected nine MathQA datasets in multimodal settings, including four for GPS, two for MWP with visual contexts of synthetic scenes, abstract diagrams, and tables, and two for TQA on college curricula (see §C.4). Annotations such as solutions, programs, parsing results, and grounded theorems are also collected, providing demonstration examples for LLMs. Each source dataset is limited to up to 400 examples to ensure a balanced representation of each source in our final compiled benchmark. In total, we collected 2,666 examples.

**Review and collection of VQA datasets.** Many existing VQA datasets feature instances requiring mathematical reasoning abilities, such as arithmetic operations or numeric common sense. Incorporating these datasets enhances problem diversity in terms of tasks, domains, visual contexts, and reasoning skills involved. We reviewed more than 70 datasets, collecting 19 of them that contain math-related instances and are publicly available, as listed in §C.4. Since these datasets are not originally math-targeted, we initially designed heuristic rules to automatically select examples likely to involve mathematical reasoning from a large pool of candidates. Examples with numeric answers or those containing quantity words (as listed in §D.1) in the questions were selected. This automatic filtration yielded 4,949 VQA-format examples, though some false positive examples remained. Therefore, we engaged three expert annotators to manually label these examples to determine if they involve mathematical reasoning (more details in § D.2). Utilizing majority voting and limiting each source dataset to 400 examples, we finalized a collection of 2,739 examples.

**Collection of three new datasets.** While the source datasets we collected encompass multiple visual contexts and mathematical reasoning abilities, certain scenarios remain unaddressed: logical reasoning on puzzle test diagrams, statistical reasoning on functional plots, and scientific reasoning on academic figures. To address these gaps, we introduced three new datasets: IQTest, FunctionQA, and PaperQA, with examples illustrated in Figure 2. IQTest comprises 228 examples requiring inductive reasoning, abstract thinking, pattern prediction, and calculations, sourced from puzzle test figures on online learning platforms. FunctionQA, with 400 examples, emphasizes subtle visual perceptions of functional plots and algebraic reasoning concerning variables, expressions, equations, and functions. PaperQA is a novel dataset featuring questions derived from informative academic illustrations, including tables, figures, and charts from online education resources, with 107 examples sourced from papers released in August 2023 on Huggingface[1]. To ensure data quality, all questions were manually annotated by graduate students in STEM fields and further refined through a rigorous review process.

## 2.3 METADATA ANNOTATION

Fine-grained metadata facilitates a comprehensive analysis of models' reasoning capabilities across various aspects. To this end, we annotate the examples in MATHVISTA with information including question type, answer type, language, source, category, task, grade level, and visual context, which can be accurately obtained from the details provided in the source datasets. MATHVISTA features seven different types of mathematical reasoning abilities, as categorized in Table 3 (§C.1). Coarse labels of mathematical reasoning can be automatically obtained from the details of the source datasets. To verify the quality of automatic annotation, expert annotators manually label the mathematical reasoning categories from seven candidates for 1,000 examples, using the annotation tool illustrated in §D.4. The results show that 94.1% of the examples from automatic and human annotations have the exact same set of reasoning types, while 98.79% of the individual labels are identical, indicating that the automatic annotation for the labeling of mathematical reasoning is highly accurate.

## 2.4 DATA PREPARATION AND RELEASE

MATHVISTA consists of 6,141 examples, divided into two subsets: *testmini* and *test*. *testmini* contains 1,000 examples, intended for model development validation or for those with limited comput-

---

| Statistic | Number |
|---|---|
| Total questions | 6,141 |
| - multiple-choice questions | 3,392 (55.2%) |
| - Free-form questions | 2,749 (44.8%) |
| - Questions with annotations | 5,261 (85.6%) |
| - Questions newly annotated | 736 (12.0%) |
| Unique number of images | 5,487 |
| Unique number of questions | 4,746 |
| Unique number of answers | 1,464 |
| Source datasets | 31 |
| - Existing VQA datasets | 19 |
| - Existing MathQA datasets | 9 |
| - Our newly annotated datasets | 3 |
| Visual context (image) classes | 19 |
| Maximum question length | 213 |
| Maximum answer length | 27 |
| Maximum choice number | 8 |
| Average question length | 15.6 |
| Average answer length | 1.2 |
| Average choice number | 3.4 |

Table 1: Key statistics of MATHVISTA.

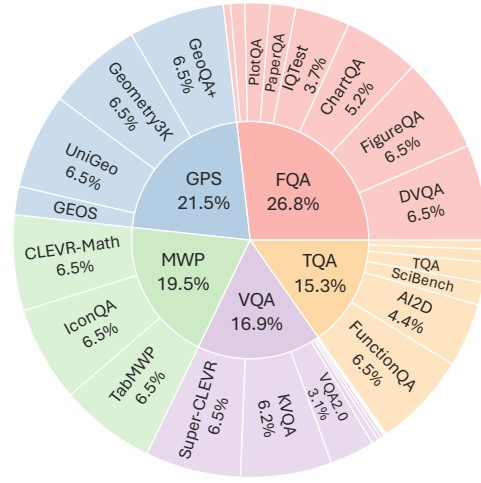

Figure 3: Source dataset distribution of MATHVISTA. FQA: figure question answering, GPS: geometry problem solving, MWP: math word problem, TQA: textbook question answering, VQA: visual question answering.

ing resources. The *test* set features the remaining 5,141 examples for standard evaluation. Notably, the answer labels for *test* will not be publicly released to prevent data contamination, and we will maintain an online evaluation platform. To ensure that each source dataset is well represented in *testmini* and to maintain a distribution in *testmini* closely resembling the whole set, we adopted this sampling strategy: (1) first, randomly sample questions with a threshold number of 4 for each source dataset; (2) then, randomly sample the remaining questions for each source dataset on its proportion in the entire set. The KL Divergence and Total Variation (TV) distance between the *testmini* set and the entire set are 0.008 and 0.035, respectively, suggesting that *testmini* is close to the distribution of the whole set. We also conducted several quality checks to address any unidentified errors.

## 2.5 DATA ANALYSIS

The main statistics of MATHVISTA are presented in Table 1. There are two types of questions: multiple-choice and free-form. Answers to free-form questions are categorized as integers, floating numbers, or lists. The large unique number of images, questions, and answers ensures pattern diversity in MATHVISTA. MATHVISTA is derived from 31 source datasets, including three newly annotated datasets to address the missing types of mathematical reasoning over specific visual contexts. Further details on data analysis are available in §E.

## 3 EXPERIMENTS

In this work, we aim to conduct qualitative and quantitative studies to provide a systematic evaluation of existing foundation models for mathematical reasoning in visual contexts using MATHVISTA.

## 3.1 EVALUATION PROTOCOLS

Recent LLMs and LMMs have been instructed to generate long responses in conventional settings instead of short text. Therefore, we propose a new strategy for benchmarking MATHVISTA, unlike using human-designed or template matching rules (Lu et al., 2022). The evaluation process consists of three stages: *response generation*, *answer extraction*, and *score calculation*. Initially, the baselines generate responses given the input query, which incorporates the task description, the question, the choices, and the metadata, using the template defined in Table 9 (§F.3). Next, the short answer text is extracted from the detailed response. We propose an answer extractor (§F.2) based on LLMs such as GPT-4, inspired by its remarkable ability for text processing (Wei et al., 2022b). A preliminary study of 200 examples shows that GPT-4 can extract the answer text with more than 99.5% accuracy. Finally, the extracted answer is normalized to a required answer format (e.g., an option letter or an integer), and the target metric scores are computed. Taking advantage of the fact that

| Model | Input | ALL | FQA | GPS | MWP | TQA | VQA | ALG | ARI | GEO | LOG | NUM | SCI | STA |
|---|---|---|---|---|---|---|---|---|---|---|---|---|---|---|
| *Heuristics baselines* | | | | | | | | | | | | | | |
| Random chance | - | 17.9 | 18.2 | 21.6 | 3.8 | 19.6 | 26.3 | 21.7 | 14.7 | 20.1 | 13.5 | 8.3 | 17.2 | 16.3 |
| Frequent guess | - | 26.3 | 22.7 | 34.1 | 20.4 | 31.0 | 24.6 | 33.1 | 18.7 | 31.4 | 24.3 | 19.4 | 32.0 | 20.9 |
| *Large Language Models (LLMs)* | | | | | | | | | | | | | | |
| Zero-shot ChatGPT | $Q$ only | 23.5 | 21.9 | 26.9 | 9.1 | 38.6 | 23.5 | 27.7 | 15.9 | 25.7 | 21.6 | 9.9 | 41.5 | 20.5 |
| Zero-shot GPT-4 | $Q$ only | 26.1 | 22.3 | 37.0 | 7.0 | 39.2 | 27.4 | 33.6 | 17.4 | 35.6 | 16.2 | 9.2 | 45.8 | 19.5 |
| Zero-shot Claude-2 | $Q$ only | 26.4 | 21.9 | 34.1 | 13.4 | 36.1 | 29.1 | 32.8 | 20.4 | 33.3 | 13.5 | 12.1 | 36.4 | 20.5 |
| 2-shot CoT Claude-2 | $Q$ only | 24.4 | 18.6 | 29.8 | 9.7 | 33.5 | 34.1 | 29.2 | 19.0 | 28.0 | 5.4 | 13.9 | 36.9 | 18.9 |
| 2-shot CoT ChatGPT | $Q$ only | 26.8 | 20.1 | 36.5 | 8.6 | 44.9 | 28.5 | 35.6 | 17.0 | 33.5 | 21.6 | 14.6 | 45.9 | 17.9 |
| 2-shot CoT GPT-4 | $Q$ only | 29.2 | 20.1 | 44.7 | 8.6 | 46.2 | 31.3 | 41.6 | 19.3 | 41.0 | 18.9 | 13.9 | 47.5 | 18.9 |
| 2-shot PoT ChatGPT | $Q$ only | 25.1 | 19.0 | 30.8 | 16.1 | 38.0 | 25.7 | 29.9 | 19.8 | 29.3 | 24.3 | 19.4 | 38.5 | 16.9 |
| 2-shot PoT GPT-4 | $Q$ only | 26.0 | 20.1 | 33.2 | 8.1 | 44.9 | 28.5 | 32.7 | 16.7 | 31.0 | 24.3 | 13.2 | 48.4 | 18.3 |
| *Augmented Large Language Models (Augmented-LLMs)* | | | | | | | | | | | | | | |
| 2-shot CoT Claude-2 | $Q, I_c, I_t$ | 33.2 | 26.0 | 31.7 | 35.5 | 48.1 | 30.2 | 32.4 | 32.3 | 33.0 | 16.2 | 17.4 | 54.9 | 36.2 |
| 2-shot CoT ChatGPT | $Q, I_c, I_t$ | 33.2 | 27.5 | 29.3 | 36.0 | 49.4 | 29.1 | 31.0 | 32.9 | 31.0 | 16.2 | 17.4 | 50.8 | 37.2 |
| 2-shot CoT GPT-4 | $Q, I_c, I_t$ | 33.2 | 27.9 | 31.7 | 31.2 | 51.9 | 28.5 | 33.5 | 30.9 | 32.2 | 13.5 | 12.5 | 58.2 | 37.9 |
| 2-shot PoT ChatGPT | $Q, I_c, I_t$ | 26.8 | 24.5 | 26.4 | 23.7 | 33.5 | 27.9 | 27.8 | 26.1 | 28.0 | 18.9 | 13.2 | 33.6 | 29.9 |
| 2-shot PoT GPT-4 | $Q, I_c, I_t$ | 33.9 | 30.1 | 39.4 | 30.6 | 39.9 | 31.3 | 37.4 | 31.7 | 41.0 | 18.9 | 20.1 | 44.3 | 37.9 |
| *Large Multimodal Models (LMMs)* | | | | | | | | | | | | | | |
| IDEFICS-9B-Instruct | $Q, I$ | 19.8 | 21.6 | 21.1 | 6.5 | 25.9 | 24.0 | 22.1 | 15.0 | 19.8 | 18.9 | 9.9 | 24.6 | 18.1 |
| mPLUG-Owl-LLaMA-7B | $Q, I$ | 22.2 | 22.7 | 23.6 | 10.2 | 27.2 | 27.9 | 23.6 | 19.2 | 23.9 | 13.5 | 12.7 | 26.3 | 21.4 |
| miniGPT4-LLaMA-2-7B | $Q, I$ | 23.1 | 18.6 | 26.0 | 13.4 | 30.4 | 30.2 | 28.1 | 21.0 | 24.7 | 16.2 | 16.7 | 25.4 | 17.9 |
| LLaMA-Adapter-V2-7B | $Q, I$ | 23.9 | 21.2 | 25.5 | 11.3 | 32.3 | 31.8 | 26.3 | 20.4 | 24.3 | 24.3 | 13.9 | 29.5 | 18.3 |
| LLaVAR | $Q, I$ | 25.2 | 21.9 | 25.0 | 16.7 | 34.8 | 30.7 | 24.2 | 22.1 | 23.0 | 13.5 | 15.3 | 42.6 | 21.9 |
| InstructBLIP-Vicuna-7B | $Q, I$ | 25.3 | 23.1 | 20.7 | 18.3 | 32.3 | 35.2 | 21.8 | 27.1 | 20.7 | 18.9 | 20.4 | 33.0 | 23.1 |
| LLaVA-LLaMA-2-13B | $Q, I$ | 26.1 | 26.8 | 29.3 | 16.1 | 32.3 | 26.3 | 27.3 | 20.1 | 28.8 | 24.3 | 18.3 | 37.3 | 25.1 |
| Multimodal Bard | $Q, I$ | 34.8 | 26.0 | 47.1 | 29.6 | 48.7 | 26.8 | 46.5 | 28.6 | 47.8 | 13.5 | 14.9 | 47.5 | 33.0 |
| GPT-4V (Playground) | $Q, I$ | 49.9 | 43.1 | 50.5 | 57.5 | 65.2 | 38.0 | 53.0 | 49.0 | 51.0 | 21.6 | 20.1 | 63.1 | 55.8 |
| *Human* | | | | | | | | | | | | | | |
| Human performance | $Q, I$ | 60.3 | 59.7 | 48.4 | 73.0 | 63.2 | 55.9 | 50.9 | 59.2 | 51.4 | 40.7 | 53.8 | 64.9 | 63.9 |

Table 2: Accuracy scores on the *testmini* subset of MATHVISTA. Input: $Q$: question, $I$: image, $I_c$: image caption, $I_t$: OCR text detected in the image. ALL: overall accuracy. Task types: FQA: figure question answering, GPS: geometry problem solving, MWP: math word problem, TQA: textbook question answering, VQA: visual question answering. Mathematical reasoning types: ALG: algebraic reasoning, ARI: arithmetic reasoning, GEO: geometry reasoning, LOG: logical reasoning, NUM: numeric commonsense, SCI: scientific reasoning, STA: statistical reasoning. The highest scores among models in each section and overall are highlighted in blue and red, respectively.

the instances in MATHVISTA are either multiple-choice questions for textual answers or free-form questions for numerical answers, accuracy scores are used as metrics for deterministic evaluation.

## 3.2 EXPERIMENTAL SETUP

We evaluate the models on MATHVISTA under three setups: (a) *Text-Only LLMs* including ChatGPT (OpenAI, 2022), GPT-4 (OpenAI, 2023a), and Claude-2 (Anthropic, 2023) in zero-shot and two-shot settings with Chain-of-Thought (CoT) (Wei et al., 2022b) and Program-of-Thought (PoT) (Chen et al., 2022b), (b) *Augmented-LLMs* where the LLMs are provided with additional visual information including the generated image captions from Multimodal Bard (Google, 2023) and the detected OCR text from EasyOCR (JaidedAI, 2020), (c) *LMMs* that include open-source models such as IDEFICS-9B (Laurençon et al., 2023), mPLUG-OWL-LLaMA-7B (Ye et al., 2023), miniGPT-4-LLaMA-2-7B (Zhu et al., 2023a), LLaMA-Adapter-V2-7B (Gao et al., 2023), InstructBLIP-Vicuna-7B (Dai et al., 2023), LLaVA-LLaMA-2-13B (Liu et al., 2024a), LLaVAR Zhang et al. (2023c), and proprietary models such as Bard and GPT-4V. Since GPT-4V does not offer API access, we resorted to manually evaluating it using the playground chatbot. We provide the prompts for LLMs and the hyperparameters used for LMMs in §F.

## 3.3 EXPERIMENTAL RESULTS

We compare the performance of several models, including Text-only LLMs, Augmented LLMs, and LMMs on MATHVISTA in Table 2. We include random chance (*i.e.*, one of the options in multiple-choice questions, and empty in the free-form questions) and frequency guess (§F.1) as naive baselines. Additionally, we established a human performance baseline using Amazon Mechanical Turk. Eligible human annotators must have a satisfactory annotating history, successfully pass qualification examples, and possess a high school degree or higher. We asked each annotator to complete five questions within 20 minutes. Further details can be found in §F.6.

Among text-only LLMs, all models outperform the random baselines, with the 2-shot GPT-4 using chain-of-thought (CoT) prompting achieving 29.2%. The limited performance of text-only LLMs suggests that our dataset requires models to reason within visual contexts for optimal results. When equipped with image captions and detected OCR text, augmented LLMs exhibit superior performance compared to their text-only counterparts on MATHVISTA. Specifically, the best-performing augmented LLM is the 2-shot GPT-4 employing program-of-thought (PoT) prompting, which scores 33.9%. This model generates Python programs for execution, thereby promoting rigorous reasoning.

On the LMM side, Multimodal Bard scores a 34.8% accuracy, which is only 58% of human performance at 60.3%. Notably, the best-performing GPT-4V model achieves 49.9%, marking a substantial 15.1% improvement over Bard; however, it still falls 10.4% short of human performance. These gaps highlight that there is a significant scope for further improvements on our benchmark. The open-source models (IDEFICS to LLaVA) achieve underwhelming performance on MATHVISTA. This can be attributed to their lack of math reasoning capabilities, text recognition (useful for math word problems), shape detection (useful for geometrical problems), and chart understanding. Notably, these models utilize different model architectures for processing the vision (e.g., OpenCLIP, CLIP, Vit-G) and language (e.g., LLaMA-1, LLaMA-2), different alignment strategies (e.g., MLP projection in LLaVA, Q-former in InstructBLIP, visual abstractor in mPLUGOwl), and instruction tuning data (e.g., 150K instruction-response pairs from LLaVA data, 3,500 instruction-response pairs from miniGPT-4). While fine-tuned with instruction-following data from text-rich images, LLaVAR does not perform well, indicating that strong text recognition abilities do not guarantee high performance on MATHVISTA, which requires comprehensive visual perception and mathematical reasoning. This underscores that there are immense possibilities for innovations in model, data, or training objectives to improve the zero-shot performance of LMMs on MATHVISTA.

## 3.4 FINE-GRAINED RESULTS

We also report fine-grained scores for a comprehensive study of the capabilities of existing models across different tasks (Table 2), mathematical reasoning abilities (Table 2, Figures 1, 33), visual context types (Figures 1, 34), and grade levels (Figure 35). Remarkably, GPT-4V surpasses most other baselines in various categories, with exceptions in problems related to logical reasoning and numeric commonsense reasoning. Notably, GPT-4V surpasses human performance not only in tasks like geometry problem solving (GPS), textbook question answering (TQA), and mathematical reasoning skills such as algebraic reasoning but also in visual contexts including function plots, geometry diagrams, scatter plots, and tables. Please refer to §G.2, §G.3, and §G.4 for more detailed analysis.

We perform an ablation study on the augmented LLMs and present the results in Table 36 (see §G.5). The gap in the performance of the Augmented LLMs can be attributed to poor image captions, which may not adequately describe the math in visual contexts, the inability of the OCR to detect shapes useful for geometrical reasoning, and the lack of mathematical reasoning capabilities. An in-depth study of GPT-4V can be found in §H.

## 3.5 QUALITATIVE ANALYSIS

**Success and failure analysis of Multimodal Bard.** In §3.3, we observe that Multimodal Bard achieves the highest average accuracy on MATHVISTA. Here, we analyze its predictions through human evaluation to understand its mode of success and failure. To do so, we ask the human workers, from Amazon Mechanical Turk (AMT), to study Bard's predictions given the math question, its associated image, and the ground truth from MATHVISTA dataset for 250 instances. Specifically, workers were instructed to decide whether the predictions contained the correct answer with the

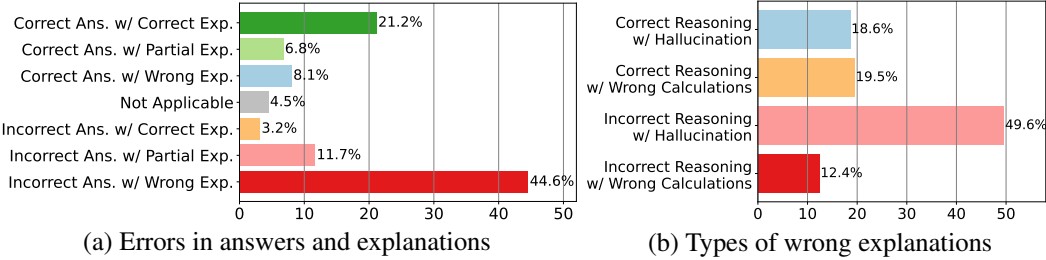

(a) Errors in answers and explanations        (b) Types of wrong explanations

Figure 4: Error analysis of Bard results: (a) presents errors in answers and explanations; (b) delves into the details of wrong explanations. Notations: "Answer" is "Ans.", "Explanation" is "Exp.", "Partially Correct" is "Partial", and "Not applicable" refers to unanswerable or indeterminate cases.

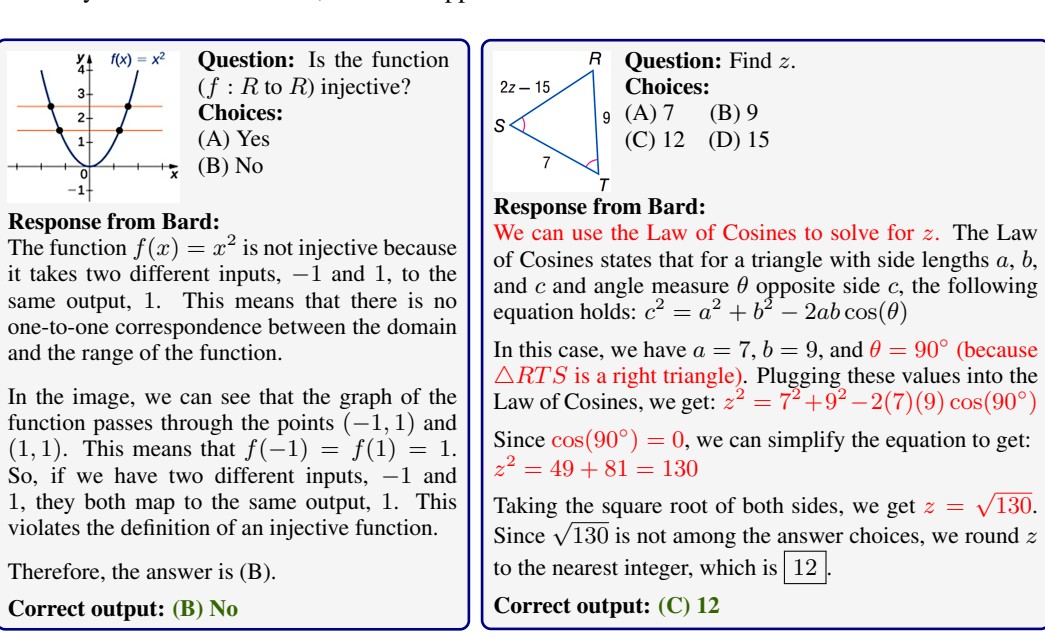

(a) **Correct answer** and **explanation**        (b) **Correct answer** but **wrong explanation**

Figure 5: Two examples from Bard. In (b), Bard does not correctly identify the geometry symbols and relationships. The accurate correct should identify the isosceles triangle and apply its properties.

correct explanation. If the workers find that the model's explanation is incorrect, they had to choose whether the wrong explanation was due to various failure modes such as incorrect reasoning with *hallucination* or wrong calculations. In our setup, we define hallucination as an introduction of incorrect facts, in the model explanation, that is not mentioned in the context of the image or question (e.g., in Figure 39 and Figure 40). More details can be found in §F.7.

We present the distribution of the quality of Bard's predictions, judged by the human annotators, in Figure 4 (a). We find that 44.6% of the Bard's predictions had incorrect answers with incorrect explanations. Interestingly, we observe that Bard responds with partial (6.8%) or completely (8.1%) incorrect explanations despite giving the correct answer to the input image and question, highlighting its failure to reach the correct answer for the wrong reasons. In Figure 4 (b), we present the distribution over possible reasons when Bard provides incorrect explanations. Notably, we find that 49.6% of its responses contain hallucinations. Our analysis highlights that hallucination is a major source of errors in the generative foundation models (Lu et al., 2023c; Ji et al., 2023). We also observe that the model responds with correct reasoning but either hallucinates (18.6%) or performs wrong calculations (19.5%) leaving an overall impression of being a wrong explanation.

**Qualitative examples of Multimodal Bard.** We also present a few qualitative examples of Bard's predictions. In Figure 5 (a), we find that Bard generates the correct answer with the correct explanation, including detecting the correct function (*i.e.*, $f(x) = x^2$) and analyzing its properties (*i.e.*, injective) to answer the question. However, in Figure 5 (b), we observe that the model provides the

| Laps driven | |
|---|---|
| **Day** | **Number of laps** |
| Thursday | 53 |
| Friday | 53 |
| Saturday | 53 |
| Sunday | 55 |
| Monday | 55 |

**Question:** A race car driver kept track of how many laps he drove in the past 5 days. What is the mode of the numbers?

**Detected OCR text:**
([55, 9], Laps driven),
([21, 34], Day),
([81, 37], Number of laps),
([1, 64], Thursday), ([129, 67], 53),
([3, 99], Friday), ([129, 99], 53),
([1, 126], Saturday), ([129, 129], 53),
([3, 161], Sunday), ([129, 161], 55),
([1, 188], Monday), ([129, 191], 55)

**Response from GPT-4:**

```
def mode(laps):
    return max(set(laps), key
    =laps.count)

laps = [53, 53, 53, 55, 55]
print(mode(laps))
```

**Correct output: 53**

(a) **Correct answer** and **code**

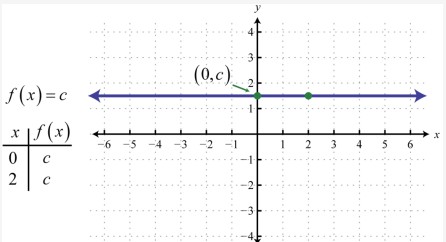

**Question:** What is this function most likely be?
**Choices:**
(A) polynomial (B) trigonometric
(C) exponential (D) logarithmic

**Caption from Bard:**
The image shows a graph of a function $f(x) = c$ on a graph paper. The $x$ axis is the horizontal axis and the $y$ axis is the vertical axis. The graph is a horizontal line that passes through the point $(0, c)$. ... The text "$y = 3$" means that the $y$-value of all points on the graph is 3. The text "$(0, c)$" means that the point $(0, c)$ is on the graph. The text "$f(x) = c$" means that the function $f(x)$ is equal to $c$ for all values of $x$. ... The graph of a constant function is a horizontal line. In this case, the horizontal line passes through the point $(0, 3)$, so the constant value of the function is 3. ...

**Response from GPT-4:**
The function shown in the image is a constant function, which is a type of polynomial function. Thus, the answer is A.

**Correct output: (A) polynomial**

(b) **Correct answer** with **partially correct outputs**

Figure 6: Two examples from GPT-4. GPT-4 depends on the qualities of the generated caption and detected OCR texts. In (b), some information is incorrect, even though the final answer is correct.

correct answer (*i.e.*, 12) but with an incorrect explanation (*i.e.*, using the law of cosines when the question requires an understanding of the properties of isosceles triangles). We present more examples in §G.9. Overall, our analysis of Bard highlights its modes of failure in detail, which could guide future foundation model design to address these issues.

**Qualitative examples of Augmented GPT-4.** Augmented with external visual models, CoT GPT-4 and PoT GPT-4 are able to achieve comparable performance with Multimodal Bard. As shown in Figure 6 (a), provided with the accurate OCR text detected in the image, PoT GPT-4 accurately understands the structural information of the image and generates a code snippet to perform precise statistical reasoning. In Figure 6 (b), the caption provides some accurate descriptions of the image (e.g., $f(x) = c$) along with hallucination (e.g., $y = 3$, the line passes through $(0, 3)$) caused by the external Bard model. Although CoT GPT-4 predicts the correct answer given the partially correct information, the qualities of visual information augmented by external models have an impact on the accurate visual perception and thus the final mathematical reasoning performance. Examples in §G.10 show failure cases due to hallucination caused by external visual models.

## 4 CONCLUSION

In this work, we introduce MATHVISTA, a benchmark designed to systematically analyze the mathematical reasoning capabilities of state-of-the-art models in visually complex scenarios. Our evaluation of 12 prominent foundation models highlights that significant advancements have been made, especially with the GPT-4V model. However, a substantial gap of 10.4% still exists between GPT-4V, the best-performing model, and human performance. This disparity sets a clear direction for future research, emphasizing the need for models that can seamlessly integrate mathematical reasoning with visual comprehension. Moreover, our exploration of GPT-4V's self-verification, self-consistency, and chatbot interactions offers valuable insights for future investigations.

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

CONTENTS

# A    RELATED WORK

**Mathematical reasoning benchmarks.**    Recently, numerous benchmarks (Amini et al., 2019; Cobbe et al., 2021; Mishra et al., 2022; Frieder et al., 2023) have been proposed to evaluate the mathematical reasoning capabilities of Large Language Models (LLMs). However, most of these are textual only (Lu et al., 2023c), despite a substantial amount of mathematical information and reasoning being encapsulated in visual modalities. Meanwhile, some datasets exhibit performance saturation; for instance, GPT-4 achieves 92.0% accuracy on GSM8K (Cobbe et al., 2021), a dataset of grade-school mathematics questions. On the other hand, the recent rapid advancement of Large Multimodal Models (LMMs) necessitates the establishment of robust multimodal benchmarks. However, current multimodal reasoning benchmarks provide limited coverage of rigorous and scientific domains (Antol et al., 2015; Kembhavi et al., 2016; Kahou et al., 2017; Mathew et al., 2022), which are key components for creating general-purpose AI assistants. To bridge this gap, it is crucial to develop a robust math reasoning dataset that integrates visual contexts.

**Vision-language reasoning benchmarks.**    High-quality evaluation datasets and benchmarks are a cornerstone for assessing the progress of machine learning models to solve real-world tasks Liao et al. (2021). Prior studies such as VQA (Antol et al., 2015; Goyal et al., 2017), VizWiz (Gurari et al., 2018), and ParsVQA-Caps (Mobasher et al., 2022) assess the general-purpose visual question answering abilities of the LMMs, with or without task-specific training, on open-ended questions about images. In addition, there are several works that focus on evaluating specific skills of the LMMs beyond natural scenes, such as abstract scenes and shapes) (Antol et al., 2015; Lu et al., 2021b; Ji et al., 2022), geometry diagrams (Seo et al., 2015; Lu et al., 2021a; Chen et al., 2022a; Cao & Xiao, 2022), figures and charts (Methani et al., 2020; Masry et al., 2022; Kahou et al., 2017; Chang et al., 2022; Kafle et al., 2018), documents (text in images) (Singh et al., 2019; Mathew et al., 2022; Liu et al., 2023c), or synthetic images (Dahlgren Lindström & Abraham, 2022; Li et al., 2023c; Bitton-Guetta et al., 2023). Besides, there has been significant progress on developing datasets to judge LMMs on skills that require external knowledge (Schwenk et al., 2022; Shah et al., 2019), common sense reasoning (Zellers et al., 2019; Yin et al., 2021), scientific-knowledge (Lu et al., 2022; Kembhavi et al., 2017; 2016), medical understanding (Zhang et al., 2023b; Lau et al., 2018). In this work, we create new datasets (IQTest, FunctionQA, PaperQA) and subsequently design a benchmark for holistic evaluation of the math reasoning capabilities of the LMMs.

**Generative foundation models and their evaluation.**    Recently, there has been a surge of generative foundation models (Bommasani et al., 2021) that are trained on web-scale data, such as GPT-3, ChatGPT, GPT-4, Claude, LLaMA, LLaMA-Adapter (Brown et al., 2020; OpenAI, 2022; 2023a; Anthropic, 2023; Touvron et al., 2023; Zhang et al., 2024), with the ability to solve a wide range of downstream tasks (Wei et al., 2022a) without any task-specific finetuning. Prior work has focused on evaluating their abilities to respond to the queries from various disciplines, grounded in text, such as QA, math, medicine, coding and science (Bubeck et al., 2023; Nori et al., 2023; Chen et al., 2021; Fu et al., 2023; Sun et al., 2023; Wang et al., 2023b; Huang et al., 2024; 2022; Liu et al., 2023a; Zhang et al., 2024; Gu et al., 2024). Prior work, such as PixStruct (Lee et al., 2023), MatCha (Liu et al., 2022), and UniChart (Masry et al., 2023), has focused on developing specialized pretraining recipe for improved math and chart reasoning in visual contexts.

On the vision-language side, there are several generative foundation models such as LLaVA, miniGPT4, InstructBLIP, Flamingo, LLaMA-Adapter V2, Multimodal Bard, Gemini, SPHINX-X, Claude 3 (Liu et al., 2024a; Zhu et al., 2023a; Dai et al., 2023; Alayrac et al., 2022; Awadalla et al., 2023; Gao et al., 2023; Google, 2023; Team et al., 2023; Gao et al., 2024; Anthropic, 2024) that are trained on vast amount of paired (Schuhmann et al., 2022; Sharma et al., 2018; Lin et al., 2014) and interleaved image-text data (Zhu et al., 2023b). In addition, there has been recent development on specialized versions of these LMMs for document understanding where visual contexts require text recognition, math understanding being one of them (Zhang et al., 2023c; Ye et al., 2023). In recent times, there have been several works, such as Visit-Bench, LVLM-eHub, MMBench (Bitton et al., 2023; Yu et al., 2023; Liu et al., 2023b; Xu et al., 2023; Shao et al., 2023), that assess their instruction-following and reasoning capabilities. As the generative foundation models become more relevant to real-world applications, unlike prior work, we propose MATHVISTA to benchmark their capabilities of math reasoning (logical, arithmetic, statistical) on a diverse set of visual contexts (word problems in images, natural scenes, geometrical shapes, and plots).

**Recent work of LLM prompting and GPT-4V.** We have witnessed the remarkable abilities of large language models (LLMs), and their reasoning capabilities are further enhanced by promoting approaches such as chain-of-thought (CoT) (Wei et al., 2022b), program-of-thought (PoT) (Chen et al., 2022b), and inductive reasoning (Wang et al., 2023a; Tan & Motani, 2023). For example, the feasibility of using LLMs to solve the Abstraction and Reasoning Corpus (ARC) challenge has been verified using zero-shot, few-shot, and context-grounded prompting (Tan & Motani, 2023). In this paper, we evaluate LLMs using zero-shot, few-shot, CoT prompting, PoT prompting, as well as tool-augmented prompting, to explore their potential in solving mathematical reasoning in visual contexts on MATHVISTA. Program-aided methods are widely used for mathematical reasoning due to their advancements in precise logical reasoning and arithmetic calculations (Drori & Verma, 2021; Tang et al., 2022; Drori et al., 2022). In this work, we have developed the LLM baselines with PoT.

Recently, OpenAI released GPT-4V, the multimodal version of GPT-4, which shows promising performance in vision-language reasoning. However, the fine-grained study of its strengths and limitations still remains underexplored. The recent work (Zhang et al., 2023a) contributes pioneering efforts in this field, studying whether large multimodal models (LMMs), like GPT-4V, execute vision and language tasks consistently or independently. As concurrent work, our paper provides, for the first time, a comprehensive quantitative and qualitative study of GPT-4V and other LLMs in mathematical reasoning within visual contexts.

## B LIMITATIONS OF THE BENCHMARK

Our benchmark, MATHVISTA, makes significant contributions by combining mathematical and visual tasks, a domain where existing models like GPT-4V have shown promise but also face challenges, especially in complex figure understanding and rigorous reasoning. While we have made strides in evaluating model performance, we acknowledge several limitations.

One limitation is the dataset coverage. While MATHVISTA encompasses a broad spectrum of tasks and visual contexts, there may be gaps in the representation of certain types of mathematical problems and visuals. Furthermore, the dataset's focus on mathematical reasoning within visual contexts, spanning specific domains like science and college-level math, necessitates a more labor-intensive process for collecting high-quality data compared to textual-only or general-purpose datasets. Thus, the scalability and generalizability of our benchmark to other domains remain a concern. Annotations were sourced from original data providers, resulting in only 85.6% of examples (Table 1) having annotations. Due to the heterogeneity of these sources, annotations lack a unified format and structure. For example, the annotations could be logic forms of the problem parsing from Geometry3K (Lu et al., 2021a), natural language solutions from TabMWP (Lu et al., 2023b), and theorems from TheoremQA (Chen et al., 2023). Given the rapid development in foundation models, our study focused exclusively on the most recent and prominent models.

In future iterations, our benchmark will be beneficial to encompass a broader array of problems and visual contexts, while also providing unified and comprehensive annotations. Our benchmark is part of an ongoing research process, and we are committed to maintaining the datasets, such as refining the potential data noise, in response to the community feedback. Also, we are committed to evolving the leaderboard in response to new models.

In conclusion, while there are limitations to our current approach, MATHVISTA represents a significant step forward in the field. We are dedicated to continuously improving our benchmark to better understand and enhance the capabilities of AI in mathematical and visual reasoning.

# C DATA COLLECTION GUIDELINES

## C.1 MATHEMATICAL REASONING DEFINITION

Seven mathematical reasoning types are defined in Table 3.

| Math Reasoning | Description |
|---|---|
| Arithmetic Reasoning (34.1%) | It covers the *fundamental operations* such as addition, subtraction, multiplication, division, and understanding of n*umber properties*. It may also include the ability to interpret numerical data in different forms. |
| Statistical Reasoning (30.5%) | It focuses on *data interpretation* and *analysis*, including measures (mean, median, mode), dispersion metrics (standard deviation, range), probability concepts, regression, correlation, and data inferences. It also identifies trends, outliers, and patterns. |
| Algebraic Reasoning (28.5%) | It encompasses understanding *variables*, *equations*, and the manipulation of *expressions* with polynomials and exponents. It also covers solving simple to complex equations, and grasping functions, their properties, and graphical depictions. |
| Geometry Reasoning (23.3%) | It emphasizes *spatial* understanding, analysis of 2D and 3D *figures*, and reasoning about their *shapes, sizes, and relationships*. It includes symmetry, congruency, similarity, area, volume, and transformations. |
| Numeric common sense (14.0%) | It involves intuitive understanding of *daily numerical concepts*, including understanding time differences, numerical judgment, and estimates. It covers temporal reasoning, spatial numeric assessments, and practical uses like budgeting and time reading. |
| Scientific Reasoning (10.7%) | It deals with the application of mathematical concepts in *scientific contexts*. This includes scientific notations, formula use, understanding rates, proportions, and percentages in practical situations, and problem-solving in scientific inquiries. |
| Logical Reasoning (3.8%) | It focuses on *critical thinking* and *deduction* from provided information, including pattern recognition, sequence understanding, predictions, and statement evaluation. Key components include premises, conclusions, and the use of abstract reasoning. |

Table 3: Definitions and proportions of seven mathematical reasoning categories in MATHVISTA.

## C.2 MATHEMATICAL REASONING EXAMPLES

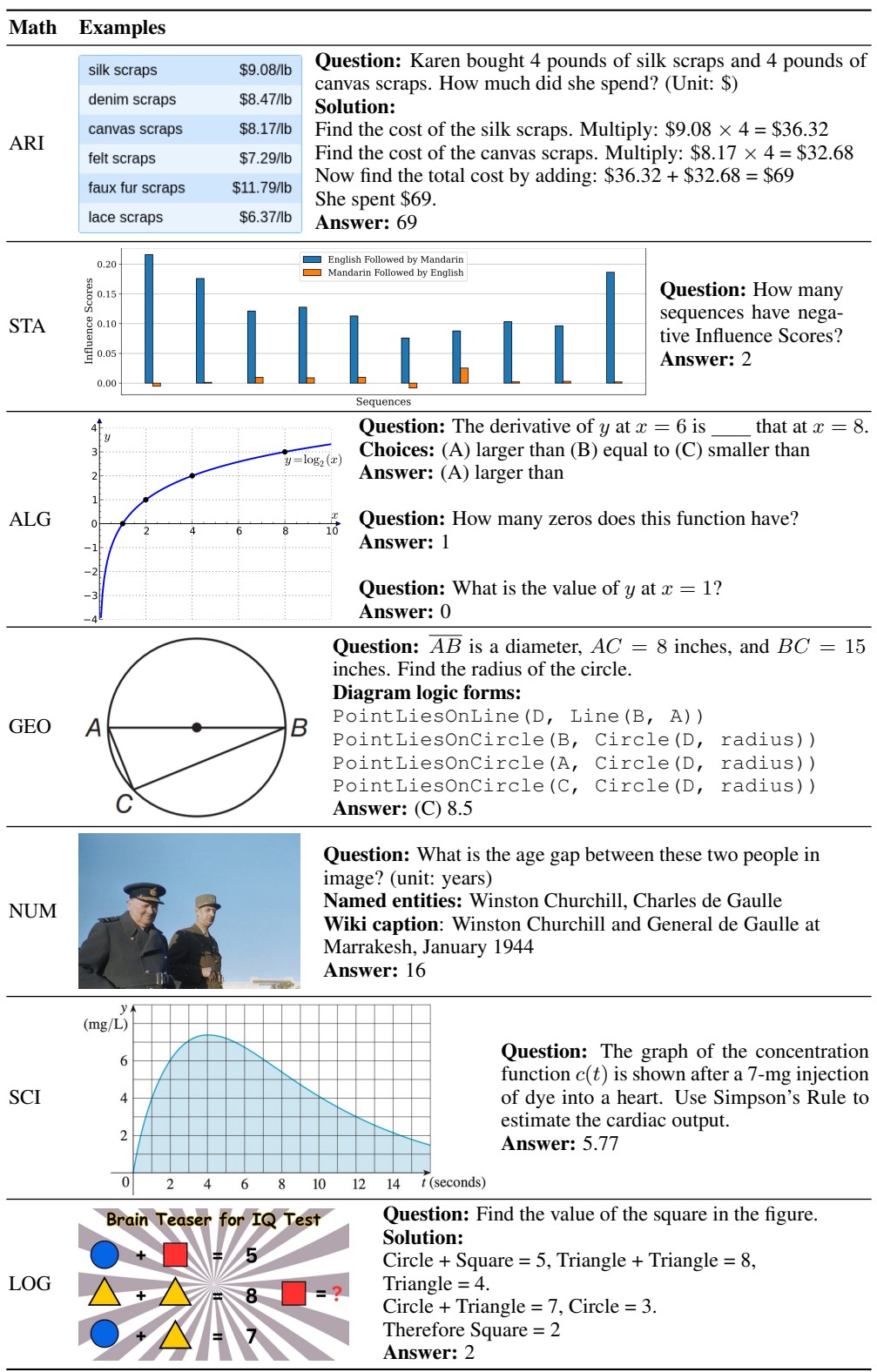

Table 4: Examples of seven mathematical reasoning categories in MATHVISTA.

## C.3 VISUAL CONTEXT TYPES

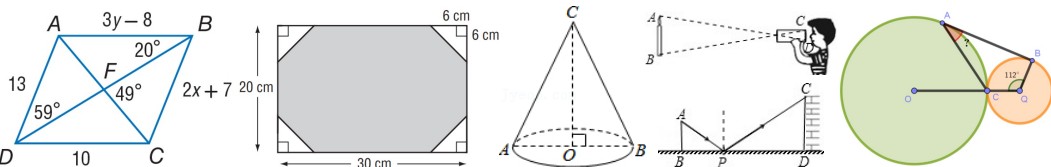

Figure 7: Examples of the visual context for the *geometry diagram* type.

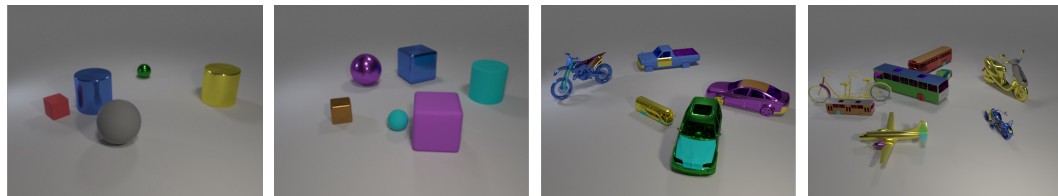

Figure 8: Examples of the visual context for the *synthetic scene* type.

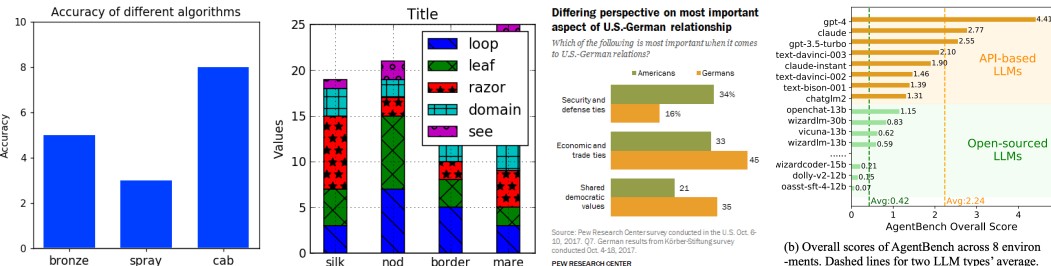

Figure 9: Examples of the visual context for the *bar chart* type.

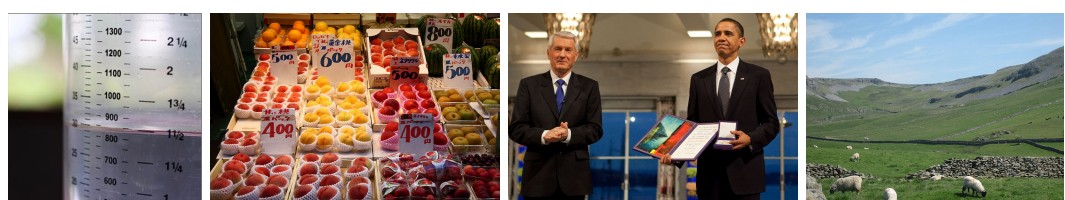

Figure 10: Examples of the visual context for the *natural image* type.

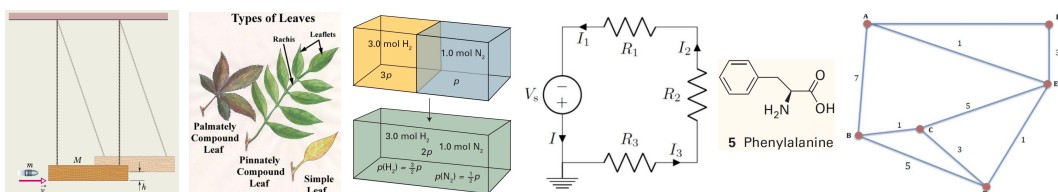

Figure 11: Examples of the visual context for the *scientific figure* type.

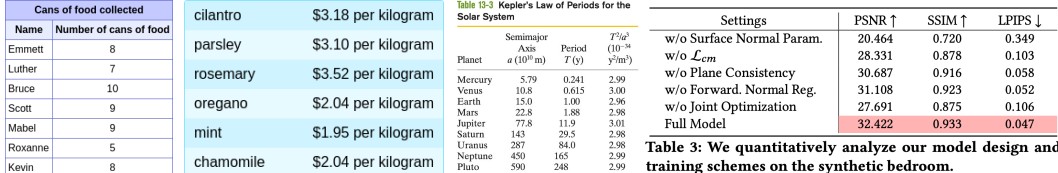

Figure 12: Examples of the visual context for the *table* type.

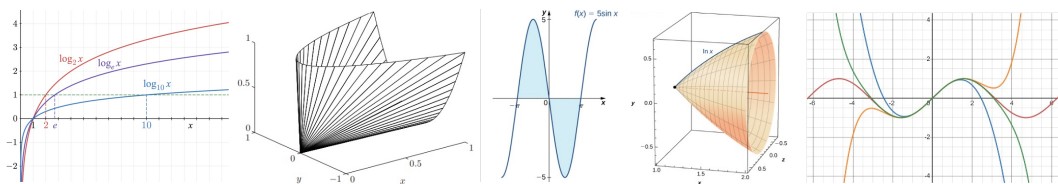

Figure 13: Examples of the visual context for the *function plot* type.

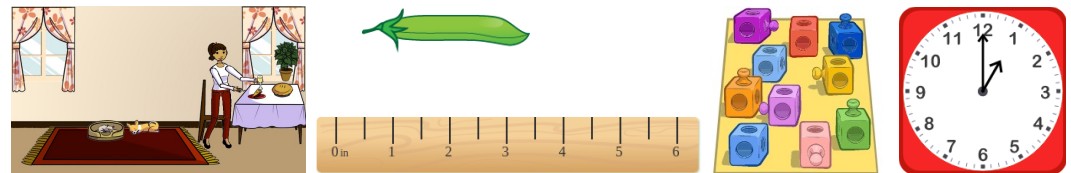

Figure 14: Examples of the visual context for the *abstract scene* type.

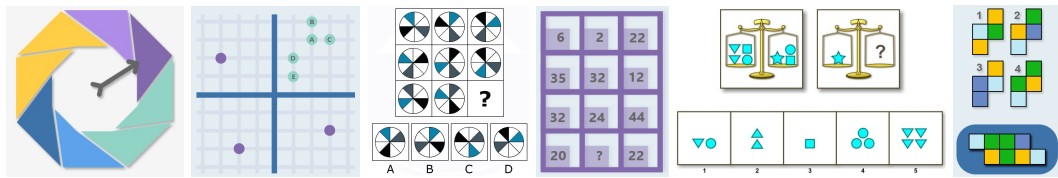

Figure 15: Examples of the visual context for the *puzzle test* type.

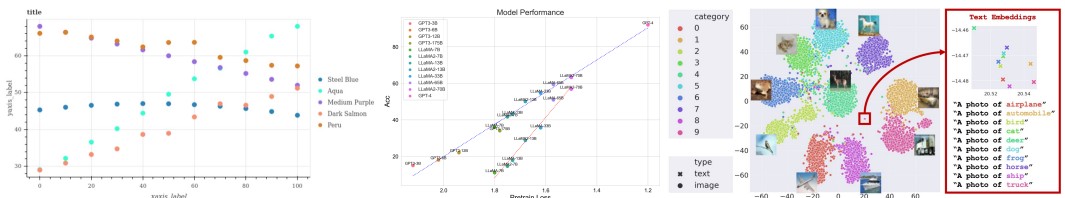

Figure 16: Examples of the visual context for the *scatter plot* type.

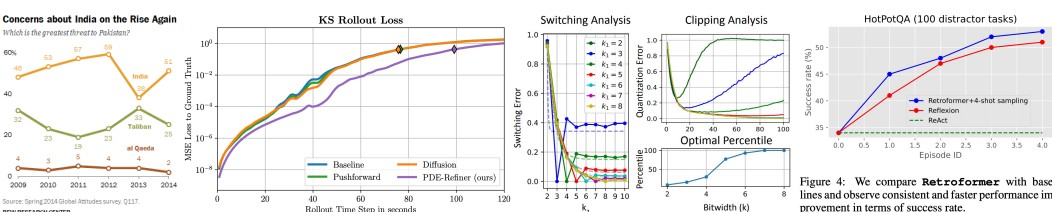

Figure 17: Examples of the visual context for the *line plot* type.

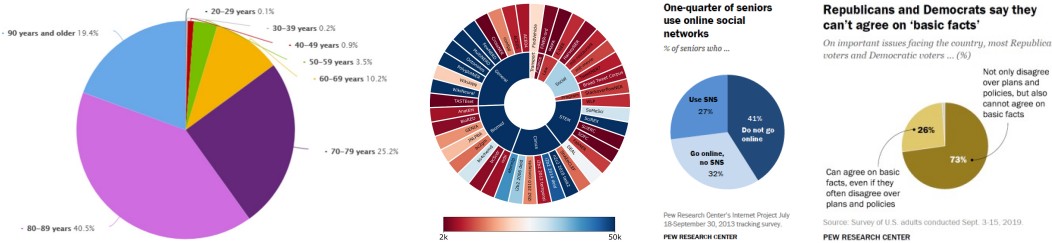

Figure 18: Examples of the visual context for the *pie chart* type.

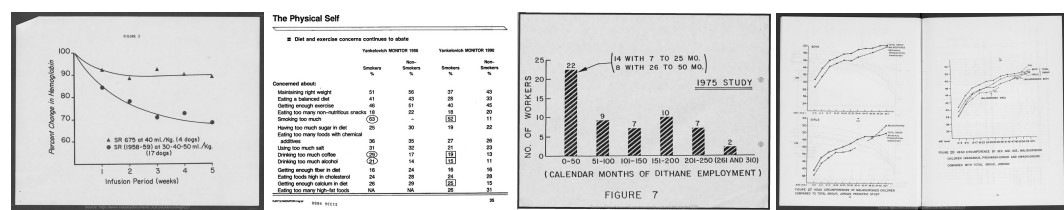

Figure 19: Examples of the visual context for the *document image* type.

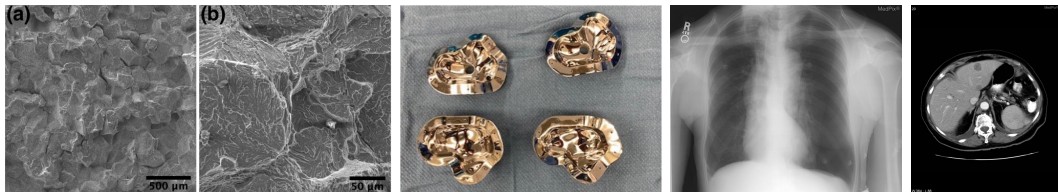

Figure 20: Examples of the visual context for the *medical image* type.

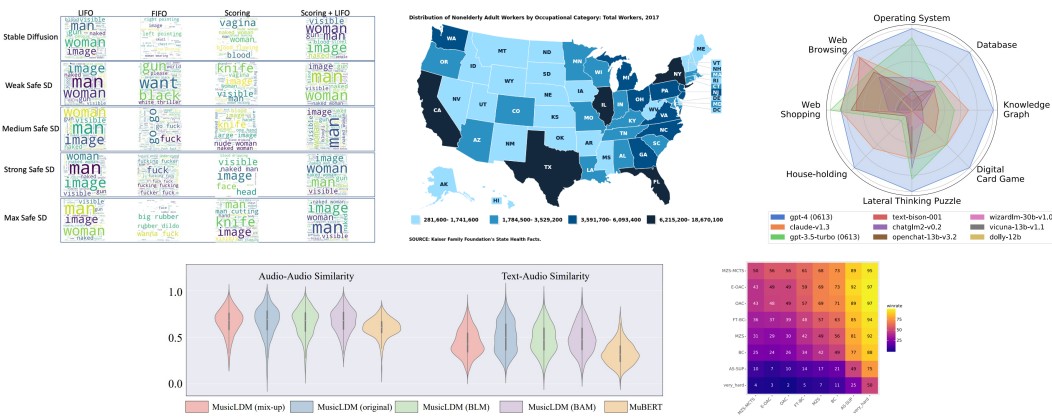

Figure 21: Examples of the visual context for *other* types, including word cloud, map chart, radar chart, violin plot, and heatmap chart.

## C.4 SOURCE DATASET SUMMARY

The source datasets are summarized in Table 5.

| Dataset | Category | Task | Context | Math Skill |
|---|---|---|---|---|
| IQTest (Ours) | Math-Targeted | FQA | Puzzle Test | Logical, Arithmetic |
| PaperQA (Ours) | Math-Targeted | FQA | Charts and Plots | Scientific |
| FunctionQA (Ours) | Math-Targeted | TQA | Function Plot | Algebraic |
| Geometry3K (2021a) | Math-Targeted | GPS | Geometry Diagram | Geometry, Algebraic |
| GeoQA+ (2022) | Math-Targeted | GPS | Geometry Diagram | Geometry, Algebraic |
| GEOS (2015) | Math-Targeted | GPS | Geometry Diagram | Geometry, Algebraic |
| UniGeo (2022a) | Math-Targeted | GPS | Geometry Diagram | Geometry, Algebraic |
| CLEVR-Math (2022) | Math-Targeted | MWP | Synthetic Scene | Arithmetic |
| IconQA (2021b) | Math-Targeted | MWP | Abstract Scene | Arithmetic |
| TabMWP (2023b) | Math-Targeted | MWP | Table | Statistical, Arithmetic |
| SciBench (2023b) | Math-Targeted | TQA | Scientific Figure | Scientific |
| TheoremQA (2023) | Math-Targeted | TQA | Scientific Figure | Scientific |
| ChartQA (2022) | General VQA | FQA | Charts and Plots | Statistical |
| FigureQA (2017) | General VQA | FQA | Charts and Plots | Statistical |
| DVQA (2018) | General VQA | FQA | Bar Chart | Statistical |
| MapQA (2022) | General VQA | FQA | Map Chart | Statistical |
| PlotQA (2020) | General VQA | FQA | Scatter Plot | Statistical |
| DocVQA (2022) | General VQA | FQA | Document Image | Statistical |
| AI2D (2016) | General VQA | TQA | Scientific Figure | Scientific |
| ScienceQA (2022) | General VQA | TQA | Scientific Figure | Scientific |
| TQA (2017) | General VQA | TQA | Scientific Figure | Scientific |
| A-OKVQA (2022) | General VQA | VQA | Natural Image | Arithmetic, Numeric |
| KVQA (2019) | General VQA | VQA | Natural Image | Arithmetic, Numeric |
| ParsVQA-Caps (2022) | General VQA | VQA | Natural Image | Arithmetic, Numeric |
| TextVQA (2019) | General VQA | VQA | Natural Image | Arithmetic, Numeric |
| VizWiz (2018) | General VQA | VQA | Natural Image | Arithmetic, Numeric |
| VQA2.0 (2017) | General VQA | VQA | Natural Image | Arithmetic, Numeric |
| PMC-VQA (2023b) | General VQA | VQA | Medical Image | Scientific |
| VQA-RAD (2018) | General VQA | VQA | Medical Image | Scientific |
| Super-CLEVR (2023c) | General VQA | VQA | Synthetic Scene | Arithmetic |
| VQA-AS (2015) | General VQA | VQA | Abstract Scene | Arithmetic |

Table 5: Summary of the 31 different source datasets in MATHVISTA. Among these, FunctionQA, IQTest, and PaperQA are our newly annotated datasets. The table provides details on their category, task, visual context, and primary mathematical reasoning skill types.

# D    DATA COLLECTION DETAILS

## D.1    AUTOMATIC SELECTION OF MATHEMATICAL PROBLEMS

most, least, fewest more, less, fewer, largest, smallest, greatest, larger, smaller, greater, highest, lowest, higher, lower, increase, decrease, minimum, maximum, max, min, mean, average, median, total, sum, add, subtract, difference, quotient, gap, half, double, twice, triple, square, cube, root, approximate, approximation, triangle, rectangle, circle, square, cube, sphere, cylinder, cone, pyramid, multiply, divide, percentage, percent, ratio, proportion, fraction, rate

Table 6: Dictionary of quantity words used for the automatic selection of questions likely to involve mathematical reasoning.

## D.2    HUMAN LABELING OF MATHEMATICAL PROBLEMS

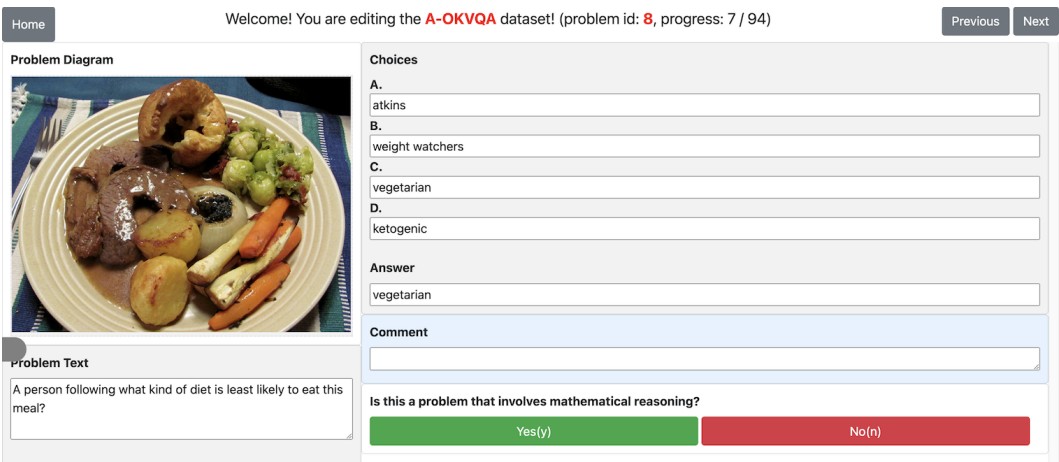

Figure 22: GUI for labeling if a problem involves mathematical reasoning.

We are compiling a dataset that incorporates image context and involves mathematical reasoning (MathQA in visual contexts). We have gathered a set of examples in which some involve mathematical reasoning, while others do not.

In our task, a question can be classified as a mathematical problem if it

- Involves numbers or symbols in the question text or the image context, AND requires further operations or transformations to be performed on them to reach a solution.
- Involves more complex forms of mathematical reasoning, including logical reasoning, abstract thought, and understanding of patterns.

Based on the definition above, a problem is classified as a negative example (NOT involving mathematical reasoning) if it:

- Does not involve any numbers or quantity words, OR
- Involves only counting, reading, or recognizing numbers, OR
- Relies solely on factual information, such as recalling years and dates.

Table 7: Instructions for human annotators to identify if a problem involves mathematical reasoning.

We developed an annotation tool, as illustrated in Figure 22, to enable expert annotators to label problems that involve mathematical reasoning. Annotators were trained using detailed instructions,

as shown in Table 7, along with a variety of examples—positive ones that involve mathematical reasoning and negative ones that do not. We provided three labeling options:

- *Yes* - This indicates that the problem involves mathematical reasoning.
- *No* - This indicates that the problem does not involve mathematical reasoning.
- *Unsure* - This option should be selected if it is uncertain whether the problem involves mathematical reasoning. (Annotators are advised to use this option sparingly.)

They may leave comments if they find anything incorrect or offensive for removal at a later stage.

In our study, we employed the Fleiss Kappa score to conduct an inter-annotator agreement analysis among three annotators tasked with labeling examples based on mathematical reasoning. The Fleiss Kappa score is a statistical measure used to evaluate the reliability of agreement between multiple raters, providing a quantifiable metric to assess the consistency across different annotators. A score of 1 indicates perfect agreement, while a score of 0 suggests no agreement beyond what would be expected by chance. Our analysis yielded a Fleiss Kappa score of 0.775, indicating a substantial level of consistency among the annotators. This high degree of agreement underscores the reliability of our annotation process and affirms the quality of the labeled data generated for our study.

### D.3   ANNOTATING THREE NEW DATASETS

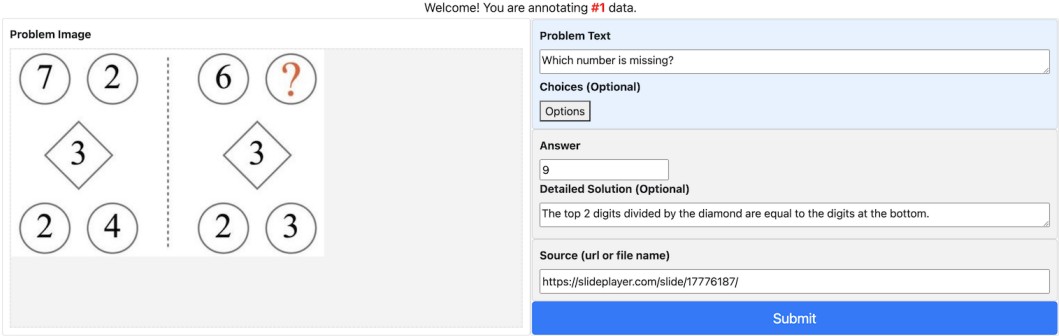

Figure 23: GUI for annotating our new source datasets.

### D.4   HUMAN LABELING OF MATHEMATICAL REASONING

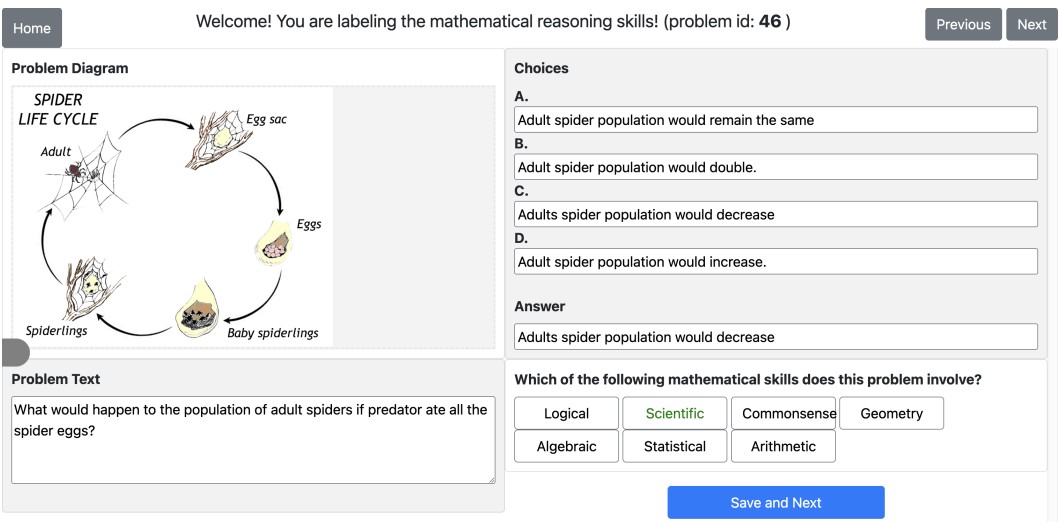

Figure 24: GUI for labeling mathematical reasoning skills.

## E MORE DATASET ANALYSIS

**Question distribution.** Apart from English questions, MATHVISTA contains 6.57% non-English questions, including languages such as Chinese and Persian. The multilingual feature necessitates that models be capable of understanding and processing multiple languages to ensure accurate results across the dataset. As illustrated in Table 3, the average number of words in English questions within MATHVISTA is 15.58, while the maximum number of words in a question reaches 213.

Figure 25 further elucidates the distribution of word counts, highlighting the diverse patterns of questions. MATHVISTA features two types of questions: multiple-choice questions and free-form questions. For multiple-choice questions, the average number of choices is 3.4, while the maximum number of choices is 8. In the case of free-form questions, answers can be integers, floating-point numbers, or lists, which can be converted into a standard format. The standard settings in question and answer types facilitate consistent accuracy evaluation for existing models.

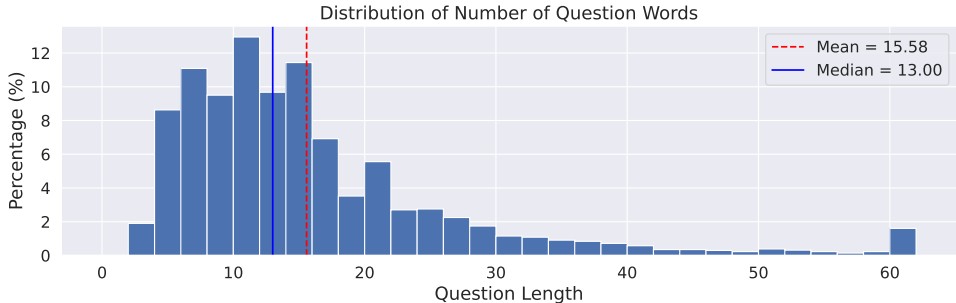

Figure 25: The distribution of the number of words per question in MATHVISTA. Questions with a length greater than 60 are categorized as 61 for visualization simplicity.

**Dataset category and task type.** Source datasets in MATHVISTA can be categorized into two types: math-targeted VQA datasets, which are originally proposed for assessing mathematical reasoning, and general VQA datasets, which address visual reasoning in everyday scenarios. The distribution proportions of these two categories (55.4% vs. 44.6%, as illustrated in Figure 26) within MATHVISTA enable a balanced examination of mathematical reasoning in both domain-specific and general-purpose applications. The distribution of the five tasks contained within MATHVISTA is visualized in Figure 27. The relatively balanced distribution of these tasks enhances the benchmarking robustness that our dataset provides.

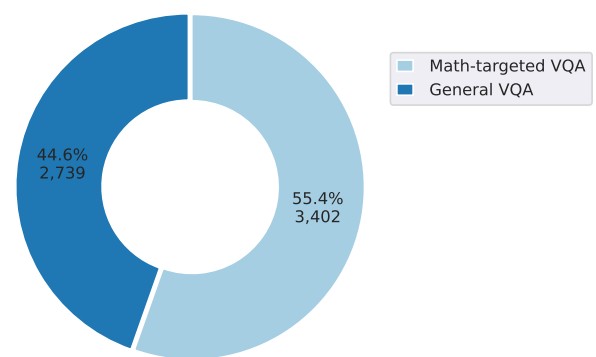

Figure 26: Category distribution of problems within MATHVISTA.

**Grade level.** The datasets within MATHVISTA are categorized into four distinct grade levels: *elementary school*, *high school*, *college*, and *not applicable*, each representing a different level of reasoning complexity and contextual application. The *elementary school* category aligns with the typical mathematical curriculum of elementary education, introducing basic topics such as arithmetic operations and introductory geometry. *High school* level questions delve into more complex

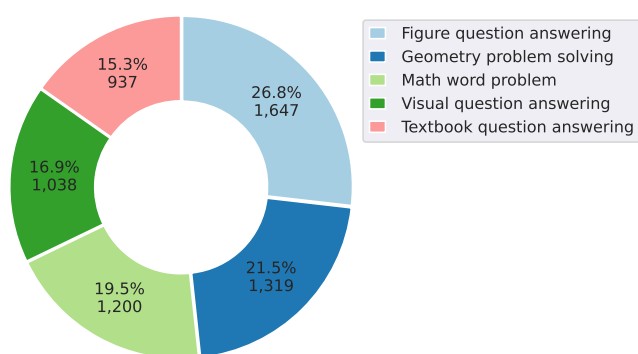

Figure 27: Task type distribution of problems within MATHVISTA.

mathematical concepts such as algebra, geometry, and introductory calculus. The *college* category encapsulates the highest level of complexity, featuring questions on advanced mathematical and scientific concepts like calculus, linear algebra, and physics. Questions without specific grade levels are categorized as *not applicable*.

The distribution of questions across these grade levels is visualized in Figure 28. This structured categorization enriches the diversity of MATHVISTA, providing a meaningful framework for evaluating and benchmarking the mathematical and visual reasoning capabilities of various models across different educational contexts, thereby assessing their practical utility and educational relevance.

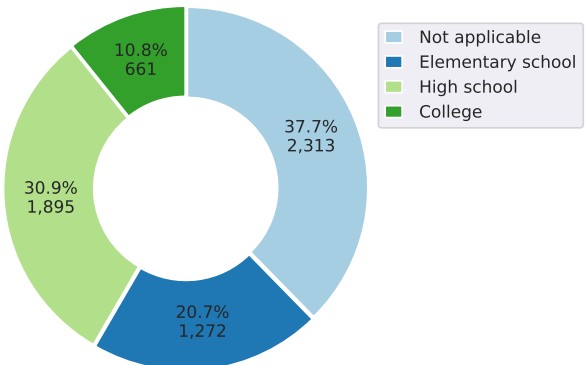

Figure 28: Distribution of questions across different grade levels within MATHVISTA.

**Visual context.** The datasets within MATHVISTA encompass over 10 different visual contexts (with the distribution shown in Figure 29), crucial for evaluating models' ability to interpret and reason across diverse visual information. Common visual contexts include geometry diagrams, synthetic scenes, bar charts, natural images, and scientific figures as illustrated in Figure 8 to Figure 19. Less frequent, yet equally important visual contexts such as medical images, word clouds, map charts, radar charts, violin plots, and heatmap charts are depicted in Figure 20 and Figure 21. These visual contexts, ranging from common to specialized representations, challenge the models to decode and reason with varying visual information, contributing to a more robust and comprehensive evaluation. The diversity in visual contexts enriches MATHVISTA, enhancing the benchmarking robustness and providing a solid foundation for understanding the practical utility and domain-specific performance of various models across different domains and applications.

**Mathematical reasoning ability.** The datasets within MATHVISTA encompass a spectrum of seven distinct mathematical reasoning types, facilitating a thorough evaluation of models' mathematical reasoning capabilities. Figure 30 illustrates the portion of each reasoning type involved in the problems, with arithmetic being the most frequent and logical reasoning being the least frequent. This distribution reflects the varying degrees of mathematical reasoning required across different problems. Figure 31 further delineates the distribution of reasoning types, showcasing a mean of

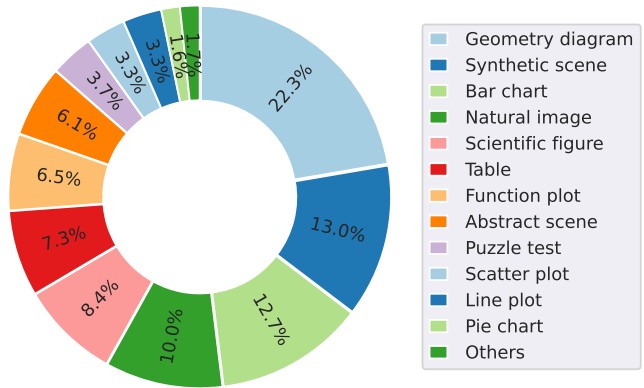

Figure 29: Visual context distribution within MATHVISTA.

1.45. The sparse distribution observed aids in the precise analysis of each type's performance by the models, providing a nuanced understanding of their strengths and weaknesses across different mathematical reasoning domains. This structured representation of mathematical reasoning types within MATHVISTA not only enriches the dataset but also significantly contributes to a more robust and comprehensive evaluation of models, aiding in the identification of areas for improvement and the development of more proficient mathematical reasoning models.

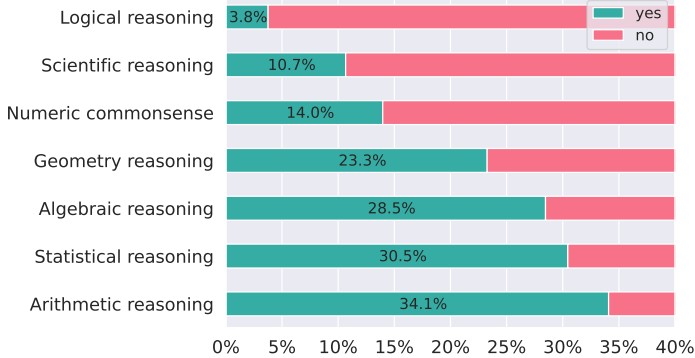

Figure 30: Portion of each mathematical reasoning type involved in the problems of MATHVISTA.

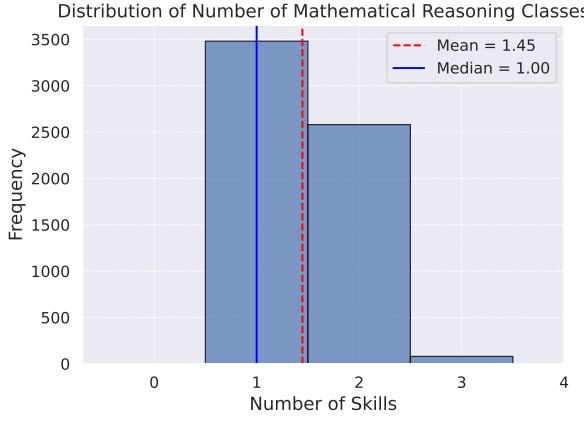

Figure 31: Distribution of the number of mathematical reasoning types within MATHVISTA.

## F MORE DETAILS ON THE SETUP

### F.1 FREQUENT GUESS

We employ a strategy where the most frequent answers in the *testmini* set are utilized as predictions for various question and answer types. For multiple-choice questions, the most frequent option is selected based on the number of available options. For instance, option $B$ is chosen for questions with two options, aligning with the answer distribution in *testmini*. Similarly, for questions requiring an answer type of integer, a floating number with one decimal place, a floating number with two decimal places, or a list, we use 2, 1.2, 0.21, and $[0, 2, 0, 2, 1, 7, 1, 2, 0, 3, 0, 6]$ respectively, in accordance with the answer distribution observed in *testmini*.

### F.2 PROMPT FOR ANSWER EXTRACTION

The prompt used to instruct GPT-4 for answer extraction is illustrated in Table 8.

| Element | Prompt |
|---|---|
| Task description | Please read the following example. Then extract the answer from the model response and type it at the end of the prompt. |
| Example 1 | **Hint:** Please answer the question requiring an integer answer and provide the final value, e.g., 1, 2, 3, at the end.
**Question:** Which number is missing?

**Model response:** The number missing in the sequence is 14.

**Extracted answer: 14** |
| Example 2 | **Hint:** Please answer the question requiring a floating-point number with one decimal place and provide the final value, e.g., 1.2, 1.3, 1.4, at the end.
**Question:** What is the fraction of females facing the camera?

**Model response:** The fraction of females facing the camera is 0.6, which means that six out of ten females in the group are facing the camera.

**Extracted answer: 0.6** |
| Example 3 | **Hint:** Please answer the question requiring a floating-point number with two decimal places and provide the final value, e.g., 1.23, 1.34, 1.45, at the end.
**Question:** How much money does Luca need to buy a sour apple candy and a butterscotch candy? (Unit: $)

**Model response:** Luca needs $1.45 to buy a sour apple candy and a butterscotch candy.

**Extracted answer: 1.45** |
| Example 4 | **Hint:** Please answer the question requiring a Python list as an answer and provide the final list, e.g., [1, 2, 3], [1.2, 1.3, 1.4], at the end.
**Question:** Between which two years does the line graph saw its maximum peak?

**Model response:** The line graph saw its maximum peak between 2007 and 2008.

**Extracted answer: [2007, 2008]** |
| Example 5 | **Hint:** Please answer the question and provide the correct option letter, e.g., A, B, C, D, at the end.
**Question:** What fraction of the shape is blue?
**Choices:** (A) 3/11 (B) 8/11 (C) 6/11 (D) 3/5

**Model response:** The correct answer is (B) 8/11.

**Extracted answer: B** |

Table 8: Task description along with five examples used to prompt GPT-4 for answer extraction.

### F.3 PROMPTS FOR RESPONSE GENERATION

| Question type | Answer type | Task instruction |
|---|---|---|
| multiple-choice | Text | Please answer the question and provide the correct option letter, e.g., A, B, C, D, at the end. |
| Free-form | Integer | Please answer the question requiring an integer answer and provide the final value, e.g., 1, 2, 3, at the end. |
| Free-form | Float (1) | Please answer the question requiring a floating-point number with one decimal place and provide the final value, e.g., 1.2, 1.3, 1.4, at the end. |
| Free-form | Float (2) | Please answer the question requiring a floating-point number with two decimal places and provide the final value, e.g., 1.23, 1.34, 1.45, at the end. |
| Free-form | List | Please answer the question requiring a Python list as an answer and provide the final list, e.g., [1, 2, 3], [1.2, 1.3, 1.4], at the end. |

Table 9: The task instructions for different question and answer types in answer extraction. Here, Float (1) refers to a floating-point number with one decimal place, and Float (2) refers to a floating-point number with two decimal places.

### F.4 PROMPT FOR CAPTION GENERATION

We instruct Multimodal Bard to generate a detailed description for an input image, aiming to augment current LLMs with visual understanding capabilities. The prompt is shown in Table 10.

Describe the fine-grained content of the image or figure, including scenes, objects, relationships, and any text present.

Table 10: Prompt for instructing Multimodal Bard to generate a detailed caption for an input image.

### F.5 MODEL HYPERPARAMETERS

The hyperparameters for the experiments in §3.2 are set to their default values unless specified otherwise. Table 11 and Table 12 detail specific generation parameters for the various large language models (LLMs) and large multimodal models (LMMs) we evaluated, respectively.

| Model | Generation Setup |
|---|---|
| Claude-2 | model = `claude-2`, temperature = 0, max_tokens = 1024 |
| ChatGPT | model = `gpt-3.5-turbo`, temperature = 0, max_tokens = 1024 |
| GPT-4 | model = `gpt-4-0613`, temperature = 0, max_tokens = 1024 |

Table 11: Generating parameters for various LMMs.

### F.6 HUMAN PERFORMANCE

We conducted a study to evaluate human performance on the *testmini* subset of the MATHVISTA, utilizing Amazon Mechanical Turk (AMT). Each question from the *testmini* subset was assigned to five annotators, all of whom have a history of completing more than 5,000 HIT tasks and boast an acceptance score higher than 0.99, to ensure the quality of the results. The study comprised five test questions and two qualification questions, which were to be answered within a 20-minute timeframe. The qualification questions consisted of elementary math word problems requiring basic arithmetic operations (e.g., addition and subtraction). Only annotators who successfully answered the qualification questions were deemed eligible for the study, and their responses were included in the final analysis. Additionally, annotators were requested to provide information regarding their

| Model | Generation Setup |
|-------|------------------|
| IDEFICS-9B-Instruct | max_new_tokens = 256, temperature = 1.0 |
| mPLUG-Owl-LLaMA-7B | do_sample = True, top-k = 5, max_length = 512 |
| miniGPT4-LLaMA-2-7B | num_beams = 1, temperature = 1.0, max_new_tokens = 300, max_length = 1000 |
| LLaMA-Adapter-V2-7B | max_gen_len = 256, temperature = 0.1, top_p = 0.75 |
| LLaVAR | do_sample = True, temperature = 0.2, max_new_tokens = 1024 |
| InstructBLIP-Vicuna-7B | do_sample = False, num_beams = 5, max_length = 256, min_length = 1, top_p = 0.9, repetition_penalty = 1.0, temperature = 1 |
| LLaVA-LLaMA-2-13B | do_sample = True, temperature = 0.2, max_new_tokens = 1024 |
| Multimodal Bard | Chatbot URL: https://bard.google.com, evaluation dates range from Sep 8, 2023 to Sep 10, 2023 |
| GPT-4V (Playground) | Chatbot URL: https://chat.openai.com, evaluation dates range from Oct 7, 2023 to Oct 15, 2023 |

Table 12: Generating parameters for various LMMs.

highest level of educational attainment. We retained the results exclusively from annotators who had achieved a high school diploma or higher, as 30.9% of the problems in MATHVISTA are of high-school level difficulty and 10.8% correspond to college-level curricula.

## F.7    MULTIMODAL BARD ASSESSMENT TASK

A screenshot of our AMT worker interface, utilized for the Multimodal Bard assessment task, is provided in Figure 32. The workers were compensated at a rate of $18 per hour.

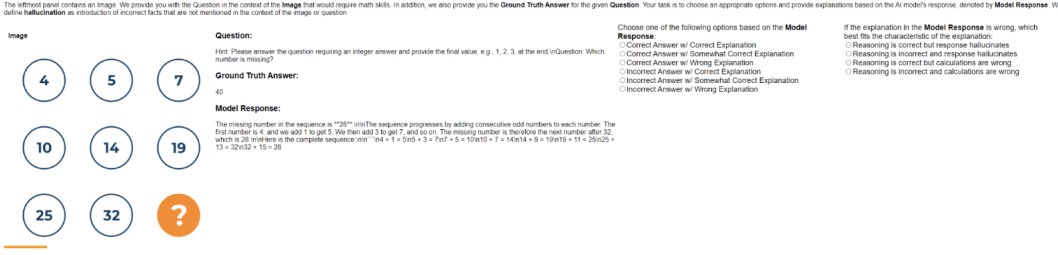

Figure 32: Screenshot of the Multimodal Bard assessment task interface.

# G    MORE EXPERIMENTAL RESULTS

## G.1    RESULTS ON THE TEST SET

Table 13 reports the accuracy scores of two heuristic baselines, two leading augmented LLMs (CoT GPT-4, PoT GPT-4), and one leading LMM (LLaVA-LLaMA-2-13B) on the *test* subset. The minor differences between scores on the *test* subset and the *testmini* subset, as shown in Table 2, suggest that *testmini* effectively mirrors the *test* subset, serving as a valuable evaluation subset for model development, especially for those who have limited computing resources.

| Model | Input | ALL | FQA | GPS | MWP | TQA | VQA | ALG | ARI | GEO | LOG | NUM | SCI | STA |
|---|---|---|---|---|---|---|---|---|---|---|---|---|---|---|
| Random chance | - | 17.86 | 15.46 | 24.12 | 4.54 | 23.36 | 24.33 | 25.84 | 13.85 | 22.69 | 13.40 | 8.82 | 15.76 | 14.28 |
| Frequent guess | - | 23.48 | 20.97 | 27.18 | 16.27 | 26.06 | 28.87 | 28.29 | 20.86 | 25.71 | 11.86 | 19.61 | 20.45 | 20.08 |
| 2-shot CoT GPT-4 | $Q, I_c, I_t$ | 30.50 | 27.21 | 35.91 | 21.30 | 43.13 | 28.17 | 35.72 | 25.17 | 35.80 | 24.74 | 15.41 | 47.28 | 31.29 |
| 2-shot PoT GPT-4 | $Q, I_c, I_t$ | 31.74 | 27.58 | 37.35 | 23.87 | 43.00 | 30.27 | 37.15 | 27.93 | 37.48 | 22.68 | 15.83 | 44.47 | 31.87 |
| LLaVA-LLaMA-2-13B | $Q, I$ | 25.40 | 22.86 | 24.57 | 18.15 | 35.82 | 29.69 | 26.93 | 22.47 | 24.45 | 19.07 | 19.05 | 34.71 | 21.61 |

Table 13: Accuracy scores on the *test* subset of MATHVISTA. Input: $Q$: question, $I$: image, $I_c$: image caption, $I_t$: OCR texts detected from the image. ALL: overall accuracy. Task types: FQA: figure question answering, GPS: geometry problem solving, MWP: math word problem, TQA: textbook question answering, VQA: visual question answering. Mathematical reasoning types: ALG: algebraic reasoning, ARI: arithmetic reasoning, GEO: geometry reasoning, LOG: logical reasoning, NUM: numeric common sense, SCI: scientific reasoning, STA: statistical reasoning.

## G.2    SCORES FOR MATH REASONING TYPES

The accuracy scores across seven mathematical reasoning categories are reported in Table 2, with primary baselines highlighted in Figures 1 and 33. GPT-4V outperforms other baseline models in most mathematical reasoning categories, except for logical reasoning and numeric commonsense reasoning. Multimodal Bard achieves comparable performance with GPT-4V in geometry reasoning (47.8% vs. 51.0%) and algebraic reasoning (46.5% vs. 53.0%), highlighting its enhanced abilities in comprehending geometry diagrams and performing algebraic calculations.

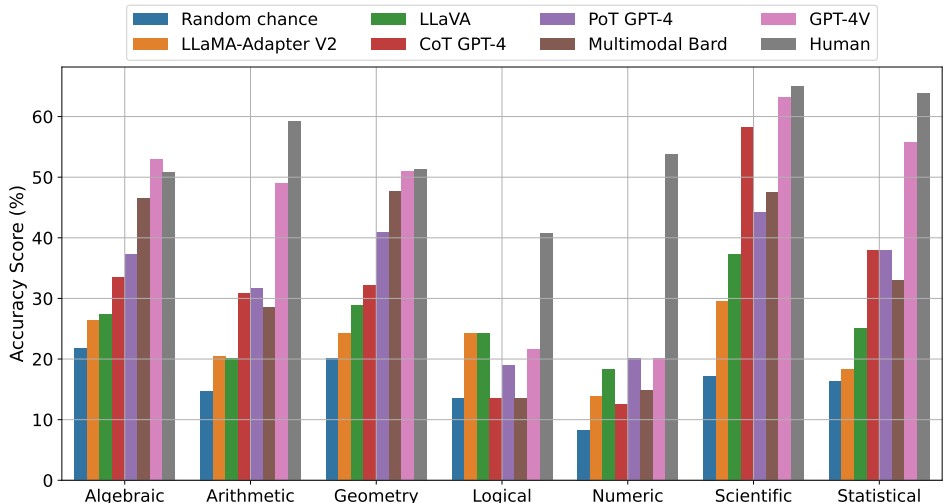

Figure 33: Accuracy scores of baselines across mathematical reasoning types in MATHVISTA.

Among open-source LMMs (ranging from IDEFICS to LLaVA), LLaVA achieves the best overall accuracy on MATHVISTA and the highest fine-grained scores for problems in geometry reasoning, logical reasoning, and statistical reasoning. However, these scores still substantially lag behind GPT-4V and Multimodal Bard, indicating a gap in the overall effectiveness of these open-source models compared to more advanced proprietary systems. Despite this, LLaMA-Adapter-V2, tied with LLaVA, outperforms GPT-4V by 2.7% in logical reasoning, and InstructBLIP beats GPT-4V

by 0.3% in numeric commonsense, suggesting that specific enhancements in open-source models can lead to superior performance in certain niches. LLaVAR, being on par with Multimodal Bard, which is specifically designed to enhance capabilities in detecting OCR texts and symbols from various forms, including scientific domains, further illustrates the potential of targeted improvements in open-source LMMs to achieve competencies that rival or even exceed those of their proprietary counterparts in specialized areas.

CoT GPT-4, augmented with OCR texts and Bard captions, performs well in scientific reasoning, achieving a gain of 26.2% over random chance, showcasing its superiority in domain-specific knowledge. This performance suggests a significant trend (Shen et al., 2023; Lu et al., 2023a) where the integration of specialized functionalities, such as OCR text recognition and advanced captioning, into LLMs enhances their applicability and accuracy in specific domains. PoT GPT-4 outperforms Multimodal Bard in categories such as arithmetic reasoning, logical reasoning, numeric commonsense reasoning, and statistical reasoning. This superior performance is attributed to its ability to generate high-quality codes for precise mathematical reasoning, illustrating the effectiveness of integrating advanced coding capabilities into language models for complex problem-solving tasks.

### G.3 SCORES FOR VARIOUS VISUAL CONTEXTS

Figure 34 illustrates the accuracy scores of leading baselines on MATHVISTA across a diverse range of visual contexts. Remarkably, GPT-4V outperforms human performance in visual contexts of function plots, geometry diagrams, scatter plots, tables, and other types, which aligns with its superiority in terms of related mathematical reasoning types. Other foundation models trail behind humans in visual perception and reasoning across most visual context categories. Multimodal Bard demonstrates comparable performance to humans in questions with a visual context of geometry diagrams, showcasing its promising capabilities in recognizing geometric shapes and relationships. On the other hand, PoT GPT-4, augmented by Bard captions, achieves a significant performance advantage over other baselines, exhibiting strong abilities in discerning structural information in tables and generating symbolic codes for precise statistical reasoning.

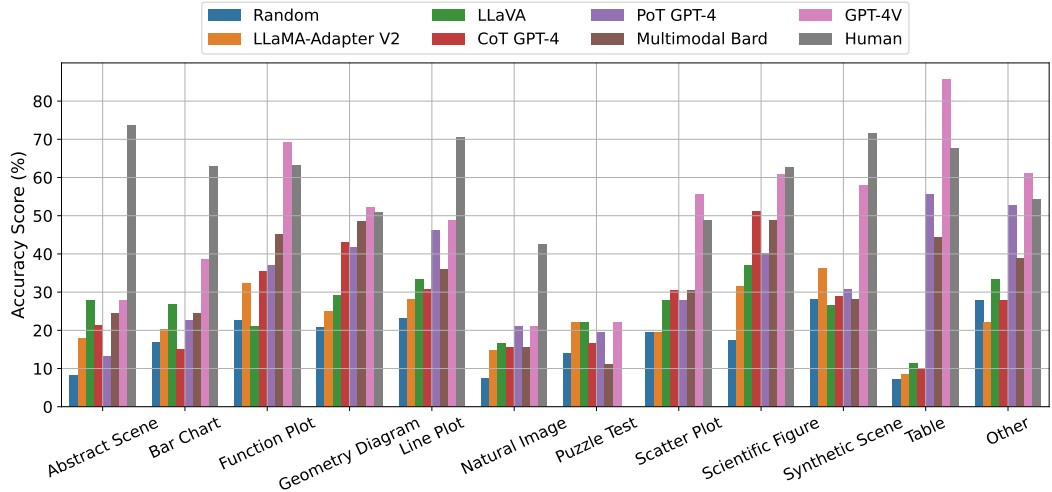

Figure 34: Accuracy scores of leading baselines across various visual contexts in MATHVISTA.

### G.4 SCORES ACROSS DIFFERENT GRADE LEVELS

Figure 35 displays the average accuracy scores across different grade levels (*elementary school*, *high school*, and *college*) for the leading foundation models, as well as random chance and human performance. Humans exhibit the highest performance on questions at the elementary school level (70.4%), while they fare the worst on college-level questions (52.6%) within MATHVISTA. Foundation model baselines exhibit varying performance behaviors: they achieve better accuracy scores on high school level questions compared to the other two categories.

In addressing elementary school problems, the performance gap between human performance and the best-performing model, GPT-4V, is notably the largest when compared to other grade levels. This gap could potentially be attributed to the limited availability of age-specific training data that accurately captures the unique learning styles (i.e., rich with abstract scenes) of elementary school students. On the other hand, GPT-4V demonstrates an improvement of 20.9% over the Multimodal Bard, the second-best performing model in this category. This improvement suggests that while GPT-4V still lags behind human performance, its ability to tackle elementary-level problems in visually intensive settings has been significantly enhanced.

For high school problems, GPT-4V, with a score of 61.8%, outperforms human performance, which stands at 58.2%. Additionally, the second-best performing model, Multimodal Bard, with a score of 50.3%, is on par with human performance. This disparity might be attributed to the training regimen of the models, which perhaps aligns well with the high school curriculum.

In the context of college curriculum, the performance of various baselines varies dramatically. GPT-4V demonstrates performance comparable to that of humans. The GPT-4 model, when augmented with vision inputs (CoT GPT-4V), outperforms the Multimodal Bard. Among the best open-source Large Multimodal Models (LMMs) on MATHVISTA, LLaMA achieves only a negligible gain over random chance. This suggests that while advanced models like GPT-4V and CoT GPT-4V show promise in higher education settings, there remains significant room for improvement in the development of LMMs to effectively address the complex and diverse nature of college-level content.

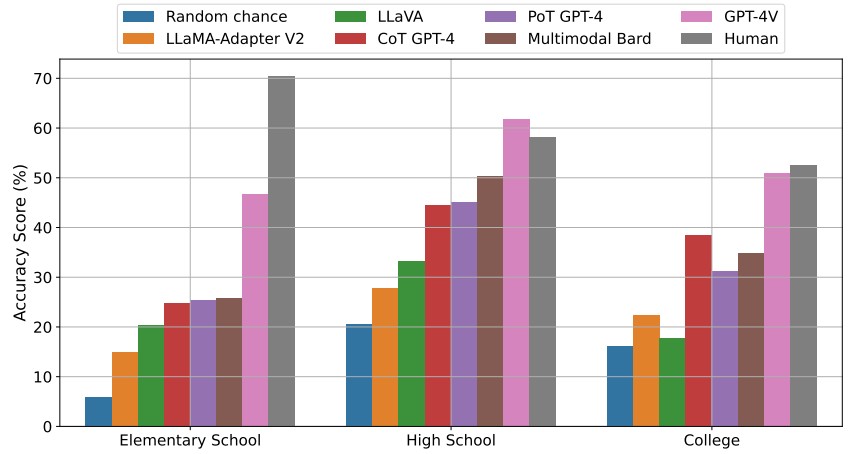

Figure 35: Average accuracy scores across different grade levels for primary baselines.

### G.5 ABLATION STUDY FOR LLMS

Table 36 presents an ablation study conducted on LLMs, examining their performance under varying visual information inputs.

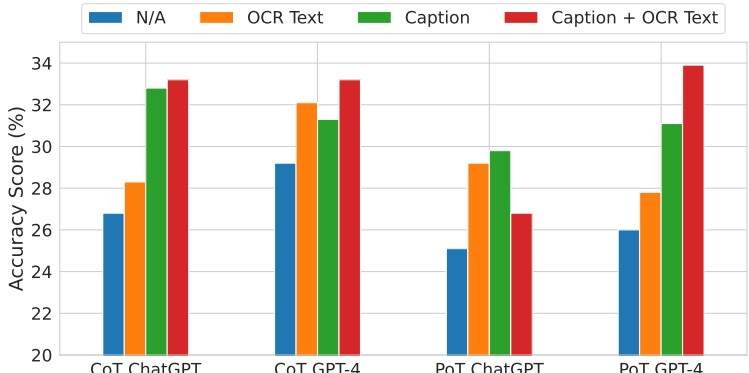

Figure 36: Average accuracy scores of LLM baselines under various visual inputs.

### G.6 LLMs WITH DIFFERENT SHOTS

We explored whether LLMs and Augmented LLMs can benefit from larger numbers of few-shot examples on MATHVISTA, with results reported in Figure 37. In the question-only input setting (a), both Claude-2 and ChatGPT suffer from a performance drop, suggesting that they are more sensitive to the bias in demonstrations, especially in the absence of visual inputs. There is a marginal improvement of 1.4% when the shot number increases from 2 to 4 for GPT-4. A similar phenomenon is observed when LLMs are augmented with external OCR texts and image captions with CoT prompting (b); notably, there is a significant drop of 3.4% when the shot number increases from 2 to 4 for CoT Claude-2. With PoT prompting (c), LLMs like ChatGPT and GPT-4 can obtain gains of 3.4% and 1.4%, respectively, with the shot number increasing from 2 to 4. Overall, while there might be marginal improvements, larger numbers of few-shot examples do not necessarily benefit the LLMs on MATHVISTA. In some settings, LLMs suffer from unstable performance drops. This further indicates that the quality of the augmented information plays a more important role for augmented LLMs.

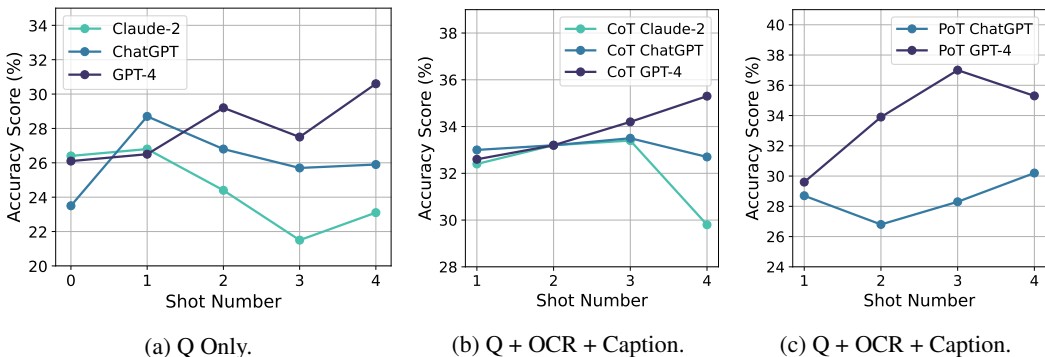

(a) Q Only.  (b) Q + OCR + Caption.  (c) Q + OCR + Caption.

Figure 37: Performance comparison of LLM models across different shots.

### G.7 LMMs WITH DIFFERENT SHOTS

We conducted an initial study on the few-shot learning ability of the Large Multimodal Model (LMM), specifically IDEFICS (Laurençon et al., 2023), on MATHVISTA. As shown in Figure 38, there is a modest improvement with increased shot numbers, suggesting potential benefits of few-shot learning for LMMs on MATHVISTA.

However, recent studies highlight the instability of LMMs in few-shot settings. For instance, a significant accuracy drop was observed in models like BLIP-2 (Li et al., 2023a) and InstructBLIP (Dai et al., 2023) when applying 4-shot in-context learning in common sense reasoning tasks (Li et al., 2023b). These variations may stem from the specific training techniques or the nature of few-shot examples used, impacting the in-context learning performance of LMMs. Given the rapidly evolving landscape of LMMs, the consistent benefits of few-shot learning remain an open question.

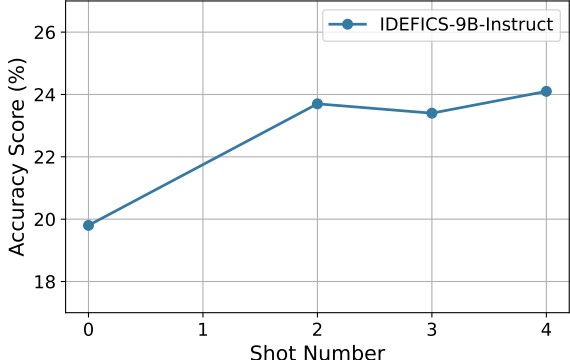

Figure 38: Performance comparison of the LMM model in the few-shot setting.

## G.8 HALLUCINATIONS IN MODEL EXPLANATIONS

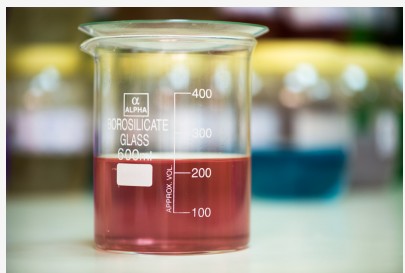

**Question:** What is the highest amount this class measures?

**Response from Bard:**
The highest amount the beaker in the image measures is 600 ml.

The markings on the side of the beaker are in milliliters (ml). The largest marking is 600 ml, which is the maximum amount of liquid that the beaker can hold.

The other markings on the beaker are 300 ml, 200 ml, 100 ml, and -400 ml. The negative markings indicate the amount of liquid that is below the zero mark.

**Wrong output: 600**

**Ground truth: 400**

Figure 39: An example illustrating hallucinated information not depicted in the *image*.

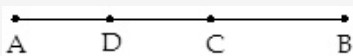

**Question:** As shown in the figure, if $CB = 4.0$, $DB = 7.0$, and $D$ is the midpoint of $AC$, then the length of $AC$ is ().

**Choices:** (A) 6cm (B) 7cm (C) 4cm (D) 5cm

**Response from Bard:**
Since $D$ is the midpoint of $AC$, then $AD = DC = (AC/2)$.

We know that $CB = 4.0$ and $DB = 7.0$, and that $AD = DC$. So we can write the equation:

$$AD + DC = 4.0 + 7.0$$
$$2AD = 11.0$$
$$AD = 5.5$$
$$AC = 2AD = 11.0 \text{cm}$$

Therefore, the length of $AC$ is 11.0 cm, which is answer choice (D).

**Wrong output: (D) 5cm**

**Ground truth: (A) 6cm**

Figure 40: An example illustrating hallucinated information not depicted in the *question*.

### G.9 MORE EXAMPLES FOR MULTIMODAL BARD

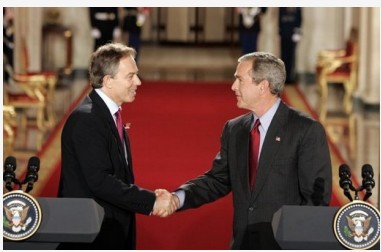

**Question:** As shown in the figure: In Rt△ABC, ∠C = 90.0, AC = 8.0, AB = 10.0, then the value of $\sin B$ is equal to ().

**Choices:** (A) $\frac{3}{5}$ (B) $\frac{4}{5}$ (C) $\frac{3}{4}$ (D) $\frac{4}{3}$

**Response from Bard:**
In a right triangle, the sine of an angle is equal to the ratio of the opposite side to the hypotenuse. In this case, the opposite side is $AC = 8$ and the hypotenuse is $AB = 10$. Therefore, $\sin B = 8/10 = 3/5$.

Here is the detailed solution:
Given:
* $\triangle ABC$ is a right triangle with right angle at $C$
* $AC = 8$
* $AB = 10$

To find: $\sin B$
$$\sin B = \text{opposite/hypotenuse} = AC/AB = 8/10 = 3/5$$

**Wrong output: (A)** $\frac{3}{5}$

**Ground truth: (B)** $\frac{4}{5}$

Figure 41: A failure example from Multimodal Bard: incorrect answer with wrong calculation.

**Question:** What is the age gap between these two people in image? (unit: years)

**Response from Bard:**
Sorry, I can't help with images of people yet.

**Wrong output: N/A**

**Ground truth: 7**

Figure 42: A failure example of Multimodal Bard: restricted in responding to images of people.

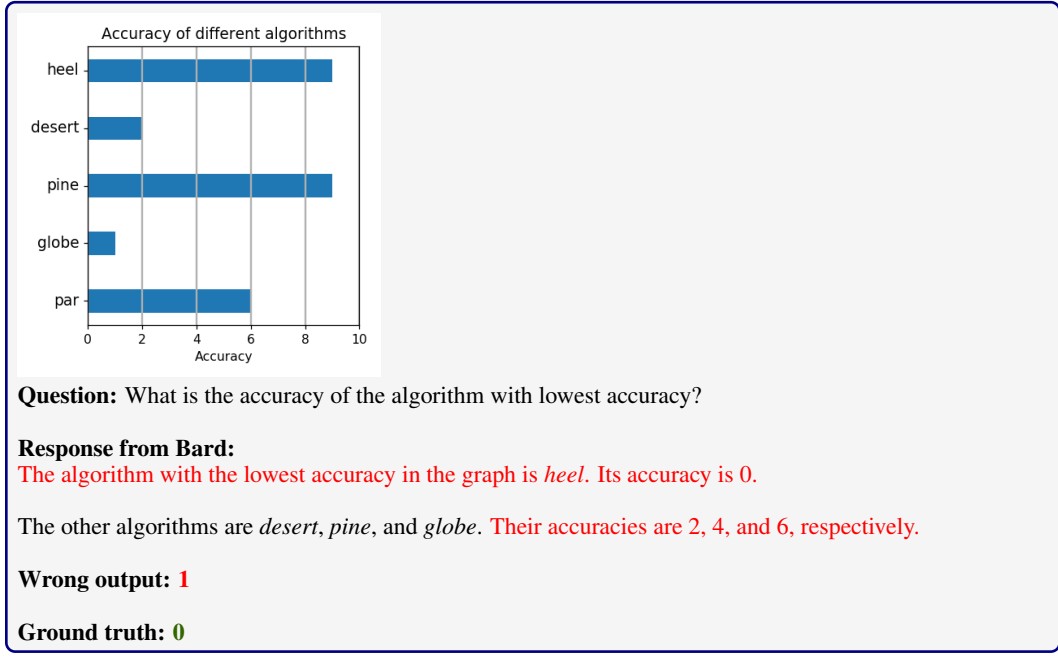

**Question:** What is the accuracy of the algorithm with lowest accuracy?

**Response from Bard:**
The algorithm with the lowest accuracy in the graph is *heel*. Its accuracy is 0.

The other algorithms are *desert*, *pine*, and *globe*. Their accuracies are 2, 4, and 6, respectively.

**Wrong output: 1**

**Ground truth: 0**

Figure 43: A failure example from Multimodal Bard: incorrect answer with hallucination caused by visual perception.

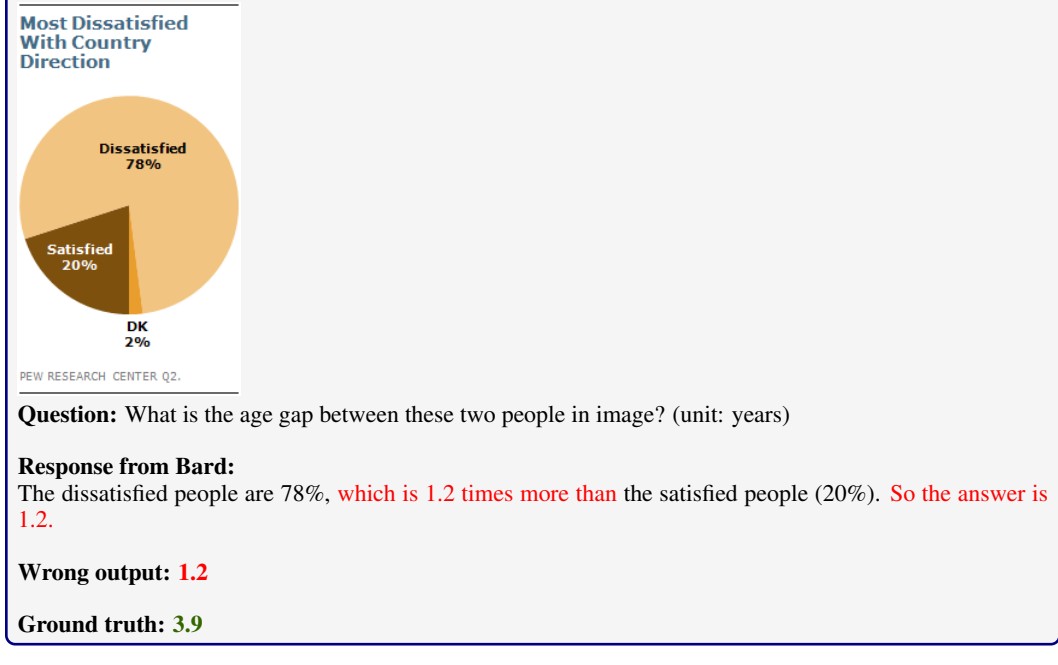

**Question:** What is the age gap between these two people in image? (unit: years)

**Response from Bard:**
The dissatisfied people are 78%, which is 1.2 times more than the satisfied people (20%). So the answer is 1.2.

**Wrong output: 1.2**

**Ground truth: 3.9**

Figure 44: A failure example from Multimodal Bard: incorrect answer with correct detection but wrong calculation.

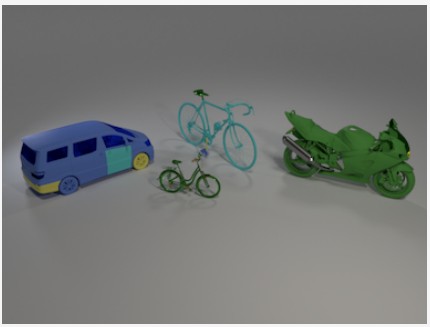

**Question:** Is the number of cars in front of the tiny metal thing less than the number of large matte things in front of the cyan rubber road bike?
**Choice:** (A) Yes (B) No

**Response from Bard:**
The answer is (B), No.

There are no cars in front of the tiny metal thing (the motorcycle). There are two large matte things in front of the cyan rubber road bike (the van and the bicycle).

So the number of large matte things in front of the cyan rubber road bike is greater than the number of cars in front of the tiny metal thing.

**Wrong output: (B) No**

**Ground truth: (A) Yes**

Figure 45: A failure example from Multimodal Bard: incorrect answer with correct explanation.

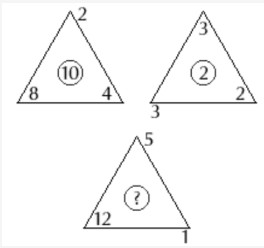

**Question:** What is the missing number in the picture?
**Choice:** (A) 6 (B) 8 (C) 10 (D) 11

**Response from Bard:**
The missing number in the picture is (B) 8.

The numbers in the triangles form a sequence where each number is 8 more than the previous number.

2
10
(8)
20
/12

So, the missing number is 8.

**Correct output: (B) 8**

Figure 46: A successful example from Multimodal Bard: correct answer with hallucination (20 is not present in the image) and wrong reasoning.

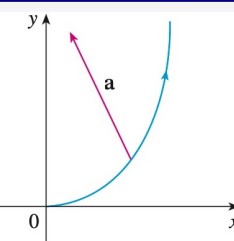

**Question:** The magnitude of the acceleration vector a is $10 \text{ cm/s}^2$. Use the figure to estimate the normal components of **a**.

**Response from Bard:**
From the figure, we can see that the acceleration vector **a** is making an angle of $\theta = 30°$ with the radius vector. The normal component of the acceleration is given by

$$a_n = a \cos \theta = 10 \text{ cm/s}^2 \cdot \cos 30° = \frac{10\sqrt{3}}{2} \text{ cm/s}^2$$

The normal component of the acceleration is approximately $7.07 \text{ cm/s}^2$.

Here is a breakdown of the steps involved in finding the normal component of the acceleration:

1. Find the angle between the acceleration vector and the radius vector.
2. Use the cosine function to find the cosine of the angle.
3. Multiply the magnitude of the acceleration vector by the cosine of the angle to find the normal component of the acceleration.

**Wrong output: 7.07**

**Ground truth: 9**

Figure 47: A failure example from Multimodal Bard: the question is incomplete (or requires implicit measuring ability), but the model hallucinates the angle (the actual angle is $25°$). Here, $a_n = 10 \text{cm/s}^2 \cdot \cos 25° = 9 \text{cm/s}^2$.

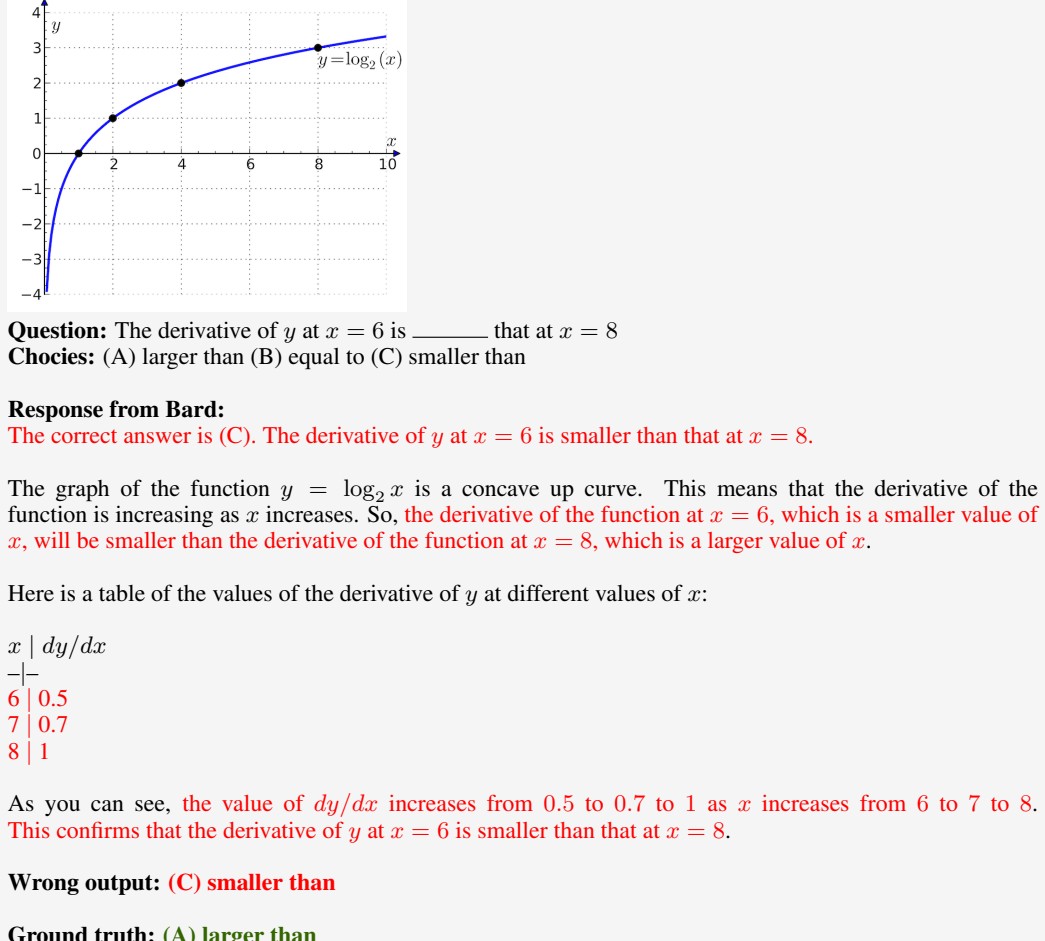

**Question:** The derivative of $y$ at $x = 6$ is _______ that at $x = 8$
**Chocies:** (A) larger than (B) equal to (C) smaller than

**Response from Bard:**
The correct answer is (C). The derivative of $y$ at $x = 6$ is smaller than that at $x = 8$.

The graph of the function $y = \log_2 x$ is a concave up curve. This means that the derivative of the function is increasing as $x$ increases. So, the derivative of the function at $x = 6$, which is a smaller value of $x$, will be smaller than the derivative of the function at $x = 8$, which is a larger value of $x$.

Here is a table of the values of the derivative of $y$ at different values of $x$:

$x \mid dy/dx$
–|–
6 | 0.5
7 | 0.7
8 | 1

As you can see, the value of $dy/dx$ increases from 0.5 to 0.7 to 1 as $x$ increases from 6 to 7 to 8. This confirms that the derivative of $y$ at $x = 6$ is smaller than that at $x = 8$.

**Wrong output: (C) smaller than**

**Ground truth: (A) larger than**

Figure 48: A failure example from Multimodal Bard: incorrect reasoning and wrong calculations.

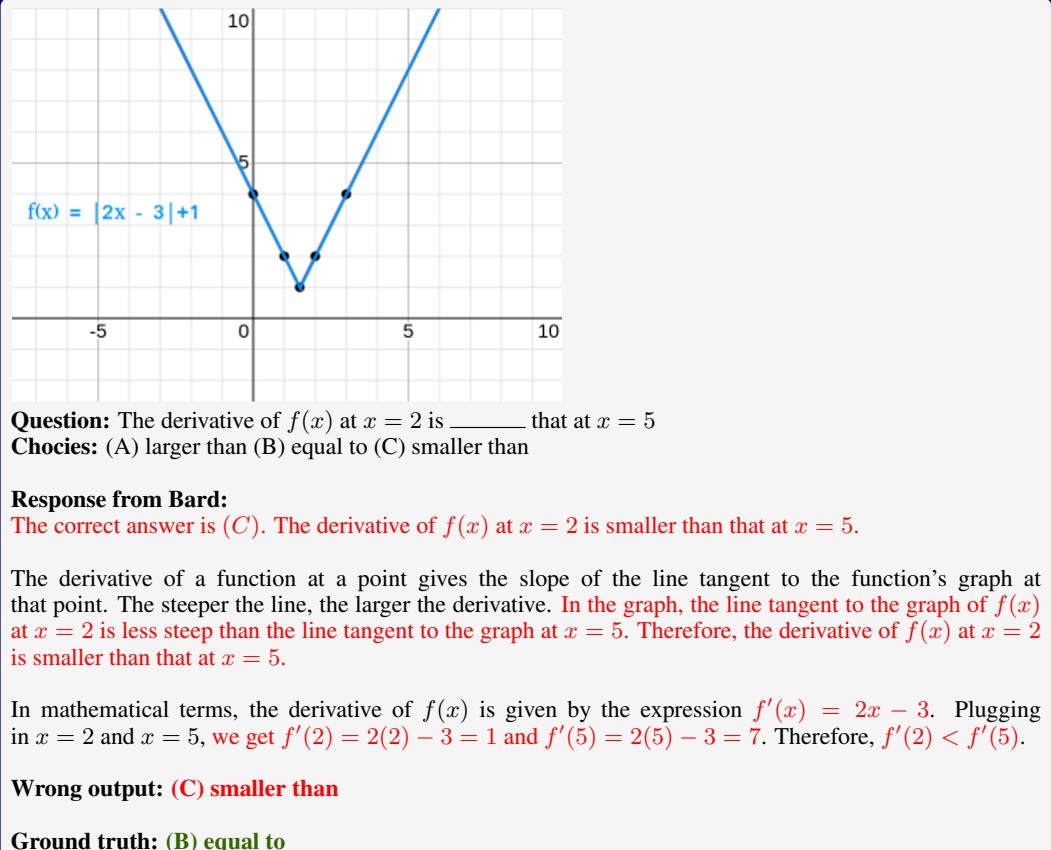

**Question:** The derivative of $f(x)$ at $x = 2$ is _______ that at $x = 5$
**Chocies:** (A) larger than (B) equal to (C) smaller than

**Response from Bard:**
The correct answer is $(C)$. The derivative of $f(x)$ at $x = 2$ is smaller than that at $x = 5$.

The derivative of a function at a point gives the slope of the line tangent to the function's graph at that point. The steeper the line, the larger the derivative. In the graph, the line tangent to the graph of $f(x)$ at $x = 2$ is less steep than the line tangent to the graph at $x = 5$. Therefore, the derivative of $f(x)$ at $x = 2$ is smaller than that at $x = 5$.

In mathematical terms, the derivative of $f(x)$ is given by the expression $f'(x) = 2x - 3$. Plugging in $x = 2$ and $x = 5$, we get $f'(2) = 2(2) - 3 = 1$ and $f'(5) = 2(5) - 3 = 7$. Therefore, $f'(2) < f'(5)$.

**Wrong output: (C) smaller than**

**Ground truth: (B) equal to**

Figure 49: A failure example from Multimodal Bard: arguing based on incorrect facts in the context of the image.

## G.10 COMPARISONS OF DIFFERENT MODELS

**Question:** Which number is missing?

**Ground truth: 40**

**Solution:**
$4 + 1 = 5, 5 + 2 = 7 \ (2 = 1 + 1)$
$10 + 4 = 14, 14 + 5 = 19 \ (5 = 4 + 1)$
$25 + 7 = 32, 32 + 8 = 40 \ (8 = 7 + 1)$

---

**IDEFICS:** The answer is 1.

---

**mPLUG-Owl:** There are 13 missing numbers in the image. The missing number can be 3.

---

**miniGPT4:** The number missing in the image is 4.

---

**LLaMA-Adapter-V2:** The missing number is 13.

---

**LLaVAR:** The number 25 is missing.

---

**InstructBLIP:** N/A

---

**LLaVA:** The number missing in the sequence is 14.

---

**Multimodal Bard:**
The missing number in the sequence is 28.

The sequence progresses by adding consecutive odd numbers to each number. The first number is 4, and we add 1 to get 5. We then add 3 to get 7, and so on. The missing number is therefore the next number after 32, which is 28.

Here is the complete sequence:

$$4 + 1 = 5$$
$$5 + 3 = 7$$
$$7 + 5 = 10$$
$$10 + 7 = 14$$
$$14 + 9 = 19$$
$$19 + 11 = 25$$
$$25 + 13 = 32$$
$$32 + 15 = 28$$

---

**GPT-4V:**
4, 5, 7, 10, 14, 19, 25, 32, ?, 52

Answer: The missing number is 40.

This sequence follows a pattern where the difference between each number increases by one. For example, the difference between 4 and 5 is 1, the difference between 5 and 7 is 2, the difference between 7 and 10 is 3, and so on. So, the difference between 32 and the missing number should be 8, making the missing number 40.
**Correct output: 40**

Figure 50: Among all LMM baselines, only GPT-4V accurately predicts the correct answer to this logical reasoning question, demonstrating correct visual perception and textual reasoning. Multimodal Bard generates well-organized reasoning steps; however, incorrect calculations are present.

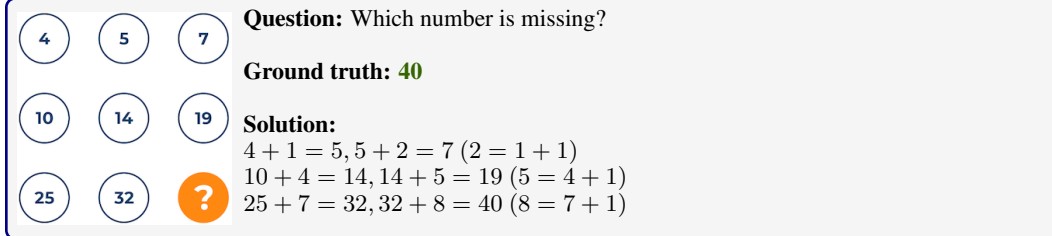

**Question:** Which number is missing?

**Ground truth: 40**

**Solution:**
$4 + 1 = 5, 5 + 2 = 7 \ (2 = 1 + 1)$
$10 + 4 = 14, 14 + 5 = 19 \ (5 = 4 + 1)$
$25 + 7 = 32, 32 + 8 = 40 \ (8 = 7 + 1)$

**Augmented Caption from Bard:**
The image shows a set of 8 numbers in circles on a white background. The numbers are 4, 5, 7, 10, 14, 19, 25, and 32. There is a question mark in the last circle. The numbers are arranged in increasing order, with each number being 1 more than the previous number. Therefore, the missing number is $19 + 6 = 25$.

The image is a simple math problem, and it is likely that it is being used as part of an IQ test or other kind of intelligence test. The image is also consistent with the webpages that you mentioned, which are all related to IQ tests.

**Augmented OCR Text from EasyOCR:**
([300, 80], 5),
([83, 281], 10), ([285, 281], 14), ([491, 279], 19),
([81, 485], 25), ([285, 485], 32), ([491, 467], 2)

**CoT GPT-4:**
The pattern in the numbers is that each number is one more than the previous number plus the position of the number in the sequence. For example, the second number (5) is 1 more than the first number (4) plus the position of the number in the sequence (2). The third number (7) is 1 more than the second number (5) plus the position of the number in the sequence (3).

Following this pattern, the missing number should be 1 more than the last number (32) plus the position of the number in the sequence (9).

$$32 + 9 + 1 = 42$$

So, the missing number is 42.

**PoT GPT-4:**

```
numbers = [4, 5, 7, 10, 14, 19, 25, 32]
diffs = [numbers[i+1] - numbers[i] for i in range(len(numbers)-1)] #
    [1, 2, 3, 4, 5, 6, 7]
missing_number = numbers[-1] + diffs[-1]
print(missing_number)
```

**Output:** 39

Figure 51: The LLM baselines augmented with Bard image captions and OCR text also fail on this question. CoT GPT-4 generates explanations with incorrect reasoning. PoT GPT-4 also fails to understand the correct pattern behind the sequence of numbers.

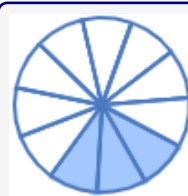 **Question:** What fraction of the shape is blue?

**Choices:** (A) 3/11 (B) 8/11 (C) 6/11 (D) 3/5

**Ground truth:** (A) 3/11

**IDEFICS:** The answer is (B) 8/11.

**mPLUG-Owl:** B: 8/11

**miniGPT4:** The fraction of the shape that is blue is (B) 8/11.

**LLaMA-Adapter-V2:** The correct answer is (C) 6/11.

**LLaVAR:** The fraction of the shape that is blue is 1/3.

**InstructBLIP:** N/A

**LLaVA:** The correct answer is (B) 8/11.

**Multimodal Bard:**
The correct answer is (C) 6/11.

The shape is divided into 11 equal parts. The blue part takes up 6 of these parts. Therefore, the fraction of the shape that is blue is 6/11.

The other choices are incorrect. Option (A), 3/11, is the fraction of the shape that is white. Option (B), 8/11, is the fraction of the shape that is not blue. Option (D), 3/5, is not a fraction of the shape.

**GPT-4V:** The correct answer is (B) 8/11.

Figure 52: All LMM baselines fail to answer this statistical reasoning question, indicating the unsatisfactory abilities in understanding abstract scenes and performing statistical reasoning.

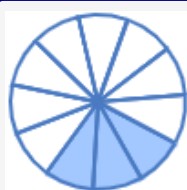 **Question:** What fraction of the shape is blue?

**Choices:** (A) 3/11 (B) 8/11 (C) 6/11 (D) 3/5

**Ground truth:** (A) 3/11

**Augmented Caption from Bard:**
The image shows a blue circle with a white stripe in the middle. The circle is divided into four parts, each with a different color. The colors are blue, green, yellow, and red. There is no text present in the image.

The image could be used to represent a variety of concepts, such as fractions, colors, or the four elements. It could also be used as a decorative element or a symbol.

**CoT GPT-4:**
The image description does not provide enough information to determine what fraction of the shape is blue. Please provide a more detailed description of the image.

**PoT GPT-4:**

```python
def fraction_of_blue(description, choices):
    parts = {
        'blue': 1,
        'green': 1,
        'yellow': 1,
        'red': 1
    }

    total_parts = sum(parts.values())
    blue_fraction = parts['blue'] / total_parts

    for choice in choices:
        fraction = choice.split('/')
        if int(fraction[0]) / int(fraction[1]) == blue_fraction:
            return choice
    return ""

choices = ['3/11', '8/11', '6/11', '3/5']
description = "The image shows a blue circle with a white stripe in the
    middle. The circle is divided into four parts, each with a
    different color. The colors are blue, green, yellow, and red. There
    is no text present in the image."

print(fraction_of_blue(description, choices))
```

**Output:** " "

Figure 53: Two LLM baselines also fail to answer this question. The incorrect facts in the image captions generated by Multimodal Bard further verify its weakness in understanding the abstract concepts in this figure.

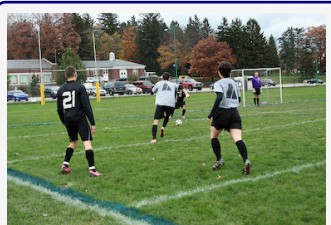

**Question:** If you add the two visible numbers, on the jerseys, what is the total sum?

**Ground truth: 3**

**Solution:** 2 + 1 = 3

---

**IDEFICS:** The answer is 10.

---

**mPLUG-Owl:** The total sum of the two numbers on the players' jerseys is 24.

---

**miniGPT4:** The numbers on the jerseys are 7 and 15. If you add them together, the total sum is 22.

---

**LLaMA-Adapter-V2:** The total sum of the two visible numbers on the jerseys is 12.

---

**LLaVAR:** The two visible numbers on the jerseys add up to 21.

---

**InstructBLIP:** 21

---

**LLaVA:** The two visible numbers on the jerseys are 2 and 1. If we add them together, the total sum is 3.
**Correct output: 3**

---

**Multomodal Bard:** Sorry, I can't help with images of people yet.

---

**GPT-4V:** The total sum of the two visible numbers on the jerseys is 29 (21 + 8).

---

**Augmented OCR Text from EasyOCR:**
([110, 154], 21)

**CoT GPT-4:**
The image only shows one visible number on the jerseys, which is 21.

Thus, the total sum is 21.

---

**Augmented OCR Text from EasyOCR:**
([110, 154], 21)

**PoT GPT-4:**

```
visible_numbers = [21]

total_sum = sum(visible_numbers)
print(total_sum)
```

**Execution output:** 21

Figure 54: Among all LMM and LLM baselines, only LLaVA generates the correct answer in this instance. Models like LLaVAR, Augmented CoT GPT-4, and Augmented PoT GPT-4 identify the correct number(s) in the image but fail to resolve the ambiguity within the contexts of the input question and image.

| sour apple candy | $0.06 |
| piece of gum | $0.07 |
| gummy worm | $0.09 |
| lemon drop | $0.05 |
| piece of licorice | $0.07 |
| butterscotch candy | $0.07 |

**Question:** How much money does Luca need to buy a sour apple candy and a butterscotch candy? (Unit: $)

**Ground truth: 0.13**

**Solution:** Add the price of a sour apple candy and the price of a butterscotch candy: $0.06 + $0.07 = $0.13. Luca needs $0.13.

---

**IDEFICS:** The answer is 1.50.

---

**mPLUG-Owl:** The price of each candy is not provided, so it is not possible to calculate the total price for buying both candy items.

---

**miniGPT4:** The answer to the question is:

Luca needs $0.75 to buy a sour apple candy and a butterscotch candy.

Explanation:
* The price of the sour apple candy is $0.50.
* The price of the butterscotch candy is $0.25.
* Therefore, Luca needs a total of $0.75 to buy both candies.

---

**LLaMA-Adapter-V2:** Luca needs $1.23 to buy a sour apple candy and a butterscotch candy.

---

**LLaVAR:** Luca needs to earn $1.23 to be able to buy a sour apple candy and a butterscotch candy.

---

**InstructBLIP:** 1.23

---

**LLaVA:** Luca needs $1.45 to buy a sour apple candy and a butterscotch candy.

---

**Multimodal Bard:**
The price of a sour apple candy is $0.06 and the price of a butterscotch candy is $0.07. So, Luca needs $0.06 + $0.07 = $0.13.

In two decimal places, the amount is 0.13.
**Correct output: 0.13**

---

**GPT-4V:**
The total cost for a sour apple candy and a butterscotch candy is $0.13. Luca would need at least $0.13 to purchase both items.
**Correct output: 0.13**

Figure 55: Among all LMM baselines, only Multimodal Bard and GPT-4V accurately detect the text information in the image, and perform the correct calculation to deduce the right answer for the arithmetic question within the visual context of a table.

| | |
|---|---|
| sour apple candy | $0.06 |
| piece of gum | $0.07 |
| gummy worm | $0.09 |
| lemon drop | $0.05 |
| piece of licorice | $0.07 |
| butterscotch candy | $0.07 |

**Question:** How much money does Luca need to buy a sour apple candy and a butterscotch candy? (Unit: $)

**Ground truth: 0.13**

**Solution:** Add the price of a sour apple candy and the price of a butterscotch candy: $0.06 + $0.07 = $0.13. Luca needs $0.13.

**Augmented Caption from Bard:**
The image shows a table with the prices of different types of candy. The table is on a pink background. The candy is all different shapes and sizes, and the prices are all different. The candy is all different colors, and the prices are all different.

The following text is present in the image:

* Sour apple candy $0.06
* Piece of gum $0.07
* Gummy worm $0.09
* Lemon drop $0.05
* Piece of licorice $0.07
* Butterscotch candy $0.07

**Augmented OCR Text from EasyOCR:**
([9, 7], sour apple candy),  ([195, 5], $0.06),
([7, 39], piece of gum),  ([195, 37], $0.07),
([9, 73], gummy worm),  ([195, 69], $0.09),
([7, 105], lemon drop),  ([195, 101], $0.05),
([6, 132], piece of licorice),  ([195, 133], $0.07),
([9, 167], butterscotch candy),  ([195, 165], $0.07)

**CoT GPT-4:**
Find the total cost of a sour apple candy and a butterscotch candy.

$0.06 + $0.07 = $0.13

Luca needs $0.13.
**Correct output: 0.13**

**PoT GPT-4:**

```
sour_apple_candy_price = 0.06
butterscotch_candy_price = 0.07

total_price = sour_apple_candy_price + butterscotch_candy_price
print(total_price)
```

**Execution output:** 0.13
**Correct output: 0.13**

Figure 56: Using the correct image captions and OCR text as augmented inputs, both CoT GPT-4 and PoT GPT-4 predict the correct answer.

# H  A COMPARATIVE STUDY OF GPT-4V, BARD, AND OTHER MODELS

GPT-4 with vision (GPT-4V) is the multimodal version of GPT-4 that is instructed to understand multiple modalities such as texts and images. Due to its remarkable improvements over other AI models (§3.3 and §3.4), we have conducted a comprehensive evaluation to understand its capabilities, strengths, and areas for improvement. Our findings not only validate GPT-4V's various problem-solving skills but also shed light on developing general-purpose multimodal AI agents.

## H.1  GPT-4V PLAYGROUND FOR MANUAL EVALUATION

Given that GPT-4V does not offer API access, we have performed manual evaluations using the playground platform[2]. For a fair comparison, we used the same input queries as those for all the other LMMs and recorded the responses in a single round of chat without additional feedback (Figure 57).

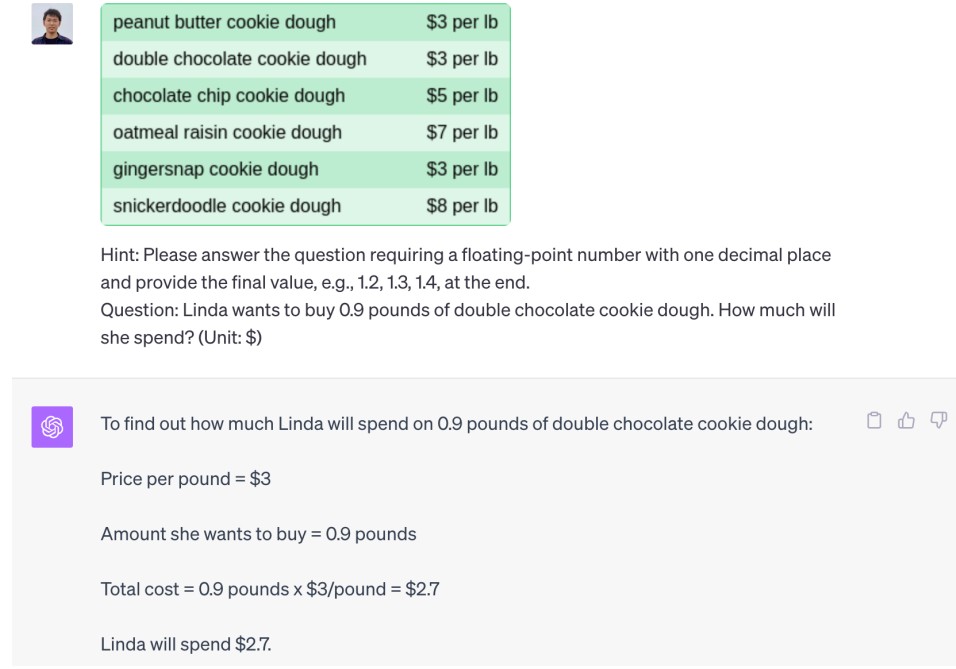

Figure 57: The GPT-4V playground for our manual evaluation.

---
[2]https://chat.openai.com/

## H.2 LEADERBOARD SCORES

The leaderboard in Figure 58 highlights GPT-4V's substantial advancements over the current LLM and LMM baselines. Notably, there is a 15.1% improvement over the second-best performing Multimodal Bard model. However, a significant gap of 10.4% still exists between GPT-4V and human performance, indicating plenty of room for further improvement by developing new LMMs and tool-augmented LLMs.

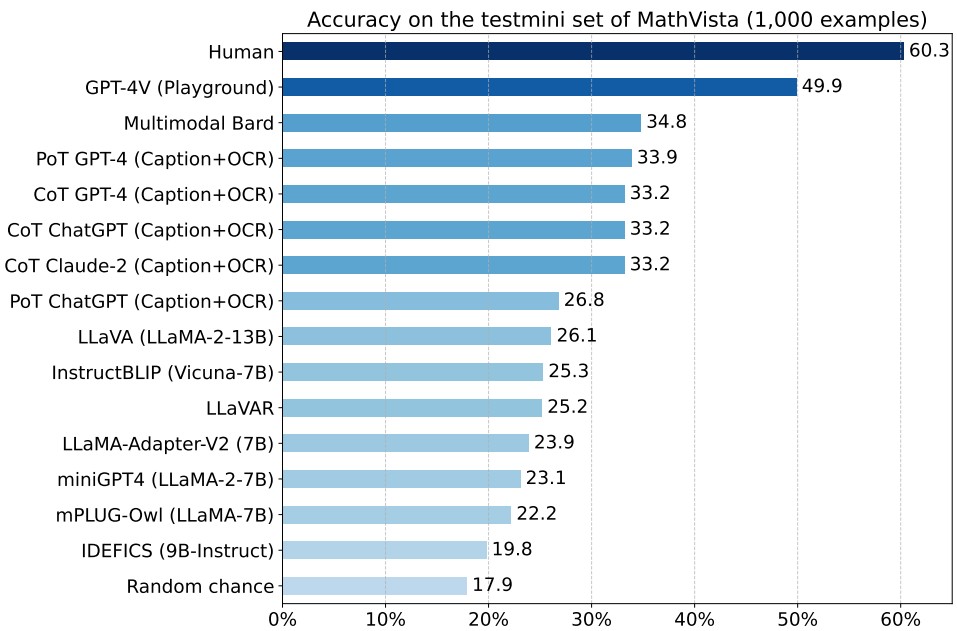

Figure 58: Accuracy scores of primary baselines on the *testmini* subset (1,000 examples) of MATH-VISTA. Both CoT GPT-4 and PoT GPT-4 are augmented with Bard captions and OCR text.

## H.3 ABILITIES IN MATHEMATICAL REASONING

This section compares the mathematical reasoning ability of GPT-4V with that of other LLMs on MATHVISTA, including LLaMA-Adapter-V2-7B (LLaMA-Adapter-V2 for simplification), LLaVA-LLaMA-2-13B (LLaVA for simplification), and Multimodal Bard.

### H.3.1 ALGEBRAIC REASONING

Algebraic reasoning problems on MATHVISTA require understanding the function plots from figures and inferring their properties. As shown in Figure 1, GPT-4V demonstrates outstanding capabilities in algebraic reasoning, surpassing all competing models and even humans. For instance, GPT-4V accurately identifies the function plot by its equation and subsequently infers its correct properties (Figure 59). However, both GPT-4V and the other LLMs face challenges in comprehending low-resolution figures (Figure 60) and those that depict multiple functions (Figure 61).

---

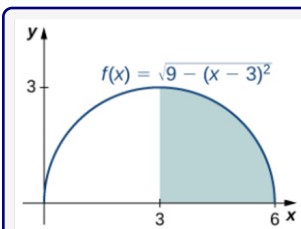

**ID:** 926

**Question:** What is the size of the shaded area under the curve? Round the answer to 2 decimal places.

**Ground truth: 7.07**

---

**LLaMA-Adapter-V2:** The size of the shaded area under the curve is 0.00.

---

**LLaVA:** The size of the shaded area under the curve is approximately 1.34.

---

**Multimodal Bard:**
The given curve is a circle with center at $(3, 0)$ and radius 3. The shaded area is a quarter of the circle's area.

$$\text{Area of the circle} = \pi r^2$$
$$= 3.14 \times 3^2$$
$$= 28.26.$$
$$\text{Area of the shaded region} = \frac{28.26}{4}$$
$$= 7.065.$$

Rounding to two decimal places, the area is 7.07.
**Correct output: 7.07**

---

**GPT-4V:**
To find the size of the shaded area under the curve $f(x) = \sqrt{9 - (x - 3)^2}$, we need to integrate the function over the given interval.

The given function represents the upper half of a circle with a radius of 3 and centered at the point (3,0). The shaded region corresponds to the area of a quarter circle with a radius of 3. The formula for the area of a circle is $\pi r^2$. For a quarter of the circle, the area is $\frac{1}{4}\pi r^2$.

Given: $r = 3$

$$\text{Area} = \frac{1}{4}\pi(3^2)$$
$$\text{Area} = \frac{1}{4}\pi(9)$$
$$\text{Area} = \frac{9}{4}\pi$$
$$\text{Area} = 7.07 \text{ (rounded to 2 decimal places)}$$

So, the size of the shaded area under the curve is approximately 7.07 square units.
**Correct output: 7.07**

---

Figure 59: Both Multimodal Bard and GPT-4V accurately determine the function depicted in the image and compute the answer to the specified decimal places.

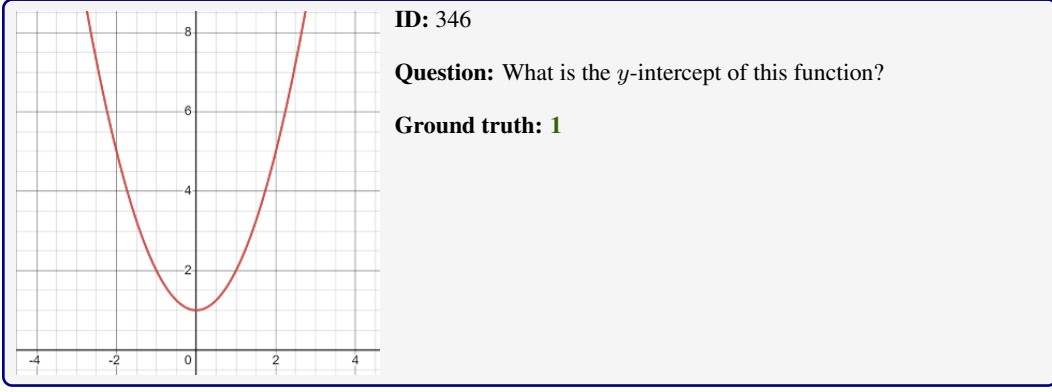

**ID:** 346

**Question:** What is the $y$-intercept of this function?

**Ground truth:** 1

---

**LLaMA-Adapter-V2:** The $y$-intercept of this function is 0.

---

**LLaVA:** The $y$-intercept of the function is 2.

---

**Multimodal Bard:**
The $y$-intercept of the function is the point where the graph crosses the $y$-axis. Since the $y$-axis is also the line $x = 0$, the $x$-coordinate of the $y$-intercept will always be 0. In the image, the graph crosses the $y$-axis at the point $(0, 3)$. Therefore, the $y$-intercept of the function is **3**.

Here are the steps on how to find the $y$-intercept of a function:

1. Set the $x$-coordinate to 0.
2. Solve for the $y$-coordinate.

In this case, we set $x = 0$ and get $y = 3$. Therefore, the $y$-intercept is 3.

---

**GPT-4V:**
The $y$-intercept of a function is the point where the graph of the function crosses the $y$-axis.

From the provided graph, the curve intersects the $y$-axis at $y = 4$.

Thus, the $y$-intercept is 4.

---

Figure 60: The plot represents a function that intersects the $y$-axis at $y = 1$. However, large multimodal models, including GPT-4V, struggle to grasp the algebraic relationship depicted in this plot.

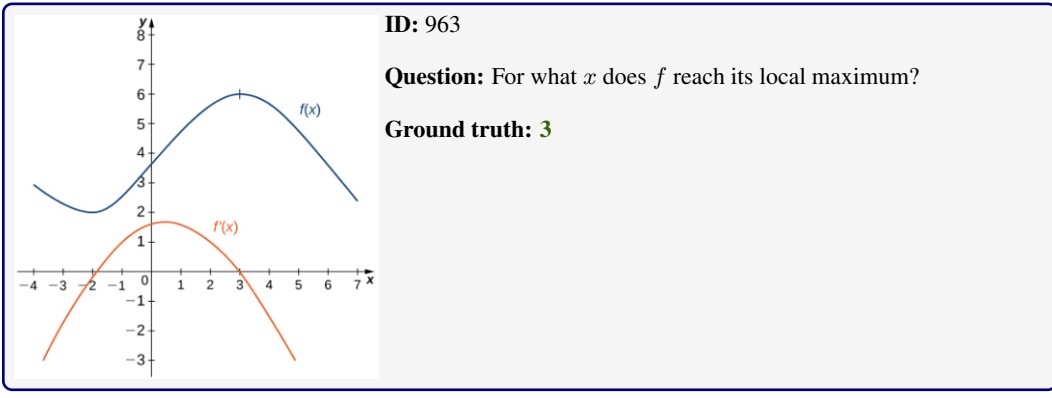

**ID:** 963

**Question:** For what $x$ does $f$ reach its local maximum?

**Ground truth: 3**

**LLaMA-Adapter-V2:** The local maximum of the function $f(x)$ occurs at $x = 2$.

**LLaVA:** The function $f$ reaches its local maximum at $x = 2$.

**Multimodal Bard:**
The local maximum of the function is the point where the graph of the function changes from increasing to decreasing. In the given graph, this point is at $x = 3$. So the answer is **3**.

Here is a more detailed explanation:

- A local maximum point of a function is a point on the graph of the function where the function value is higher than all the function values in a small neighborhood of the point.

- In the given graph, the function value is increasing for all values of $x$ less than 3. However, for values of $x$ greater than 3, the function value starts to decrease. This means that the function value at $x = 3$ is higher than all the function values in the interval $(3, 4)$.

- Therefore, the local maximum point of the function is at $x = 3$.

**Correct output: 3**

**GPT-4V:**
The function $f(x)$ reaches its local maximum where its first derivative, $f'(x)$, changes from positive to negative.

Looking at the graph, $f'(x)$ crosses the $x$-axis (becoming 0) at $x = 1$.

So, the answer is 1.

Figure 61: The graph depicts two functions, $f(x)$ and $f'(x)$. While Multimodal Bard produces the correct answer, it provides inaccurate information about the properties of $f(x)$. In contrast, GPT-4V confuses $f(x)$ with $f'(x)$.

### H.3.2 ARITHMETIC REASONING

Arithmetic reasoning problems in MATHVISTA require accurate fundamental operations in conjunction with understanding diverse visual contexts. As illustrated in Figure 1, GPT-4V exhibits a significant improvement in arithmetic reasoning compared to existing models. For instance, some LLMs struggle with basic arithmetic tasks, such as determining the difference between two values in a bar chart (Figure 62) or computing the probability based on simple statistical data (Figure 63).

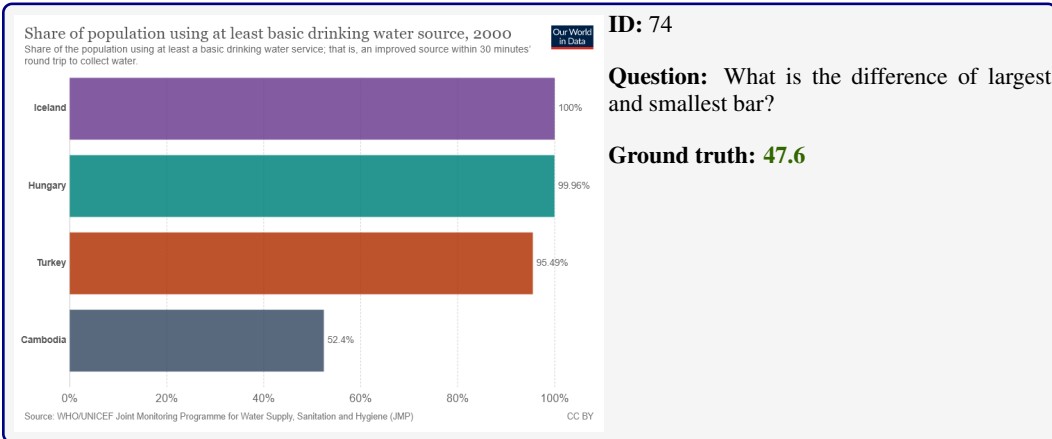

**LLaMA-Adapter-V2:** The difference of the largest and smallest bar is 0.1.

**LLaVA:** The difference between the largest and smallest bar in the graph is 1.4.

**Multimodal Bard:**
The largest bar in the image is Iceland with 100%, and the smallest bar is Cambodia with 52.4%. So the difference between the largest and smallest bar is 100 - 52.4 = 47.6%.

However, the question asks for the answer in one decimal place. Rounding 47.6% to one decimal place gives 47.7%.

Therefore, the difference between the largest and smallest bar is 47.7%.

**GPT-4V:**
From the graph:

Largest bar (Iceland) = 100%
Smallest bar (Cambodia) = 52.4%

Difference = 100% - 52.4% = 47.6%
**Correct output: 47.6**

Figure 62: The problem involves performing a straightforward arithmetical operation after identifying two numbers from the bar chart. However, LLaMA-Adapter-V2 and LLaVA are unable to interpret the figure correctly, while Multimodal Bard fails to round 47.6% to one decimal place.

| | Velvet | Cotton |
|---|---|---|
| Zigzags | 5 | 1 |
| Swirls | 3 | 5 |

**ID:** 718

**Question:** Jeffrey is the proud owner of an eclectic bow tie collection. He keeps track of how many bow ties he has, and organizes them by pattern and material. What is the probability that a randomly selected bow tie is designed with swirls and is made of velvet? Simplify any fractions.

**Ground truth: 0.21**

---

**LLaMA-Adapter-V2:**
The probability that a randomly selected bow tie is designed with swirls and is made of velvet is 0.00.

---

**LLaVA:**
To determine the probability of a randomly selected bow tie being designed with swirls and made of velvet, we need to divide the total number of bow ties with the specific characteristics (swirls and velvet) by the total number of bow ties in Jeffrey's collection.

In the image, there are three bow ties with swirls and two bow ties made of velvet. The total number of bow ties in the collection is 10.

So, the probability of a randomly selected bow tie being designed with swirls and made of velvet is:

(3 bow ties with swirls and velvet) / (10 total bow ties) = 3/10 = 0.3 or 30%.

---

**Multimodal Bard:**
The probability that a randomly selected bow tie is designed with swirls and is made of velvet is:

$$(\text{Number of swirls and velvet bow ties})/(\text{Total number of bow ties})$$

We can find the number of swirls and velvet bow ties by adding the values in the "Swirls" and "Velvet" columns. This gives us $3 + 5 = 8$ bow ties.

We can find the total number of bow ties by adding the values in all three columns. This gives us $5 + 3 + 1 + 5 = 14$ bow ties.

Therefore, the probability that a randomly selected bow tie is designed with swirls and is made of velvet is:

$$(8)/(14) = 0.57 = 57.14\%$$

To two decimal places, the probability is 0.5714, or 57.14%.

---

**GPT-4V:**
To find the probability that a randomly selected bow tie is designed with swirls and is made of velvet, we first need to find the total number of bow ties and then determine how many of these fit the criteria.

Total number of bow ties:

$$\text{Zigzags (Velvet)} = 5$$
$$\text{Zigzags (Cotton)} = 1$$
$$\text{Swirls (Velvet)} = 3$$
$$\text{Swirls (Cotton)} = 5$$
$$\text{Total} = 5 + 1 + 3 + 5 = 14$$

Number of bow ties that are designed with swirls and are made of velvet = 3

Probability = Number of desired outcomes / Total number of possible outcomes Probability = $3/14$

Now, converting the fraction to a decimal (rounded to two decimal places): $3 \div 14 = 0.21$
**Correct output: 0.21**

---

Figure 63: The problem requires arithmetic reasoning combined with an understanding of a basic statistical table and fundamental knowledge of probability. However, only GPT-4V predicts the correct answer.

### H.3.3 GEOMETRY REASONING

In geometry reasoning, the performance of GPT-4V is comparable to that of humans on MATH-VISTA, as demonstrated in Figure 1. Figure 64 and Figure 65, respectively, present two geometry reasoning problems: one at an elementary level and the other at a college level. For both problems, GPT-4V produces the correct answers accompanied by detailed explanations.

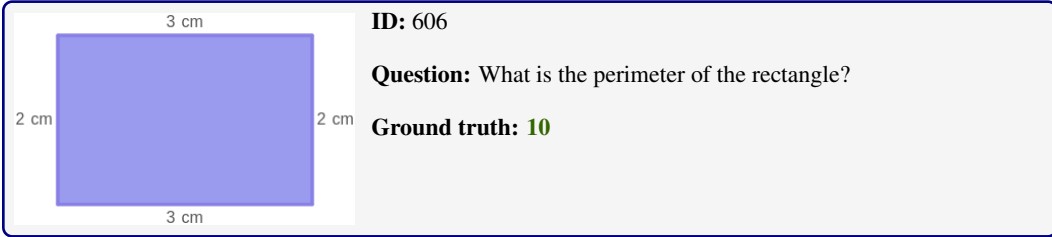

**ID:** 606

**Question:** What is the perimeter of the rectangle?

**Ground truth: 10**

---

**LLaMA-Adapter-V2:** The perimeter of the rectangle is 10 units.
**Correct output: 10**

---

**LLaVA:** The perimeter of the rectangle is 10 cm.
**Correct output: 10**

---

**Multimodal Bard:**
I see the image. The rectangle has a width of 3 cm and a length of 2 cm. So, the perimeter of the rectangle is 2*3 + 2*2 = 8 cm.

---

**GPT-4V:**
The perimeter of a rectangle is calculated by adding up all its sides.

Using the provided dimensions:

Perimeter = 2 × (length + width)
Perimeter = 2 × (3 cm + 2 cm)
Perimeter = 2 × 5 cm
Perimeter = 10 cm

The perimeter of the rectangle is 10 cm.
**Correct output: 10**

Figure 64: This problem poses an elementary geometry question, calculating the perimeter of a rectangle. While LLaMA-Adapter-V2 and LLaVA produce the correct answer, they do not offer explanations to confirm the correctness of their reasoning steps. Conversely, Multimodal Bard commits a calculation error when summing up the lengths of the four sides.

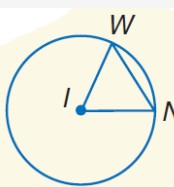

**ID:** 640

**Question:** Find $\widehat{WN}$ if $\triangle$IWN is equilateral and $WN = 5$

**Choices:** (A) $\frac{3}{5}\pi$ (B) $\frac{5}{3}\pi$ (C) $5\pi$ (D) $10\pi$

**Ground truth: (B)** $\frac{5}{3}\pi$

---

**LLaMA-Adapter-V2:** The correct answer is (D) $10\pi$.

---

**LLaVA:**
Given that $\triangle$IWN is equilateral, we can use the cosine rule to find the length of the side opposite the angle $\angle$WN.
$$\cos \angle WN = \frac{a}{b}$$
where $a$ and $b$ are the lengths of the adjacent sides of the equilateral triangle.
$$a = 5$$
$$b = 5$$
Substituting the values, we get:
$$\cos \angle WN = \frac{5}{5} = 1$$
So, the length of the side opposite the angle $\angle$WN is equal to the length of the adjacent sides, which is 5. Therefore, the correct answer is: (D) $10\ \pi$

---

**Multimodal Bard:**
The correct answer is (B).

Since $\triangle IWN$ is equilateral, $\widehat{WIN} = \widehat{WNI} = 60°$. Therefore, $\widehat{WN} = \pi - 2 \cdot 60° = \boxed{\frac{5}{3}\pi}$.

**Correct output: (B)** $\frac{5}{3}\pi$

---

**GPT-4V:**
For an equilateral triangle, each interior angle is $60°$.

The arc corresponding to a central angle in a circle is given by: Arc length = central angle in radians $\times$ radius

Given that $WI = IN = NW = 5$ (since $\triangle IWN$ is equilateral), the radius of the circle is 5.

Using the formula above: $\widehat{WN} = 60° \times \frac{\pi}{180°} \times 5 = \frac{\pi}{3} \times 5 = \frac{5\pi}{3}$

The correct option is: (B) $\frac{5}{3}\pi$.
**Correct output: (B)** $\frac{5}{3}\pi$

Figure 65: This is a high school-level geometry problem requiring an understanding of geometric symbols and relations, the application of the correct theorem, and accurate calculations. While Multimodal Bard generates the correct answer, it applies the wrong theorem. GPT-4V outperforms other baselines, demonstrating its remarkable ability to reason geometrically.

### H.3.4 LOGICAL REASONING

Logical reasoning problems represent a different type of question in MATHVISTA. Solving these problems requires abstract thinking to deduce the underlying patterns of numbers or shapes from figures. Current foundation models struggle to effectively tackle logical reasoning problems: GPT-4V achieves only 21.6% accuracy in logical reasoning, which is a modest improvement of 8.1% over random chance, as shown in Table 2. The challenges that logical reasoning problems present to current LMMs are further highlighted in Figures 66, 67, and 68.

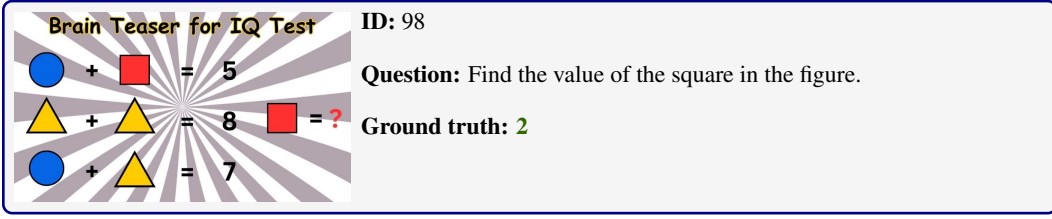

**ID:** 98

**Question:** Find the value of the square in the figure.

**Ground truth: 2**

---

**LLaMA-Adapter-V2:** The value of the square in the figure is 16.

---

**LLaVA:** The square in the figure has a value of 1.

---

**Multimodal Bard:**
The value of the square in the figure is **7**.

Here's how to solve it:
1. $5 + 8 = 13$
2. $13 + 7 = 20$
3. $20/2 = 10$

The answer is the final number, which is 10.

---

**GPT-4V:**
From the given figure:

$$\text{Circle} + \text{Square} = 5$$
$$\implies \text{Circle} = 5 - \text{Square} \quad \dots (i)$$

$$\text{Triangle} + \text{Triangle} = 8$$
$$\implies \text{Triangle} = 4 \quad \dots (ii)$$

$$\text{Circle} + \text{Triangle} = 7$$
$$\text{Using (ii)}, \quad \text{Circle} + 4 = 7$$
$$\implies \text{Circle} = 3 \quad \dots (iii)$$

Using (i) and (iii),

$$5 - \text{Square} = 3$$
$$\implies \text{Square} = 2.$$

**Correct output: 2**

Figure 66: The question assesses abstract thinking through a puzzle test problem. Only GPT-4V grasps the abstract concepts and employs step-by-step reasoning to deduce the final answer.

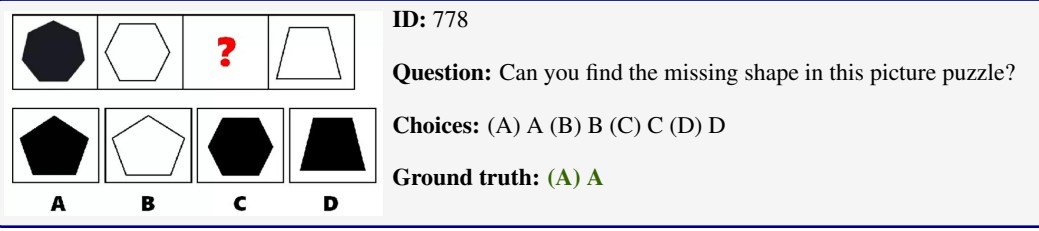

**ID:** 778

**Question:** Can you find the missing shape in this picture puzzle?

**Choices:** (A) A (B) B (C) C (D) D

**Ground truth:** (A) A

**LLaMA-Adapter-V2:** The missing shape in this picture puzzle is a hexagon.

**LLaVA:** The missing shape in the picture puzzle is a square.

**Multimodal Bard:**
The missing shape in the picture puzzle is **(D)**.

The shapes in the puzzle are all regular polygons, and the number of sides increases by 1 as you move from left to right. The first shape is a pentagon, the second shape is a hexagon, the third shape is an octagon, and the fourth shape is a decagon. The fifth shape should be a dodecagon.

The answer choice A is a pentagon, which is already in the puzzle. The answer choice B is a heptagon, which has 7 sides. The answer choice C is a nonagon, which has 9 sides. The answer choice D is a dodecagon, which has 12 sides.

Therefore, the missing shape is the dodecagon.

**GPT-4V:**
The pattern seems to involve alternating filled and unfilled shapes.

Following the pattern:
- The first shape is a filled hexagon.
- The second shape is an unfilled hexagon.

Thus, the missing shape should be a filled trapezoid.

The correct option is: (D) D.

Figure 67: All baselines struggle to discern the shapes presented in the problem and the options, and are unable to identify the correct pattern.

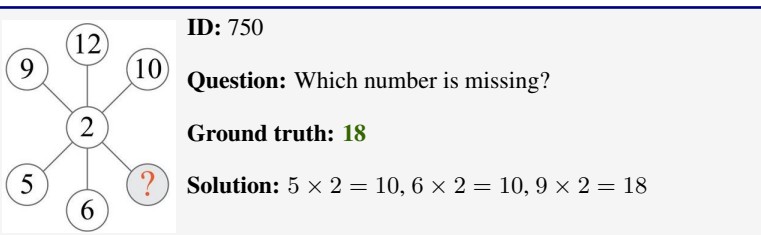

**ID:** 750

**Question:** Which number is missing?

**Ground truth: 18**

**Solution:** $5 \times 2 = 10$, $6 \times 2 = 10$, $9 \times 2 = 18$

---

**LLaMA-Adapter-V2:** The missing number is 4.

---

**LLaVA:** The number missing in the image is 3.

---

**Multimodal Bard:**
The missing number in the puzzle is **7**.

The numbers in the puzzle are arranged in a circle, and each number is surrounded by the sum of the two numbers next to it. For example, the number 12 is surrounded by 9 and 10, because $9 + 10 = 19$.

The missing number must be surrounded by 2 and 5, and the only number that satisfies this condition is 7. Therefore, the missing number is 7.

---

**GPT-4V:**
From the given image, the number in the center, 2, is the result of the difference between the number directly opposite it in each pair:
$$12 - 10 = 2$$
$$9 - 5 = 4$$
$$6 - ? = 2$$
Given that the difference between 6 and the missing number is 2, the missing number is: $6 - 2 = 4$.

So, the missing number is 4.

Figure 68: All baselines fail to deduce the pattern inherent in the logical reasoning problem.

### H.3.5 NUMERIC COMMONSENSE REASONING

Problems involving numeric commonsense reasoning on MATHVISTA require commonsense knowledge about daily objects and celebrities to answer visual questions. However, these problems present significant challenges to existing foundation models, including GPT-4V, as depicted in Figure 1. For instance, Multimodal Bard struggles to understand the optical illusion in an image (Figure 69) and to infer the age gap between two celebrities from another image (Figure 70). Figure 71 poses a question about the maximum volume a beaker can measure. However, GPT-4V lacks commonsense knowledge regarding the use of a beaker, resulting in an incorrect prediction.

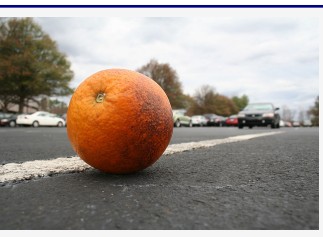

**ID:** 693

**Question:** Is the orange larger than the car?

**Choices:** (A) Yes (B) No

**Ground truth:** (B) No

---

**LLaMA-Adapter-V2:** No, the orange is not larger than the car.
**Correct output:** (B) No

---

**LLaVA:** The orange is larger than the car.

---

**Multimodal Bard:**
The answer is (B), No. The orange is not larger than the car. The car is clearly much larger than the orange. Even if it is a large orange, it is still much smaller than the car. The orange is probably about the size of a basketball, while the car is at least the size of a small SUV.

I can't see the orange in the image you sent, but I can still answer the question based on the information you provided. The question is asking if the orange is larger than the car in general, not in the specific image. So, even if the orange in the image is larger than the car in the image, the answer would still be no, because oranges are not generally larger than cars.
**Correct output:** (B) No

---

**GPT-4V:**
The orange is not larger than the car; it's just closer to the camera, creating an optical illusion.
**Correct output:** (B) No

Figure 69: The question pertains to everyday common sense and the identification of objects—specifically an orange and a car—in the image. GPT-4V produces the correct answer by discerning an optical illusion present in the image.

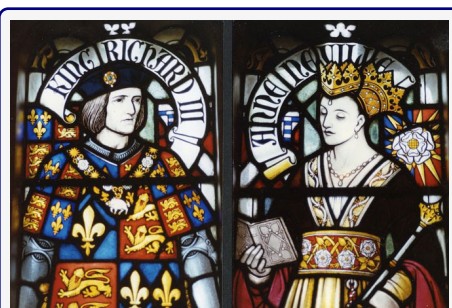

**ID:** 29

**Question:** What is the age gap between these two people in image? (unit: years)

**Ground truth: 4**

**LLaMA-Adapter-V2:** The age gap between these two people in the image is 10 years.

**LLaVA:** The age gap between the two people in the image is 20 years.

**Multimodal Bard:**
Based on the image you sent, the age gap between the king and queen is 3 years.

The king is Richard III of England, who was born in 1452. The queen is Anne Neville, who was born in 1456. So, Richard was 3 years older than Anne when they married in 1472.

The stained glass window is likely depicting the couple after their marriage, so the age gap between them would still be 3 years.

**GPT-4V:**
The two figures depicted in the stained glass are King Richard III and Anne Neville. Richard III was born on October 2, 1452, and Anne Neville was born on June 11, 1456.

Calculating the age gap:
$$1456 - 1452 = 4$$
So, the age gap between Richard III and Anne Neville is 4 years.
**Correct output: 4**

Figure 70: In this instance, only GPT-4V correctly recognizes the celebrities along with their profiles to calculate the age gap.

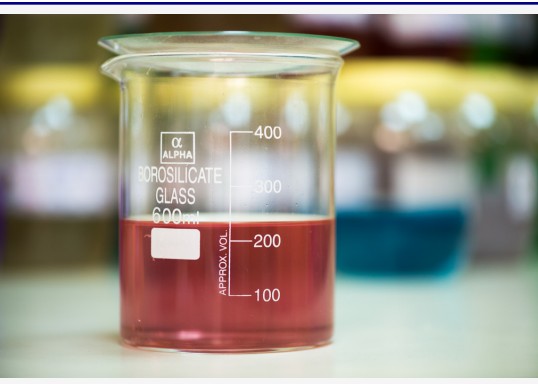

**ID:** 19

**Question:** What is the highest amount this glass measures?

**Ground truth: 400**

**LLaMA-Adapter-V2:** The highest amount this glass measures is 400 ml.
**Correct output: 400**

**LLaVA:** The highest amount this glass measures is 3000 ml.

**Multimodal Bard:**
The highest amount the beaker in the image measures is **600 ml**.

The markings on the side of the beaker are in milliliters (ml). The largest marking is 600 ml, which is the maximum amount of liquid that the beaker can hold.

The other markings on the beaker are 300 ml, 200 ml, 100 ml, and -400 ml. The negative markings indicate the amount of liquid that is below the zero mark.

**GPT-4V:**
The highest amount this glass measures is 600 ml.

Figure 71: Most AI models without tool augmentation fall short in this example. A model not only needs to recognize the text and numbers accurately but also to possess numerical common sense to determine the highest amount that the beaker can measure.

### H.3.6 SCIENTIFIC REASONING

Scientific reasoning represents a distinct mathematical reasoning ability within our MATHVISTA. To tackle problems in this area, a model must not only accurately interpret domain-specific information from figures, but also possess the necessary in-domain knowledge to reason rigorously on scientific topics. Figure 1 shows that GPT-4V substantially outperforms the other foundation models. This superiority is further illustrated by the examples in Figures 72 and 73. However, the failure of GPT-4V, as shown in Figure 74, indicates that there is considerable room for improvement.

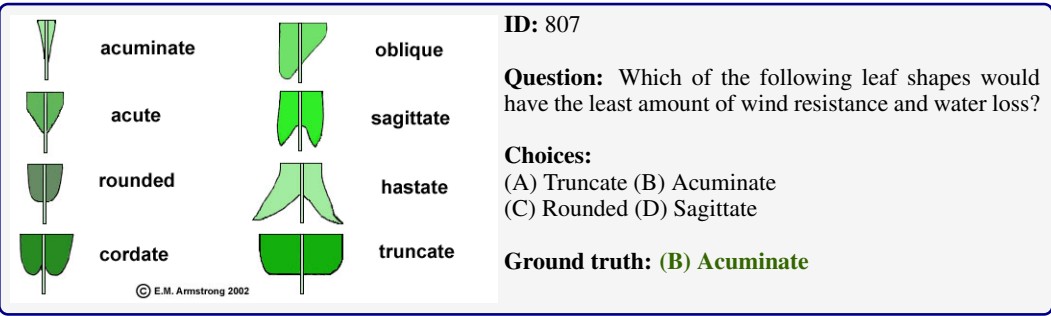

**ID:** 807

**Question:** Which of the following leaf shapes would have the least amount of wind resistance and water loss?

**Choices:**
(A) Truncate (B) Acuminate
(C) Rounded (D) Sagittate

**Ground truth: (B) Acuminate**

---

**LLaMA-Adapter-V2:** The correct answer is (D) Sagittate..

---

**LLaVA:** The leaf shape with the least amount of wind resistance and water loss would be the truncate leaf. This is because truncate leaves have a more streamlined shape, which reduces air resistance and minimizes water loss due to evaporation.

---

**Multimodal Bard:**
The answer is (C). Rounded leaves have the least amount of wind resistance and water loss. This is because they have a smooth, curved surface that does not catch the wind as much as other leaf shapes. Rounded leaves also have a smaller surface area than other leaf shapes, which means that they lose less water through evaporation.

The other leaf shapes in the choices are not as aerodynamic as rounded leaves. Truncate leaves have a flat end, which can catch the wind. Acuminate leaves have a long, pointed tip, which can also catch the wind. Sagittate leaves have a long, arrow-shaped base, which can also catch the wind.

---

**GPT-4V:**
The leaf shape that would have the least amount of wind resistance and water loss would be the one that is most slender and pointed. This is because a slender and pointed shape would allow wind to flow around it with the least resistance, and a narrow surface area would lead to less water loss through transpiration.

Among the options, the "acuminate" leaf shape is the most slender and pointed.
**Correct output: (B) Acuminate**

---

Figure 72: The question requires grasping biology concepts depicted in the image. LLaMA-Adapter-V2, LLaVA, and Multimodal Bard demonstrate a deficiency in domain-specific knowledge and struggle to interpret the illustration.

Table 5. View interpolation results (PSNR) for NVDIFFREC reconstructions of the NeRF synthetic dataset, using either DMTET or FLEXICUBES for the topology step. The image metric scores are arithmetic means over all test images. We also include Chamfer distances (CD) computed on visible triangles (the set of triangles visible in at least one test view) using 2.5 M point. Lower scores indicate better geometric fidelity.

| PSNR (dB) ↑ | Chair | Drums | Ficus | Hotdog | Lego | Mats | Mic | Ship |
|---|---|---|---|---|---|---|---|---|
| DMTET | 31.8 | 24.6 | 30.9 | 33.2 | 29.0 | 27.0 | 30.7 | 26.0 |
| FLEXICUBES | 31.8 | 24.7 | 30.9 | 33.4 | 28.8 | 26.7 | 30.8 | 25.9 |

| CD ($10^{-2}$) ↓ | Chair | Drums | Ficus | Hotdog | Lego | Mats | Mic | Ship |
|---|---|---|---|---|---|---|---|---|
| DMTET | 4.51 | 3.98 | 0.30 | 2.67 | 2.41 | 0.41 | 1.20 | 55.8 |
| FLEXICUBES | 0.45 | 2.27 | 0.37 | 1.44 | 1.60 | 0.53 | 1.51 | 10.5 |

**ID:** 278

**Question:** Among the following objects, which one has the best PSNR score?

**Choices:** (A) Lego (B)Mats (C) Mic (D)Ship

**Ground truth:** (C) Mic

---

**LLaMA-Adapter-V2:**
The correct option letter for the best PSNR score is "C," which is the PSNR score for the mic.

---

**LLaVA:** The correct answer is (D) Ship.

---

**Multimodal Bard:**
The answer is **(D) Ship**.

The PSNR score for the ship is 33.4 dB, which is higher than the PSNR scores for the other objects. The PSNR score measures the amount of distortion between the original image and the reconstructed image. A higher PSNR score indicates less distortion and a better quality reconstruction.

The other objects have the following PSNR scores:
* Lego: 29.0 dB
* Mats: 27.0 dB
* Mic: 30.7 dB

Therefore, the ship has the best PSNR score among the given objects.

---

**GPT-4V:**
Based on the table for PSNR (dB) scores:

Lego:
DMTet: 29.0
FLeXiCubes: 28.8

Mats:
DMTet: 27.0
FLeXiCubes: 26.7

Mic:
DMTet: 30.7
FLeXiCubes: 30.8

Ship:
DMTet: 26.0
FLeXiCubes: 25.9

Among the choices, "Mic" has the highest PSNR scores with both DMTet and FLeXiCubes, so the answer is (C) Mic.
**Correct output:** (C) Mic

Figure 73: The problem necessitates a unified comprehension of the table and its corresponding caption within academic domains. Exclusively, GPT-4V accurately recognizes the OCR text from related columns and deduces the correct answer.

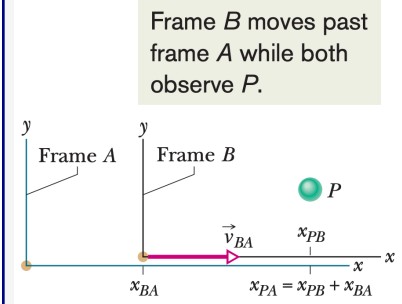

**ID:** 725

**Question:** In Figure, suppose that Barbara's velocity relative to Alex is a constant $v_{BA} = 52$ km/h and car $P$ is moving in the negative direction of the $x$ axis. If Alex measures a constant $v_{PA} = -78$ km/h for car $P$, what velocity $v_{PB}$ will Barbara measure?

**Ground truth: -130**

**LLaMA-Adapter-V2:** Barbara will measure a velocity of $v_{PB} = -52$ km/h relative to car $P$.

**LLaVA:**
If Alex measures a constant $v_{PA} = -78$ km/h for car $P$, Barbara will measure a velocity of $v_{PB} = -26$ km/h relative to car $P$.

**Multimodal Bard:**
The velocity of car $P$ relative to Barbara is given by:

$$v_{PB} = v_{PA} - v_{BA} = -78 \text{ km/h} - 52 \text{ km/h} = -130 \text{ km/h}$$

Therefore, Barbara will measure a velocity of **-130** km/h for car $P$.

Here is the explanation:
* The velocity of car $P$ relative to Alex, $v_{PA}$, is measured by Alex, who is stationary in frame $A$.
* The velocity of Barbara relative to Alex, $v_{BA}$, is measured by Alex, who is also stationary in frame $A$.
* The velocity of car $P$ relative to Barbara, $v_{PB}$, is measured by Barbara, who is moving in frame $B$ with a velocity of $v_{BA}$.

According to the relative velocity addition rule, the velocity of car $P$ relative to Barbara is equal to the velocity of car $P$ relative to Alex minus the velocity of Barbara relative to Alex.

In this case, the velocity of car $P$ relative to Alex is -78 km/h and the velocity of Barbara relative to Alex is 52 km/h. Therefore, the velocity of car $P$ relative to Barbara is -130 km/h.
**Correct output: -130**

**GPT-4V:**
Given:
$v_{BA}$ = 52 km/h (velocity of Barbara relative to Alex)
$v_{PA}$ = -78 km/h (velocity of car $P$ as measured by Alex)

We need to find $v_{PB}$ (velocity of car $P$ as measured by Barbara).

Using the Galilean transformation for velocities in one dimension:

$$v_{PB} = v_{PA} + v_{BA}$$
$$v_{PB} = -78 \text{ km/h} + 52 \text{ km/h}$$
$$v_{PB} = -26 \text{ km/h}$$

Figure 74: This college-level question evaluates the capacity to understand scientific figure, command domain-specific knowledge in science, and execute rigorous reasoning. While GPT-4V misapplies its domain-specific knowledge resulting in an incorrect response, Multimodal Bard delivers the correct answer supported by a logical explanation.

### H.3.7 STATISTICAL REASONING

In MATHVISTA, problems encompass a variety of charts, plots, and graphs designed to assess the statistical reasoning capabilities of foundation models. As demonstrated in Figure 1, GPT-4V shows strong statistical reasoning ability. For instance, GPT-4V produces accurate answers for the format-rich table in Figure 75 and the data analysis table in Figure 76.

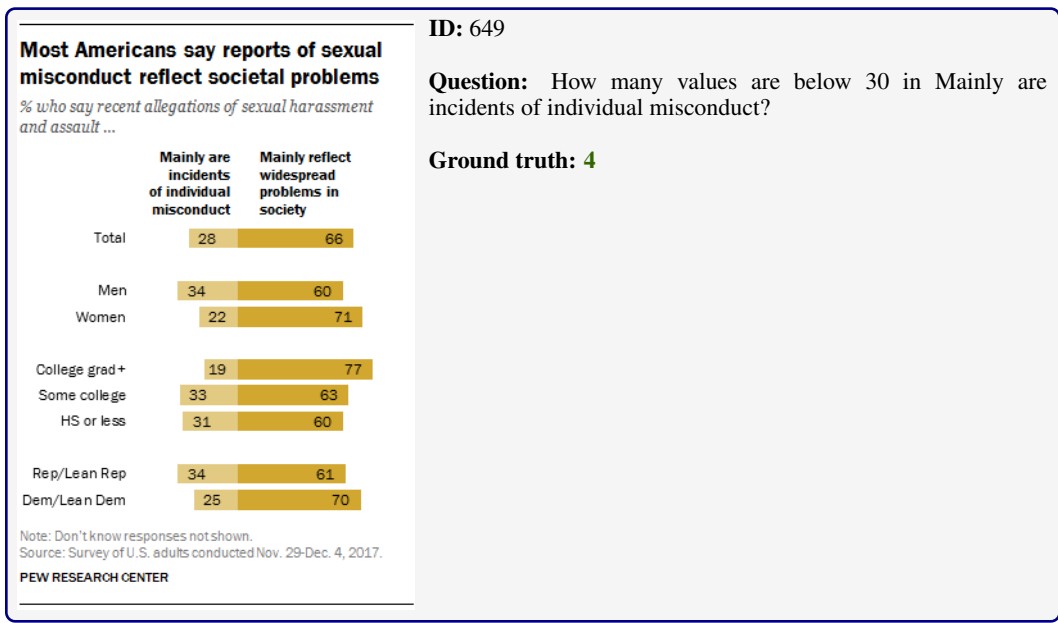

**ID:** 649

**Question:** How many values are below 30 in Mainly are incidents of individual misconduct?

**Ground truth: 4**

---

**LLaMA-Adapter-V2:** There are 2 values below 30 in Mainly are incidents of individual misconduct.

---

**LLaVA:** There are two values below 30 in Mainly are incidents of individual misconduct.

---

**Multimodal Bard:**
There is only 1 value below 30 in the "Mainly are incidents of individual misconduct" column. This value is 22, which is the percentage of women who believe that recent allegations of sexual harassment and assault are mainly incidents of individual misconduct.

The other values in this column are 34, 31, 33, and 25. All of these values are greater than 30.

So the answer to your question is 1.

---

**GPT-4V:**
Based on the "Mainly are incidents of individual misconduct" column, the following values are below 30:

- 28 (Total)
- 22 (Women)
- 19 (College grad+)
- 25 (Dem/Lean Dem)

That's a total of 4 values.

So, the answer is 4.
**Correct output: 4**

Figure 75: The question with a richly formatted statistical figure presents challenges for prominent multimodal models such as LLaMA-Adapter-V2, LLaVA, and Multimodal Bard. However, GPT-4V successfully extracts essential information from the figure, producing the correct answer.

| Price | Quantity demanded | Quantity supplied |
|-------|-------------------|-------------------|
| $270  | 20,200            | 10,600            |
| $320  | 20,000            | 12,500            |
| $370  | 19,800            | 14,400            |
| $420  | 19,600            | 16,300            |
| $470  | 19,400            | 18,200            |

**ID:** 389

**Question:** Look at the table. Then answer the question. At a price of $320, is there a shortage or a surplus?

**Choices:** (A) shortage (B) surplus

**Ground truth: (A) shortage**

---

**LLaMA-Adapter-V2:** The table shows a surplus of 1000 units at a price of $320.

---

**LLaVA:** At a price of $320, there is a surplus of 12,000 units.

---

**Multimodal Bard:**
The correct answer is **(B)**, surplus.

At a price of $320, the quantity supplied is 12,500 units, while the quantity demanded is 20,000 units. This means that there are 7,500 more units supplied than demanded. So the answer is surplus.

Here is a table summarizing the supply and demand at each price:

| Price | Quantity Supplied | Quantity Demanded | Surplus/Shortage |
|-------|-------------------|-------------------|------------------|
| $270  | 10,600            | 20,200            | Shortage of 9,600 |
| $320  | 12,500            | 20,000            | Surplus of 7,500 |
| $370  | 14,400            | 19,800            | Surplus of 5,400 |
| $420  | 19,600            | 19,400            | No surplus or shortage |
| $470  | 18,200            | 16,300            | Shortage of 1,900 |

---

**GPT-4V:**
At a price of $320:
Quantity demanded = 20,000
Quantity supplied = 12,500

Since the quantity demanded (20,000) is greater than the quantity supplied (12,500), there is a shortage.
**Correct output: (A) shortage**

Figure 76: LLaMA-Adapter-V2, LLaVA, and Multimodal Bard struggle to discern the structured information within the table, leading to incorrect answers.

## H.4 ABILITIES ACROSS VISUAL CONTEXTS

This section compares the reasoning abilities of GPT-4V with other large multimodal models (LLMs) on MATHVISTA, considering various types of visual contexts. Models used for comparison include LLaMA-Adapter-V2-7B (simplified as LLaMA-Adapter-V2), LLaVA-LLaMA-2-13B (simplified as LLaVA), and Multimodal Bard.

### H.4.1 ABSTRACT SCENE

Based on Figure 1, current foundation models lag behind human performance in mathematical reasoning in abstract scenes by a substantial margin. Consider the problems in Figures 77 and 78 that are derived from math word problems found in elementary school curricula. Despite their advanced capabilities, foundation models such as Multimodal Bard and GPT-4V fail to produce the correct responses.

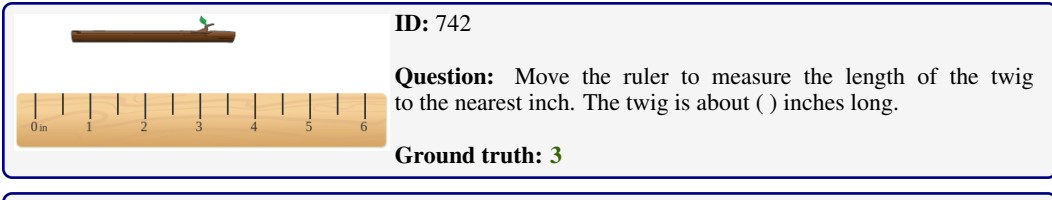

**ID:** 742

**Question:** Move the ruler to measure the length of the twig to the nearest inch. The twig is about ( ) inches long.

**Ground truth:** 3

**LLaMA-Adapter-V2:** The twig is about 2 inches long.

**LLaVA:** The twig is about 2 inches long.

**Multimodal Bard:**
According to the image, the 0 inch mark of the ruler is aligned with the base of the twig. The tip of the twig is just past the 5 inch mark. Therefore, the twig is about **5 inches** long.

**GPT-4V:** The twig is about 4 inches long.

Figure 77: Large multimodal models like GPT-4V struggle to answer this elementary-level question with the abstract scene as the context.

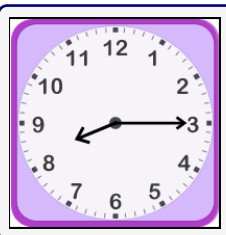

**ID:** 7

**Question:** What time is shown? Answer by typing a time word, not a number. It is ( ) after eight.

**Choices:** (A) half (B) quarter (C) o'clock (D) quarter to (E) quarter past

**Ground truth:** (B) quarter

**LLaMA-Adapter-V2:** The time shown is "half past eight."

**LLaVA:** The correct answer is (C) o'clock.

**Multimodal Bard:**
The image you sent shows a clock with the hands pointing to 8:15. So the time is **quarter past eight**. (E)

The hour hand is between 8 and 9, which means it is past 8 o'clock. The minute hand is pointing to the 3, which is 15 minutes past. Therefore, the time is quarter past eight.

**GPT-4V:**
The minute hand is pointing to the 9, which indicates that it is 45 minutes past the hour. The hour hand is pointing a little past the 8, which means it's after 8 o'clock.

45 minutes past the hour is also described as "quarter to" the next hour.

So, the time shown is "quarter to nine."
The correct answer is (D) quarter to.

Figure 78: Large multimodal models face challenges in answering this elementary-level question which entails interpreting the time from an abstract clock diagram.

### H.4.2 BAR CHART

As shown in Figure 1, foundation models, including GPT-4V, significantly underperform humans in mathematical reasoning when bar charts serve as the visual context. Neither Multimodal Bard nor GPT-4V can solve the problems depicted in Figures 79 and 80, which do not need complex understanding and reasoning.

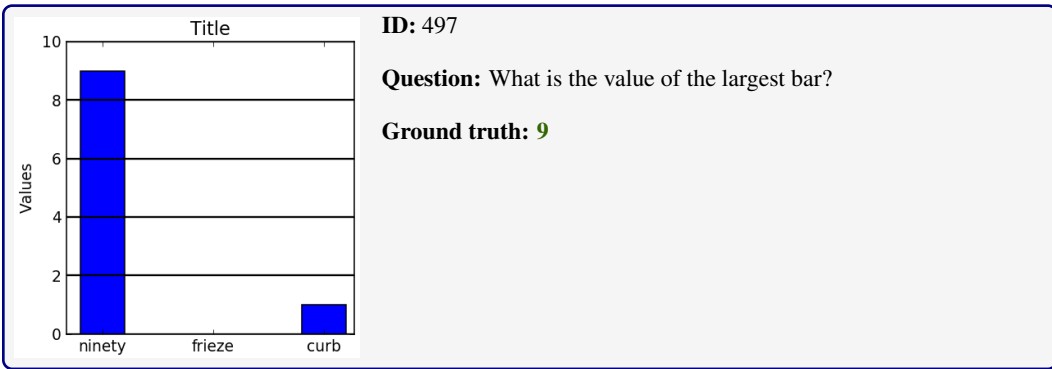

**LLaMA-Adapter-V2:** The value of the largest bar is 10.

**LLaVA:** The value of the largest bar is 90.

**Multimodal Bard:** The largest bar in the image is the one for the title "ninety". It has a value of 10.

**GPT-4V:** The value of the largest bar is 10.

Figure 79: Large Multimodal Models such Multimodal Bard and GPT-4 fail to extract the correct value for the tallest bar in this basic bar chart.

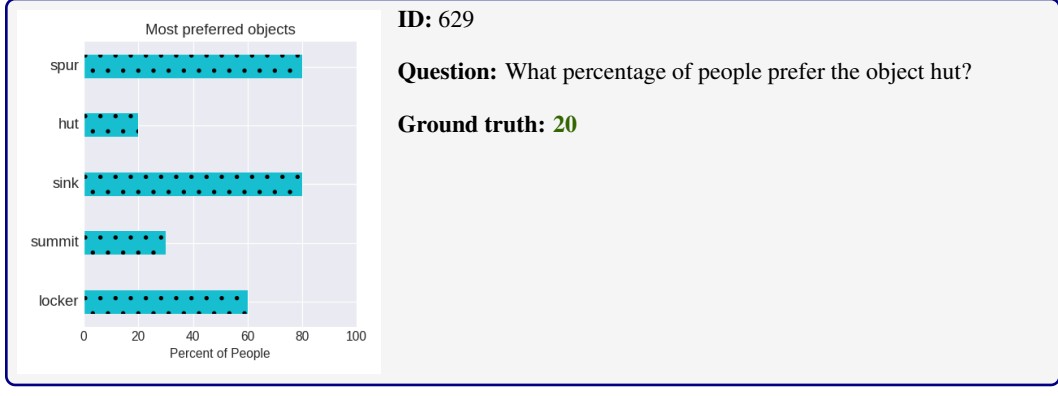

**LLaMA-Adapter-V2:** 10% of people prefer the object hut..

**LLaVA:** Based on the image, 30% of people prefer the object hut..

**Multimodal Bard:** Based on the image, 100% of people prefer the object hut. The line graph shows that the percentage of people who prefer the hut is 100%, which is the maximum value on the graph.

**GPT-4V:** The percentage of people who prefer the object "hut" is approximately 40%.

Figure 80: To address this question, the model simply needs to identify the "hut" and determine its value on the x-axis. However, current foundation models such as GPT-4V struggle to produce the correct answer.

### H.4.3 FUNCTION PLOT

GPT-4V outperforms other baselines on problems related to function plots and even exceeds human performance. Figures 81 and 82 show questions with digital and hand-drawn function plots, respectively. In both cases, GPT-4V accurately identifies their functions and infers the correct properties.

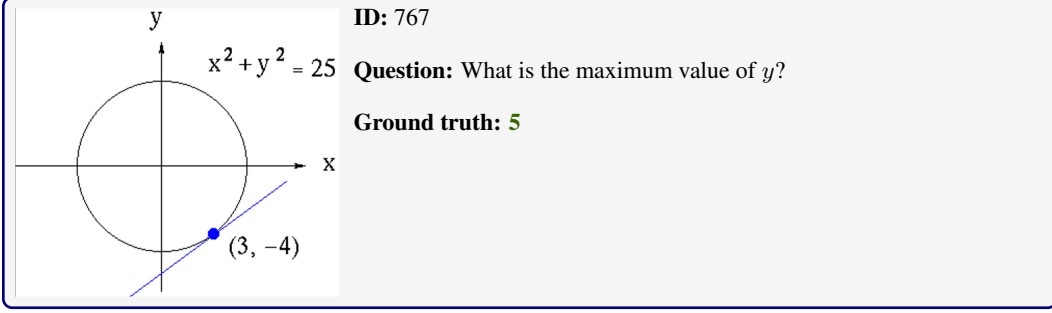

**ID:** 767

**Question:** What is the maximum value of $y$?

**Ground truth: 5**

**LLaMA-Adapter-V2:** The maximum value of $y$ is 3.

**LLaVA:** The maximum value of $y$ is 3.

**Multimodal Bard:**
The maximum value of $y$ is **3**.

The given equation is $y^2 + x^2 = \frac{25}{2}$. This is the equation of a circle with center $(0,0)$ and radius $\frac{5}{\sqrt{2}}$. The point $(3, -4)$ lies on the circle, so the maximum value of $y$ is 3.

**GPT-4V:**
The equation given is for a circle: $x^2 + y^2 = 25$.

The radius of this circle is the square root of 25, which is 5.

The maximum value of $y$ corresponds to the highest point of the circle, which is equal to the radius.

So, the maximum value of $y$ is 5.
**Correct output: 5**

Figure 81: The figure illustrates a function plot accompanied by its equation. GPT-4V correctly identifies the equation, leading to the correct answer.

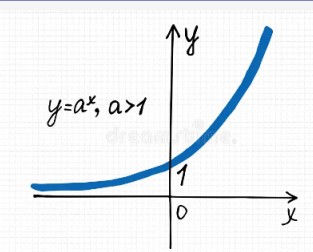

**ID:** 39

**Question:** Is this function odd or even?

**Choices:** (A) odd (B) even

**Ground truth: (A) odd**

---

**LLaMA-Adapter-V2:** The function is odd.
**Correct output: (A) odd**

---

**LLaVA:** The function in the image is odd.
**Correct output: (A) odd**

---

**Multimodal Bard:**
The function in the image is **odd**.

An even function is a function where $f(-x) = f(x)$. An odd function is a function where $f(-x) = -f(x)$.

In the case of the function $y = a_2x - a_1$, if we substitute $x = -x$, we get:

$$y = a_2(-x) - a_1$$
$$= -a_2x + a_1$$

This is not equal to $f(x)$, so the function is **odd**.

The correct choice is (A).
**Correct output: (A) odd**

---

**GPT-4V:**
The given function is $y = a^x$ where $a > 1$.

To determine if the function is odd or even, we can use the following definitions:

- A function $f(x)$ is even if $f(-x) = f(x)$ for all $x$ in the domain of $f$.
- A function $f(x)$ is odd if $f(-x) = -f(x)$ for all $x$ in the domain of $f$.

For the given function $y = a^x$, when we plug in $-x$ for $x$:

$$y = a^{-x}$$

This is not equal to $-a^x$ and it's not equal to $a^x$ either.

Thus, the function is neither odd nor even. However, since the given choices only include "odd" and "even", the best answer is neither. But since I must choose one of the provided options (A) odd.
**Correct output: (A) odd**

Figure 82: The figure showcases a hand-drawn function plot. GPT-4V identifies the function's expression and accurately determines its property.

### H.4.4 GEOMETRY DIAGRAM

Geometry diagrams are a distinct type of visual context in MATHVISTA. To answer questions involving these diagrams, a model must comprehend the fine-grained details, including symbols, variables, and relations from the figures. Additionally, it should apply associated theorems before executing calculations to produce final responses. GPT-4V surpasses other models and even humans due to its superior capabilities in geometry recognition and reasoning. In the examples shown in Figures 83 and 84, GPT-4V delivers the correct results through the application of relevant theorems and subsequent calculations.

---

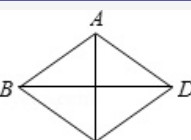

**ID:** 478

**Question:** As shown in the figure, in the diamond $ABCD$, two diagonal lines $AC = 12.0$, $BD = 16.0$, then the edge length of this diamond is ().

**Choices:** (A) 10 (B) 8 (C) 6 (D) 5

**Ground truth:** 10

---

**LLaMA-Adapter-V2:** The edge length of this diamond is 8.

---

**LLaVA:** Based on the given information, the edge length of the diamond ABCD is 8.0.

---

**Multimodal Bard:**
The answer is (A). 10.

In a rhombus, the diagonals bisect each other at right angles. So, we can draw two right triangles within the rhombus, each with a leg of length 6 and a hypotenuse of length 10.

By the Pythagorean Theorem, the length of each side of the rhombus is $\sqrt{6^2 + 6^2} = \sqrt{72} = 10$.

So the answer is (A).
**Correct output:** 10

---

**GPT-4V:**
Given that $AC = 12.0$ and $BD = 16.0$ in the rhombus (diamond) $ABCD$.

Since all sides of a rhombus are congruent, the diagonals of a rhombus are perpendicular bisectors of each other. This means that each half of the diagonal $BD$ is 8.0 and each half of diagonal $AC$ is 6.0.

Using the Pythagorean theorem for one of the right triangles (formed by the diagonals):

$$\text{Side}^2 = (\text{Half of } BD)^2 + (\text{Half of } AC)^2$$
$$\text{Side}^2 = 8^2 + 6^2$$
$$\text{Side}^2 = 64 + 36$$
$$\text{Side}^2 = 100$$
$$\text{Side} = 10$$

**Correct output:** 10

---

Figure 83: For this geometry problem, GPT-4V correctly applies the appropriate theorem and arrives at the right answer. In contrast, Multimodal Bard commits calculation errors during its reasoning process.

**ID:** 410

**Question:** Find the perimeter of $\triangle DEF$, if $\triangle DEF \sim \triangle CBF$, perimeter of $\triangle CBF = 27, DF = 6$, and $FC = 8$.

**Choices:** (A) 20.25 (B) 21 (C) 27 (D) 36

**Ground truth:** (A) 20.25

**LLaMA-Adapter-V2:** The perimeter of $\triangle DEF$ is 20.25, as shown in the image.
**Correct output:** 20.25

**LLaVA:**
The perimeter of $\triangle DEF$ can be calculated using the similarity theorem, which states that if two triangles are similar, their corresponding sides are proportional. In this case, we know that $\triangle DEF \sim \triangle CBF$ and the perimeter of $\triangle CBF$ is 27.

Since the ratio of the corresponding sides of similar triangles is the same, we can use the perimeter of $\triangle CBF$ to find the perimeter of $\triangle DEF$. The perimeter of $\triangle DEF$ is proportional to the perimeter of $\triangle CBF$, so we can multiply the perimeter of $\triangle CBF$ by the ratio of the corresponding sides.

The ratio of the corresponding sides of $\triangle DEF$ and $\triangle CBF$ is 6:8, which means that the perimeter of $\triangle DEF$ is $6 \times 8 = 48$.

Therefore, the correct answer is (D) 48.

**Multimodal Bard:**
Since $\triangle DEF \sim \triangle CBF$, we have $\frac{DE}{CB} = \frac{DF}{FC}$. Since $DE = 6$ and $FC = 8$, we have $\frac{DE}{CB} = \frac{6}{8} = \frac{3}{4}$.

The perimeter of $\triangle CBF$ is 27, so $CB + BF = 27$. Substituting $\frac{3}{4}$ for $\frac{CB}{BF}$, we get $\frac{3}{4}BF + BF = 27$. Solving for $BF$, we get $BF = 12$.

The perimeter of $\triangle DEF$ is $DE + EF + DF = 6 + EF + 6 = EF + 12$. Substituting $\frac{3}{4}$ for $\frac{EF}{BF}$, we get $EF + 12 = \frac{4}{3}EF$. Solving for $EF$, we get $EF = 9$.

Therefore, the perimeter of $\triangle DEF$ is $EF + DE + DF = 9 + 6 + 6 = \boxed{21}$.

**GPT-4V:**
Given $\triangle DEF \sim \triangle CBF$ and $DF = 6$ and $FC = 8$, we can find the scale factor of the two similar triangles.

Scale factor = $\frac{DF}{FC} = \frac{6}{8} = \frac{3}{4}$.

Given the perimeter of $\triangle CBF = 27$, we can find the perimeter of $\triangle DEF$ by multiplying by the scale factor.

Perimeter of $\triangle DEF = 27 \times \frac{3}{4} = 20.25$.
**Correct output:** 20.25

Figure 84: For this geometry problem, GPT-4V correctly applies the appropriate theorem and arrives at the right answer. In contrast, Multimodal Bard commits calculation errors during its reasoning process.

### H.4.5 LINE PLOT

As evidenced by Figure 1, current models such as GPT-4V do not perform as well as humans in mathematical reasoning involving line plots. We speculate that the low performance is mainly due to the difficulty in detecting OCR text in the figures and accurately grounding the values, as illustrated by the examples in Figures 85 and 86.

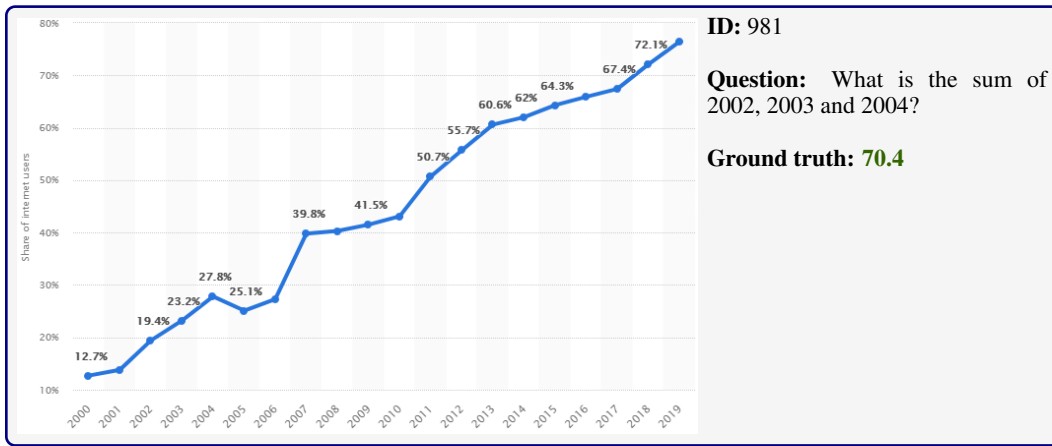

**ID:** 981

**Question:** What is the sum of 2002, 2003 and 2004?

**Ground truth: 70.4**

**LLaMA-Adapter-V2:** The sum of 2002, 2003 and 2004 is 6.6.

**LLaVA:** The sum of 2002, 2003, and 2004 is 6007.

**Multimodal Bard:**
I see the image. The sum of 2002, 2003 and 2004 is 1.40.

The year 2002 has a share of 67.4%, 2003 has a share of 64.3%, and 2004 has a share of 62%. When we add these three values, we get 1.937%. Rounding to one decimal place, the sum is 1.40.

**GPT-4V:**
The sum for the years 2002, 2003, and 2004 from the graph are:

2002: 19.4%
2003: 23.2%
2004: 25.1%

Sum = 19.4 + 23.2 + 25.1 = 67.7%

Figure 85: The figure displays a low-resolution line plot with numerous points. Current foundation models, including GPT-4V, appear to struggle with accurately interpreting the plot.

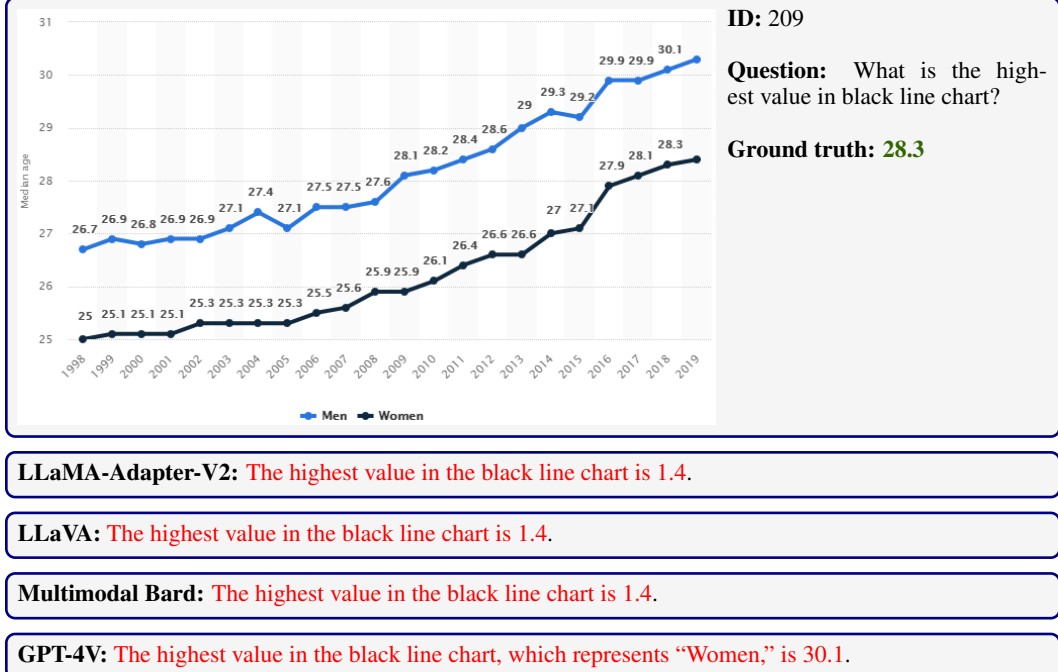

**ID:** 209

**Question:** What is the highest value in black line chart?

**Ground truth: 28.3**

**LLaMA-Adapter-V2:** The highest value in the black line chart is 1.4.

**LLaVA:** The highest value in the black line chart is 1.4.

**Multimodal Bard:** The highest value in the black line chart is 1.4.

**GPT-4V:** The highest value in the black line chart, which represents "Women," is 30.1.

Figure 86: The figure displays two low-resolution line plots comprising numerous points. Current foundation models, including GPT-4V, seem to struggle with interpreting the plots accurately.

### H.4.6 NATURAL IMAGE

MATHVISTA includes questions that require numeric and spatial reasoning based on text and objects in natural images. If models have limited abilities to recognize text (OCR), as shown in Figure 87, or to identify visual objects, as in Figure 88, they are unlikely to generate correct answers to visual questions.

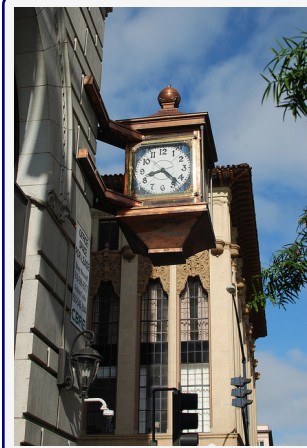

**ID:** 980

**Question:** What is the highest number shown?

**Ground truth:** 12

---

**LLaMA-Adapter-V2:** The highest number shown is 12.
**Correct output:** 12

---

**LLaVA:** The highest number shown on the clock tower is 3.

---

**Multimodal Bard:** I see the image. The highest number shown is 3.

---

**GPT-4V:** The highest number shown on the clock is 12.
**Correct output:** 12

Figure 87: LLaVA and Multimodal Bard are unable to identify the highest number displayed on the clock panel in the image.

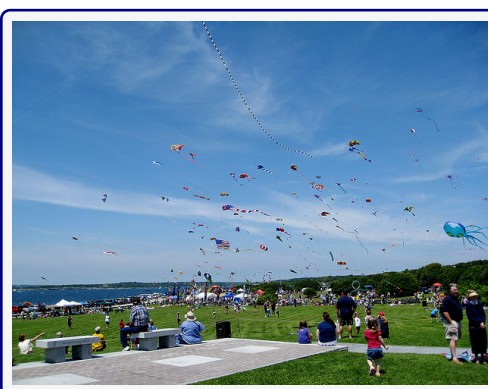

**ID:** 884

**Question:** What is the blue kite in the lower right corner shaped like?

**Choices:**
(A) ferret (B) cat (C) cloud (D) octopus

**Ground truth: (D) octopus**

**LLaMA-Adapter-V2:** The blue kite in the lower right corner is shaped like a cloud.

**LLaVA:** The blue kite in the lower right corner is shaped like a cloud.

**Multimodal Bard:** Sorry, I can't help with images of people yet.

**GPT-4V:** The blue kite in the lower right corner is shaped like (D) octopus.
**Correct output: (D) octopus**

Figure 88: The question demands spatial reasoning ability from the model to pinpoint the object within the image. In this context, only GPT-4V delivers a correct prediction.

### H.4.7 PUZZLE TEST

Math reasoning with puzzle text figures is challenging for current AI foundation models because interpreting these figures requires discerning underlying patterns from sets of shapes, as illustrated in Figure 89, and numbers, as in Figure 90. There is plenty of room for improvement.

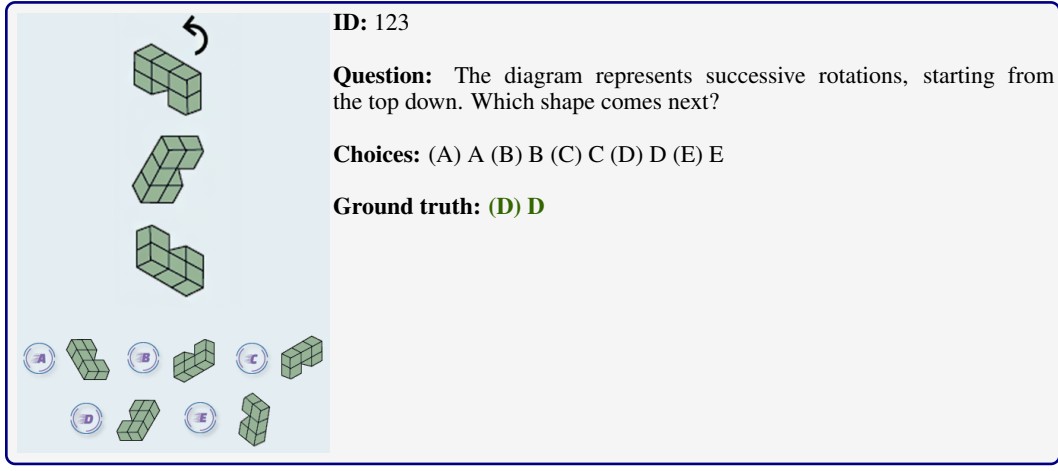

**ID:** 123

**Question:** The diagram represents successive rotations, starting from the top down. Which shape comes next?

**Choices:** (A) A (B) B (C) C (D) D (E) E

**Ground truth:** (D) D

**LLaMA-Adapter-V2:**
The diagram represents successive rotations, starting from the top down. The next shape that comes next is letter D.
**Correct output:** (D) D

**LLaVA:** The correct answer is (E) E.

**Multimodal Bard:**
The answer is (C).

The diagram shows a 3D cube being rotated 90 degrees on each axis. Starting from the top, the first shape is a square. The second shape is a cube with a square missing from the top face. The third shape is a cube with a square missing from the front face. The fourth shape is a cube with a square missing from the left face.

To find the next shape, we need to rotate the cube 90 degrees on the z-axis. This will bring the square that is missing from the left face to the front face. So the next shape is the one shown in option (C).

**GPT-4V:**
The diagram shows a pattern of rotating the shape in a specific sequence. To determine the next shape:

Starting with the first shape, when we rotate it 90 degrees counterclockwise, we get the second shape.

Rotating the second shape 90 degrees counterclockwise gives the third shape.

Thus, rotating the third shape 90 degrees counterclockwise will give us the next shape.

From the choices provided, the shape that matches this rotation is (E) E.

Figure 89: This puzzle test figure demands intricate skills in abstract thinking and spatial reasoning, which prove challenging for current foundation models. Although GPT-4V correctly discerns the transformation pattern of the shapes, it errs in spatially distinguishing between shapes (D) and (E).

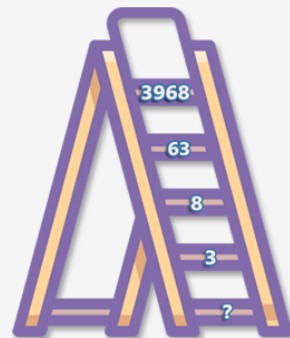

**ID:** 941

**Question:** Which is the next number in the series?

**Ground truth: 2**

**Solution:**
$(3968 + 1) = 63^2$
$(63 + 1) = 8^2$
$(8 + 1) = 3^2$
$(3 + 1) = 2^2$

---

**LLaMA-Adapter-V2:** The next number in the series is 3.

---

**LLaVA:** The next number in the series is 4.

---

**Multimodal Bard:**

The next number in the series is 9.

The series is based on the square root of the number plus 1. So, the first number is 3968, and its square root is 63. Then, 63 + 1 = 64, and its square root is 8. Continuing the pattern, 8 + 1 = 9, and its square root is 3.

Therefore, the next number in the series is 9.

---

**GPT-4V:**

The pattern in the ladder involves dividing the number by the step number to get the next number.

Step 1:   $3968 \div 1 = 3968$
Step 2:   $3968 \div 2 = 1984$  but we have 63, which doesn't follow this pattern.
Step 3:   $63 \div 3 = 21$  but we have 8.
Step 4:   $8 \div 4 = 2$  but we have 3.
Step 5:   $3 \div 5 = 0.6$

The given sequence doesn't seem to have a consistent pattern based on the values presented.

However, if we assume the ladder sequence is based on dividing by a consistent factor (rather than step number), we can attempt to find a common factor:

$$3968 \div 63 \approx 63$$
$$63 \div 8 = 7.875 \text{ which isn't a round number.}$$
$$8 \div 3 = 2.6666 \text{ again not a round number.}$$

Without a clear and consistent pattern, it's challenging to determine the next number in the sequence.

---

Figure 90: The question demands advanced abstract thinking to discern the consistent pattern within a sequence of four numbers. Despite their capabilities, current foundation models including Multimodal Bard and GPT-4V show limited abilities of logical reasoning.

### H.4.8  SCATTER PLOT

A scatter plot is a graphical representation of data points on a two-dimensional axis, where each point represents a value from the dataset. MATHVISTA includes the reasoning task that requires comprehending scatter plots taken from daily life and academic papers, as shown in Figures 91 and 92. Although GPT-4V outperforms other LMMs, such as Multimodal Bard, and even humans in overall accuracy (Figure 1), it often fails in the cases where fine-grained understanding is required, as in Figure 92.

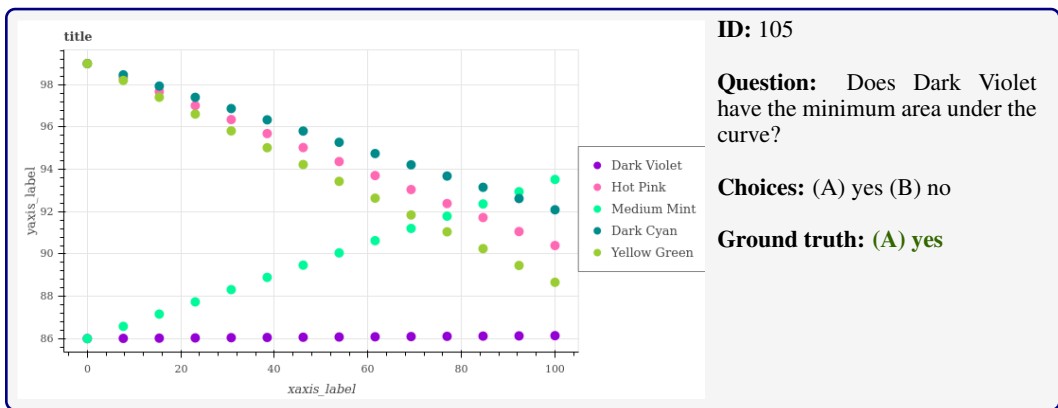

**ID:** 105

**Question:** Does Dark Violet have the minimum area under the curve?

**Choices:** (A) yes (B) no

**Ground truth:** (A) yes

**LLaMA-Adapter-V2:** No, Dark Violet does not have the minimum area under the curve.

**LLaVA:** Based on the image, Dark Violet does not have the minimum area under the curve. The area under the curve is the highest for the color Green.

**Multimodal Bard:**
The correct answer is (B), no. Dark Violet does not have the minimum area under the curve. The color with the minimum area under the curve is Yellow Green.

The area under the curve is calculated by summing the areas of all the rectangles under the curve. The rectangle for Dark Violet has a height of 92 and a width of 20, so its area is 1840. The rectangle for Yellow Green has a height of 88 and a width of 40, so its area is 3520. Therefore, the area under the curve for Yellow Green is larger than the area under the curve for Dark Violet.

**GPT-4V:**
Based on the scatter plot provided, "area under the curve" is not represented since there is no curve shown. However, if by "area under the curve" we mean the overall distribution and position of the points for "Dark Violet" compared to others, we can analyze the positions of the points.

From the scatter plot, the color "Dark Violet" consistently appears to be the lowest on the y-axis across the entire range of the x-axis.

Given the choices: (A) yes (B) no

The correct answer is: (A) yes.
**Correct output:** (A) yes

Figure 91: The scatter plot presented in this problem is template-generated. While models such as LLaMA-Adapter-V2, LLaVA, and Multimodal Bard encounter difficulties in discerning quantitative relationships between different plot lines, GPT-4V successfully discerns the correct relationships and provides an accurate answer.

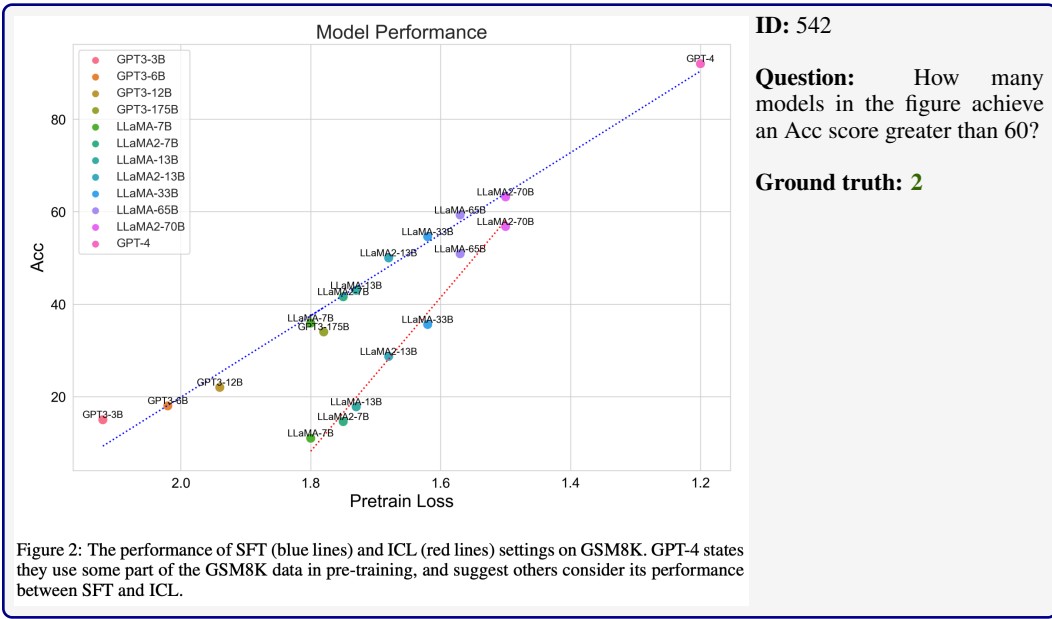

**ID:** 542

**Question:** How many models in the figure achieve an Acc score greater than 60?

**Ground truth: 2**

Figure 2: The performance of SFT (blue lines) and ICL (red lines) settings on GSM8K. GPT-4 states they use some part of the GSM8K data in pre-training, and suggest others consider its performance between SFT and ICL.

**LLaMA-Adapter-V2:** There are 10 models in the figure that achieve an Acc score greater than 60..

**LLaVA:** There are two models in the figure that achieve an Acc score greater than 60.
**Correct output: 2**

**Multimodal Bard:**
There are 8 models in the figure that achieve an Acc score greater than 60. These models are:

* GPT-4
* GPT3-175B
* LLAMA-7B
* LLAMA2-7B
* LLAMA-13B
* LLAMA2-13B
* LLAMA-33B
* LLAMA-65B
* LLAMA2-70B

The Acc score is the accuracy of the model, which is the percentage of test cases that the model correctly classifies. An Acc score of 60 means that the model correctly classifies 60% of the test cases.

**GPT-4V:** There are 6 models in the figure that achieve an Acc score greater than 60.

Figure 92: The example showcases a scatter plot sourced from an academic paper. Current models, including GPT-4V, struggle to discern the fine-grained relationships depicted in the figure.

### H.4.9  SCIENTIFIC SCENE

Answering questions based on scientific scenes poses a challenge in aligning the scientific concepts present in the question text and those in the accompanying figures. GPT-4V demonstrates its superior ability to reason about scientific scenes compared to other models, as evidenced in Figure 1. In the example of Figure 93, GPT-4V adeptly identifies two organisms in the food web and elucidates their relationship. In another instance, shown in Figures 94 and 95, both Multimodal Bard and GPT-4V are able to use knowledge in the physical domain to effectively ground the symbols and variables depicted in the figure.

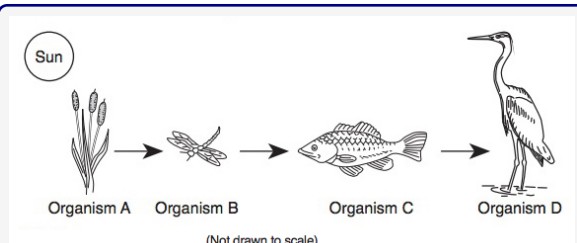

**ID:** 685

**Question:** What would happen to organism c if organism b increased?

**Choices:** (A) decrease (B) increase (C) can't predict (D) stay same

**Ground truth: (B) increase**

---

**LLaMA-Adapter-V2:**
If organism b increases, organism c would decrease. This is because organism c is dependent on organism b for its food supply, and an increase in organism b would lead to a decrease in organism c.

---

**LLaVA:**
If organism b increased, organism c would likely decrease. This is because the growth of organism b would lead to increased competition for resources, such as food and space. As a result, organism c would have to compete more fiercely for these resources, which could lead to a decrease in its population.

---

**Multimodal Bard:**
The answer is (A). If organism B increases, then organism C will **decrease**.

In the food chain, organism B is the predator of organism C. This means that organism B eats organism C. If there are more organism B, then there will be more predation on organism C. This will lead to a decrease in the population of organism C.

So the answer is (A).

---

**GPT-4V:**
If Organism B (which appears to be some kind of insect) increased in population, it would mean that there is more available food for Organism C (the fish).

Therefore, the population of Organism C would likely: (B) increase.
**Correct output: (B) increase**

Figure 93: Models such as LLaMA-Adapter-V2, LLaVA, and Multimodal Bard seem to struggle with discerning the relationships between organisms depicted in the food chain. Instead, GPT-4V recognizes organisms in the food chain and possesses the knowledge to infer their relations.

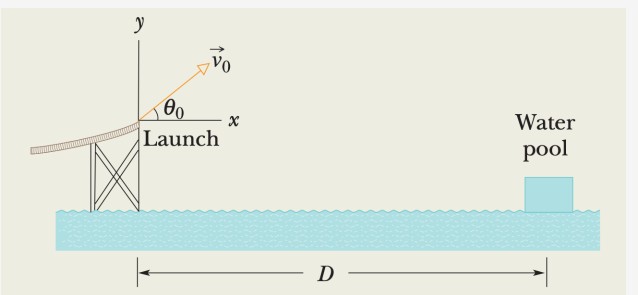

**ID:** 862

**Question:** One of the most dramatic videos on the web (but entirely fictitious) supposedly shows a man sliding along a long water slide and then being launched into the air to land in a water pool. Let's attach some reasonable numbers to such a flight to calculate the velocity with which the man would have hit the water. Figure indicates the launch and landing sites and includes a superimposed coordinate system with its origin conveniently located at the launch site. From the video we take the horizontal flight distance as $D = 20.0$ m, the flight time as $t = 2.50$ s, and the launch angle as $\theta_0 = 40.0°$. Find the magnitude of the velocity at launch and at landing.

**Ground truth:** **10.44**

---

**Multimodal Bard:**
Given the following information:

* Horizontal flight distance, $D = 20.0$ m
* Flight time, $t = 2.50$ s
* Launch angle, $\theta_0 = 40.0°$

We can find the magnitude of the velocity at launch using the following equation:

$$v_0 = \sqrt{v_{0x}^2 + v_{0y}^2}$$

where $v_{0x}$ is the horizontal component of the launch velocity and $v_{0y}$ is the vertical component of the launch velocity.

The horizontal component of the launch velocity can be found using the following equation:

$$v_{0x} = \frac{D}{t} = \frac{20.0 \text{ m}}{2.50 \text{ s}} = 8.00 \text{ m/s}$$

The vertical component of the launch velocity can be found using the following equation:

$$v_{0y} = v_0 \sin \theta_0 = 8.00 \text{ m/s} \sin 40.0° = 5.83 \text{ m/s}$$

Therefore, the magnitude of the velocity at launch is:

$$v_0 = \sqrt{v_{0x}^2 + v_{0y}^2} = \sqrt{(8.00 \text{ m/s})^2 + (5.83 \text{ m/s})^2} = 10.44 \text{ m/s} \approx 10.4 \text{ m/s}$$

The magnitude of the velocity at landing will be the same as the magnitude of the velocity at launch, since there is no air resistance in this case.

So the answer is 10.4 rounded to two decimal places.

Figure 94: This college-level question, derived from the physics curriculum, requires understanding the intricate scenario depicted in the scientific figure, possessing domain-specific knowledge, and performing multiple reasoning steps to obtain a conclusive answer. Multimodal Bard correctly processes the reasoning but failed to present the answer in the appropriate format.

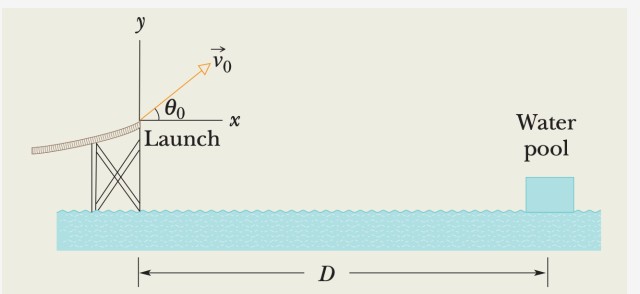

**ID:** 862

**Question:** One of the most dramatic videos on the web (but entirely fictitious) supposedly shows a man sliding along a long water slide and then being launched into the air to land in a water pool. Let's attach some reasonable numbers to such a flight to calculate the velocity with which the man would have hit the water. Figure indicates the launch and landing sites and includes a superimposed coordinate system with its origin conveniently located at the launch site. From the video we take the horizontal flight distance as $D = 20.0$ m, the flight time as $t = 2.50$ s, and the launch angle as $\theta_0 = 40.0°$. Find the magnitude of the velocity at launch and at landing.

**Ground truth: 10.44**

**GPT-4V:**
Given:
- $D = 20.0$ m
- $t = 2.50$ s
- $\theta_0 = 40.0°$

Using kinematic equations, the horizontal motion is described as:

$$D = V_{0x} \times t$$

Where $V_{0x}$ is the horizontal component of the initial velocity.

$$V_{0x} = V_0 \times \cos(\theta_0)$$

From the first equation:

$$V_0 \times \cos(\theta_0) = \frac{D}{t}$$

Rearranging for $V_0$:

$$V_0 = \frac{D}{t \times \cos(\theta_0)}$$

Substitute the given values to get $V_0$:

$$V_0 = \frac{20.0 \text{ m}}{2.50 \text{ s} \times \cos(40.0°)}$$

Using a calculator:

$$V_0 = \frac{20.0}{2.50 \times 0.766}$$
$$V_0 = \frac{20.0}{1.915}$$
$$V_0 = 10.44 \text{ m/s}$$

Thus, the magnitude of the launch velocity (which will be the same as the landing velocity, ignoring air resistance) is $10.44$ m/s.
**Correct output: 10.44**

Figure 95: Continuing from Figure 94, GPT-4V generates the correct answer and the associated reasoning steps for this scientific problem.

### H.4.10   Synthetic Scene

Problems involving synthetic scenes require a nuanced understanding of visual objects, such as the numbers, attributes, and positions of these objects, as shown in Figures 96 and 97. Although GPT-4V demonstrates notable advancements over other models, such as Multimodal Bard, it still falls significantly short of human performance, as shown in Figure 1.

---

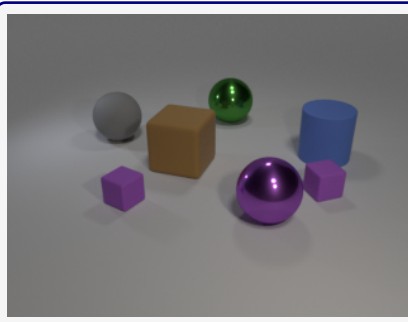

**ID:** 52

**Question:** Subtract all big red metallic spheres. Subtract all big brown matte things. How many objects are left?

**Ground truth: 6**

---

**LLaMA-Adapter-V2:**
The final value is 1, as there is only one object left after subtracting all big red metallic spheres and all big brown matte things.

---

**LLaVA:**
After subtracting all big red metallic spheres and big brown matte things, there are 3 objects left.

---

**Multimodal Bard:**
Given the image, there are 4 objects:

* 1 big red metallic sphere
* 1 big brown matte cube
* 1 small red metallic sphere
* 1 small brown matte sphere

If we subtract all big red metallic spheres, there will be 3 objects left:

* 1 big brown matte cube
* 1 small red metallic sphere
* 1 small brown matte sphere

If we subtract all big brown matte things, there will be 2 objects left:
* 1 small red metallic sphere
* 1 small brown matte sphere

Therefore, if we subtract all big red metallic spheres and big brown matte things, there will be **2 objects left**.

---

**GPT-4V:**
There are no big red metallic spheres in the image. There is 1 big brown matte object (the cube).

Total objects to be subtracted = 0 + 1 = 1

Total objects in the image = 7

Objects left = 7 - 1 = 6

There are 6 objects left.
**Correct output: 6**

---

Figure 96: In this example, GPT-4V is able to recognize the objects with specific attributes from the synthetic figure and then perform arithmetic reasoning to get the correct answer.

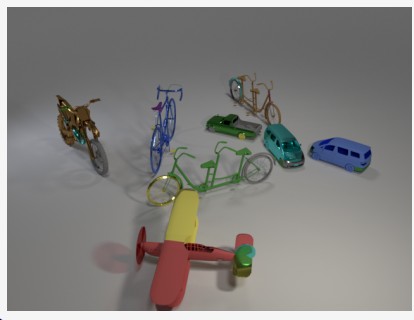

**ID:** 177

**Question:** Is the number of brown tandem bikes that are to the left of the small blue matte car greater than the number of tiny blue biplanes?

**Choices:** (A) Yes (B) No

**Ground truth:** **(A) Yes**

**LLaMA-Adapter-V2:**
Yes, the number of brown tandem bikes that are to the left of the small blue matte car is greater than the number of tiny blue biplanes.
**Correct output:** **(A) Yes**

**LLaVA:**
The correct answer is (B) No. There are more tiny blue biplanes to the left of the small blue matte car than brown tandem bikes.

**Multimodal Bard:**
The answer is (B). There are two brown tandem bikes to the left of the small blue matte car, and there is only one tiny blue biplane. So the number of brown tandem bikes is not greater than the number of tiny blue biplanes.

**GPT-4V:**
Looking at the image, there is 1 brown tandem bike to the left of the small blue matte car. There are no tiny blue biplanes in the image.

Thus, the number of brown tandem bikes that are to the left of the small blue matte car is (A) Yes greater than the number of tiny blue biplanes.
**Correct output:** **(A) Yes**

Figure 97: The question in this example requires the models to accurately extract the attributes of different objects from the synthetic figure. GPT-4V infers the correct answer based on its generated detailed description of the figure.

### H.4.11 TABLE

Tables serve as a powerful tool to present and summarize large amounts of data in a comprehensible manner. In particular, GPT-4V has shown significant advancements over other foundation models and even surpasses human performance on table-related reasoning tasks, as shown in Figure 1. The example in Figure 98 shows a complex table taken from an academic paper. GPT-4V can accurately pinpoint the target cells among numerous rows and columns. Figure 99 shows a QA task in which the answer needs to be derived from the table regarding the push-up competition. GPT-4V is the only model that can produce the correct answer.

Table 3: MM-Vet evaluation results on various LMMs regarding each *core VL capability*. For each column, the highest, the second, and the third highest figures are highlighted by green, orange and blue backgrounds. All the numbers are presented in % and the full score is 100%.

| Model | Rec | OCR | Know | Gen | Spat | Math | Total |
|---|---|---|---|---|---|---|---|
| Transformers Agent (GPT-4) [34] | 18.2 | 3.9 | 2.2 | 3.2 | 12.4 | 4.0 | 13.4±0.5 |
| LLaMA-Adapter v2-7B [28] | 16.8 | 7.8 | 2.5 | 3.0 | 16.6 | 4.4 | 13.6±0.2 |
| OpenFlamingo-9B [2, 6] | 24.6 | 14.4 | 13.0 | 12.3 | 18.0 | 15.0 | 21.8±0.1 |
| MiniGPT-4-8B [84] | 27.4 | 15.0 | 12.8 | 13.9 | 20.3 | 7.7 | 22.1±0.1 |
| BLIP-2-12B [42] | 27.5 | 11.1 | 11.8 | 7.0 | 16.2 | 5.8 | 22.4±0.2 |
| LLaVA-7B [46] | 28.0 | 17.1 | 16.3 | 18.9 | 21.2 | 11.5 | 23.8±0.6 |
| MiniGPT-4-14B [84] | 29.9 | 16.1 | 20.4 | 22.1 | 22.2 | 3.8 | 24.4±0.4 |
| Otter-9B [40] | 28.4 | 16.4 | 19.4 | 20.7 | 19.3 | 15.0 | 24.6±0.2 |
| InstructBLIP-14B [19] | 30.8 | 16.0 | 9.8 | 9.0 | 21.1 | 10.5 | 25.6±0.3 |
| InstructBLIP-8B [19] | 32.4 | 14.6 | 16.5 | 18.2 | 18.6 | 7.7 | 26.2±0.2 |
| LLaVA-13B [46] | 30.9 | 20.1 | 23.5 | 26.4 | 24.3 | 7.7 | 26.4±0.1 |
| MM-ReAct-GPT-3.5 [76] | 24.2 | 31.5 | 21.5 | 20.7 | 32.3 | 26.2 | 27.9±0.1 |
| LLaVA-7B (LLaMA-2) [46] | 32.9 | 20.1 | 19.0 | 20.1 | 25.7 | 5.2 | 28.1±0.4 |
| LLaVA-13B (V1.3, 336px) [46] | 38.1 | 22.3 | 25.2 | 25.8 | 31.3 | 11.2 | 32.5±0.1 |
| LLaVA-13B (LLaMA-2) [46] | 39.2 | 22.7 | 26.5 | 29.3 | 29.6 | 7.7 | 32.9±0.1 |
| MM-ReAct-GPT-4 [76] | 33.1 | 65.7 | 29.0 | 35.0 | 56.8 | 69.2 | 44.6±0.2 |

**ID:** 923

**Question:** Which model achieves the highest score in terms of Rec?

**Choices:**
(A) Transformers Agent (GPT-4)
(B) LLaMA-Adapter v2-7B
(C) LLaVA-7B
(D) Otter-9B
(E) MM-ReAct-GPT-3.5
(F) LLaVA-13B (LLaMA-2)
(G) MM-ReAct-GPT-4

**Ground truth:**
(F) LLaVA-13B (LLaMA-2)

**LLaMA-Adapter-V2:** The correct answer is (G) MM-ReAct-GPT-4.

**LLaVA:**
The correct answer is (B) LLaMA-Adapter v2-7B, which achieves the highest score in terms of Rec.

**Multimodal Bard:**
The answer is **(G) MM-ReAct-GPT-4**.

MM-ReAct-GPT-4 achieves the highest score of **33.1** in terms of Rec, which is the ability to recognize and understand text. The other models have the following scores in terms of Rec:

* Transformers Agent (GPT-4): 18.2
* LLaMA-Adapter v2-7B: 16.8
* LLaVA-7B: 28.0
* Otter-9B: 28.4
* MM-ReAct-GPT-3.5: 24.2
* LLaVA-13B (LLaMA-2): 32.9

**GPT-4V:**
The model that achieves the highest score in terms of Rec is "LLaVA-13B (LLaMA-2)" with a score of 39.2%. The correct option is: (F) LLaVA-13B (LLaMA-2)
**Correct output:** (F) LLaVA-13B (LLaMA-2)

Figure 98: The problem presents a complex table from an academic paper, complete with a caption and multiple rows and columns. This format poses challenges to models like LLaMA-Adapter-V2, LLaVA, and Multimodal Bard. However, GPT-4V demonstrates its exceptional reasoning abilities within tabular contexts.

| Push-up competition (number of push-ups) | | **ID:** 265 |
|---|---|---|

| Stem | Leaf |
|---|---|
| 4 | 8 |
| 5 | 5 6 |
| 6 | 3 6 7 7 |
| 7 | 1 3 3 5 7 7 8 |

**Question:** Justine's P.E. class participated in a push-up competition, and Justine wrote down how many push-ups each person could do. How many people did at least 60 push-ups? (Unit: people)

**Ground truth: 11**

---

**LLaMA-Adapter-V2:**
Justine's P.E. class participated in a push-up competition, and Justine wrote down how many push-ups each person could do. There were 8 people in the competition, and at least 60 push-ups were done by 3 people. The final value is 3.

---

**LLaVA:**
In the image, there are a total of 13 people participating in the push-up competition. Among them, there are two people who did at least 60 push-ups.

---

**Multimodal Bard:**
There are 4 people who did at least 60 push-ups.

The person with the highest number of push-ups did 778, so the only people who did at least 60 push-ups are the ones who did 60, 77, 78, and 778.

There is one person who did 60 push-ups, one person who did 77 push-ups, one person who did 78 push-ups, and one person who did 778 push-ups. So there are a total of 1 + 1 + 1 + 1 = 4 people who did at least 60 push-ups.

---

**GPT-4V:**
For the stem-and-leaf plot provided:

6 represents the tens value, and the digits 3, 6, 7, and 7 represent the ones value. So, the number of push-ups represented by the 6 row are 63, 66, 67, and 67.

7 represents the tens value, and the digits 1, 3, 3, 5, 7, 7, and 8 represent the ones value. So, the number of push-ups represented by the 7 row are 71, 73, 73, 75, 77, 77, and 78.

Counting these numbers: 4 from the 6 row + 7 from the 7 row = 11.
**Correct output: 11**

Figure 99: This figure features a stem-and-leaf plot used for statistical analysis. While LLaMA-Adapter-V2, LLaVA, and Multimodal Bard struggle to comprehend this domain-specific table, GPT-4V excels in understanding its content.

### H.4.12 OTHER VISUAL CONTEXTS

On the reasoning tasks using *other* visual contexts, GPT-4V achieves a higher overall accuracy than all the other models, as depicted in Figure 1. For instance, GPT-4V is the only model that is capable of generating the correct answer to the question regarding a violin plot, as shown in Figure 100.

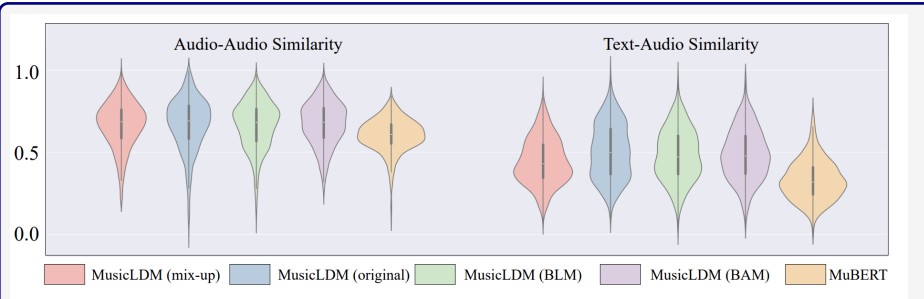

Figure 3: The violin plot of the audio-audio similarity, and the text-to-audio similarity.

**ID:** 895

**Question:** Which model has the lowest Audio-Audio Similarity and Text-Audio Similarity scores overall?

**Choices:**
(A) MusicLDM (mix-up)
(B) MusicLDM (original)
(C) MusicLDM (BLM)
(D) MusicLDM (BAM)
(E) MuBERT

---

**LLaMA-Adapter-V2:**
The model with the lowest Audio-Audio Similarity and Text-Audio Similarity scores overall is MusicLDM (BLM).

---

**LLaVA:**
Based on the image, the model with the lowest Audio-Audio Similarity and Text-Audio Similarity scores overall is MusicLDM (mix-up).

---

**Multimodal Bard:**
The answer is (C). MusicLDM (BLM) has the lowest Audio-Audio Similarity and Text-Audio Similarity scores overall.

The violin plot shows that the distribution of audio-audio similarity scores for MusicLDM (BLM) is the most spread out, indicating that there is a lot of variation in the scores. The distribution of text-audio similarity scores for MusicLDM (BLM) is also the most spread out. This suggests that MusicLDM (BLM) is not very consistent in its ability to capture the similarity between audio and text.

---

**GPT-4V:**
The model with the lowest Audio-Audio Similarity and Text-Audio Similarity scores overall is MuBERT.

**Correct output:** (E) MuBERT

---

Figure 100: Violin plots are widely used in academic papers to visualize the distribution of data and probability density. In this example, only GPT-4V is able to generate the correct output from the figure.

### H.5 SELF-VERIFICATION IN GPT-4V

Self-verification is a social psychological theory asserting that people desire others to perceive them as they see themselves. Consequently, individuals will take active measures to ensure that others view them in ways that confirm their stable self-concepts (Talaifar & Swann, 2020).

Interestingly, in our experiments, GPT-4V demonstrates an ability similar to *self-verification*. The model can inspect its own behaviors during the course of reasoning and can take active actions to correct its mistakes. Note that self-verification we discuss here differs from several recent works on improving the model's outputs based on external feedback (Peng et al., 2023) or additional generations (Yang et al., 2023b). The examples in Figures 101 and 103 show that GPT-4V, on its own, can inspect a set of candidate answers and identify the one that is valid and meets all the given constraints. The multi-step reasoning example in Figure 102 shows that GPT-4V can verify the validity of (the result of) each reasoning step, and explore alternative approaches if any invalid (intermediate) result is detected (*e.g.,* a negative value for length).

Although self-verification does not guarantee an accurate response even after multiple tries, especially when applying GPT-4V to visual perception or mathematical reasoning in intricate scenarios (see Figure 104), it is instrumental in improving the model performance on MATHVISTA. We also found that GPT-4V's self-verification is weaker for non-English tasks, such as Mandarin, as shown in Figure 105. It is also worth noting that self-verification does not emerge in other foundation models we studied, or at least it is not as robust as that of GPT-4V. As shown in Figure 106, Multimodal Bard first attempts a natural language solution, followed by a program-assisted one for verification. However, the program-aided solution leads to a different and incorrect prediction.

The emergent ability of self-verification highlights GPT-4V's potential in solving rigorous reasoning and theorem-proving tasks. One of the most exciting research topics for future work is to develop a mechanism that allows the model to activate self-verification consistently at the right time and to use a set of alternative approaches that maximize the success rate of task completion.

---

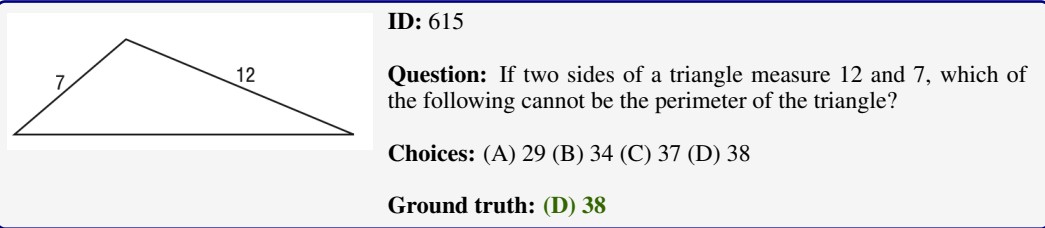

**ID:** 615

**Question:** If two sides of a triangle measure 12 and 7, which of the following cannot be the perimeter of the triangle?

**Choices:** (A) 29 (B) 34 (C) 37 (D) 38

**Ground truth:** **(D) 38**

---

**GPT-4V:**
The sum of the lengths of any two sides of a triangle must be greater than the length of the third side. Let $x$ be the length of the third side.

From the triangle inequality:
$$12 + 7 > x => x < 19$$

Similarly:
$$7 + x > 12 => x > 5$$

And:
$$12 + x > 7 => x > -5 \text{ (This will always be true since the length cannot be negative)}$$

So,
$$5 < x < 19$$

Now, let's check the given options:

(A) $29 => x = 29 - 12 - 7 = 10$ (Possible)
(B) $34 => x = 34 - 12 - 7 = 15$ (Possible)
(C) $37 => x = 37 - 12 - 7 = 18$ (Possible)
(D) $38 => x = 38 - 12 - 7 = 19$ (Not Possible)
**Correct output:** **(D) 38**

---

Figure 101: Given several possible values for the length of the third side, GPT-4V verifies each candidate answer to ensure that it meets the given constraints (highlighted in blue).

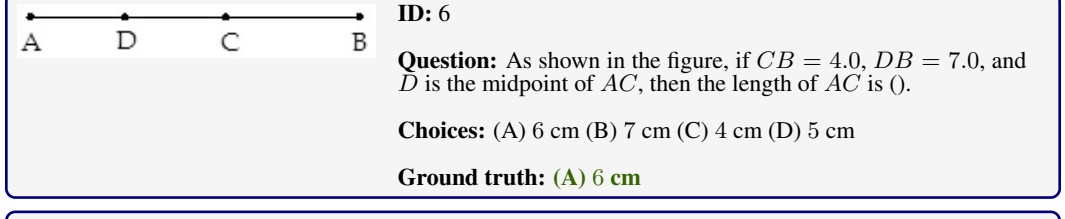

**ID:** 6

**Question:** As shown in the figure, if $CB = 4.0$, $DB = 7.0$, and $D$ is the midpoint of $AC$, then the length of $AC$ is ().

**Choices:** (A) 6 cm (B) 7 cm (C) 4 cm (D) 5 cm

**Ground truth:** (A) 6 cm

**GPT-4V:**
From the figure, we can deduce:

$$AD = DC \text{ (since } D \text{ is the midpoint of } AC)$$

Now, using the entire line segment $AB$:

$$AB = AD + DB$$
$$AB = DC + DB$$

Given $DB = 7.0$ cm, we can rearrange to find:

$$AD \text{ (or } DC) = AB - DB$$
$$AD = AB - 7.0$$

Now, let's find $AB$:
From the given information,
$$AB = AC + CB$$
$$AB = AC + 4.0$$

Using the previously deduced equation:

$$AD = AC + 4.0 - 7.0$$
$$AD = AC - 3.0$$

Since $D$ is the midpoint of $AC$, $AD$ will be half the length of AC:

$$AC = 2(AD)$$

From the above:

$$\textcolor{red}{AC - 3.0 = 2(AD)}$$

Given that $AD$ and $DC$ are equal, we can represent $DC$ as:

$$DC = AC/2$$

Using the relation $CB = 4.0$ cm and $DB = 7.0$ cm:

$$\textcolor{red}{AC/2 + 7.0 = 4.0}$$

$$\textcolor{red}{AC/2 = -3.0}$$

$\textcolor{blue}{\text{This is not possible as lengths cannot be negative. Let's re-evaluate our approach.}}$

From the original figure:
$$AB = AD + DB$$
$$AB = DC + DB$$

Given $DB = 7.0$ cm, we have:
$$AD = AB - 7.0$$
$$AD = AC/2 - 7.0$$

Using the relation $AB = AC + CB$:

$$AC/2 + 7.0 = AC + 4.0$$

Subtracting $AC/2$ from both sides:
$$7.0 = AC/2 + 4.0$$

Subtracting 4 from both sides:
$$AC/2 = 3.0$$

Multiplying both sides by 2:
$$AC = 6.0 \text{ cm}$$

**Correct output:** (A) 6 cm

Figure 102: GPT-4V generates the result as $AC/2 = -3.0$. It verifies the validity of the value and concludes it cannot be negative. The model then re-evaluates its approach and adopts a new method to obtain the correct answer, highlighted in blue.

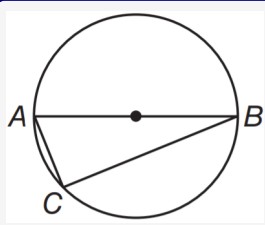

**ID:** 28

**Question:** $\overline{AB}$ is a diameter, $AC = 8$ inches, and $BC = 15$ inches. Find the radius of the circle.

**Choices:** (A) 7.5 (B) 8 (C) 8.5 (D) 17

**Ground truth:** (C) 8.5

**GPT-4V:**
Given that $AB$ is a diameter and $AC = 8$ inches and $BC = 15$ inches, we can use the Pythagorean theorem to determine the length of $AB$.

The right triangle is formed by $A$, the center of the circle, and $C$ makes $AC$ the shorter leg, the radius the longer leg, and $BC$ the hypotenuse.

Let $r$ be the radius of the circle. Using the Pythagorean theorem:

$$AC^2 + r^2 = BC^2$$

Substituting the given lengths:

$$8^2 + r^2 = 15^2$$
$$64 + r^2 = 225$$
$$r^2 = 161$$

Taking the square root of both sides:

$$r = \sqrt{161}$$

However, $r$ is not any of the choices. We need to find the diameter, which is twice the radius.

$$AB = 2r = 2\sqrt{161}$$

But, since the diameter is just twice the radius and the radius is one of the choices, the radius is:

$$r = \sqrt{161}/2 = 8.5 \text{ (approximately)}$$

**Correct output:** (C) 8.5

Figure 103: Starting by computing the initial radius length as $r = \sqrt{161}$, GPT-4V performs two verifications resulting in different answers: $r = 2\sqrt{161}$ and $r = \sqrt{161}/2$. It then selects the matching option from the choices as its final prediction, highlighted in blue.

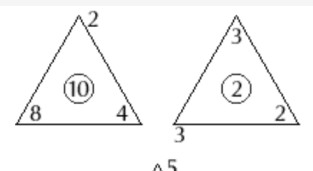

**ID:** 644

**Question:** What is the missing number in the picture?

**Choices:** (A) 6 (B) 8 (C) 10 (D) 11

**Ground truth:** (B) 8

**Solution:** $8 + 4 - 2 = 10, 3 + 2 - 3 = 2, 12 + 1 - 5 = 8$

---

**GPT-4V:**

Let's analyze the pattern in the given triangles:

1. For the first triangle:
$$(8 - 4) \times 2 = 8$$

2. For the second triangle:
$$(3 - 2) \times 3 = 3$$

Applying the same pattern to the third triangle:

$$(12 - 1) \times 5 = 55$$

However, since the answer is in single digits, it seems we take only the unit's place of the result.

So, the missing number is: 5 (unit's place of 55)

This is not in the given options. Therefore, let's explore another possible pattern:

1. For the first triangle:
$$(8 + 4)/2 = 6$$

2. For the second triangle:
$$(3 + 2)/3 = 5/3 = 1.67 \text{ (This does not match the number inside)}$$

But let's continue with this pattern for the third triangle:

$$(12 + 1)/5 = 13/5 = 2.6$$

Again, this does not match any of the options.

Considering other possible operations:

1. For the first triangle:
$$(8 \times 4)/2 = 16$$

2. For the second triangle:
$$(3 \times 2)/3 = 2$$

3. For the third triangle:
$$(12 \times 1)/5 = 12/5 = 2.4$$

Again, this does not match any of the options.

Given the information and the options provided, I am unable to find a consistent pattern that applies to all the triangles.

Therefore, I cannot determine the missing number from the given choices.

---

Figure 104: In this logical reasoning question, GPT-4V verifies its generated solution and retries twice. However, it still fails to produce an answer that matches any of the options. Consequently, it claims that it cannot determine the missing number from the given choices (highlighted in blue).

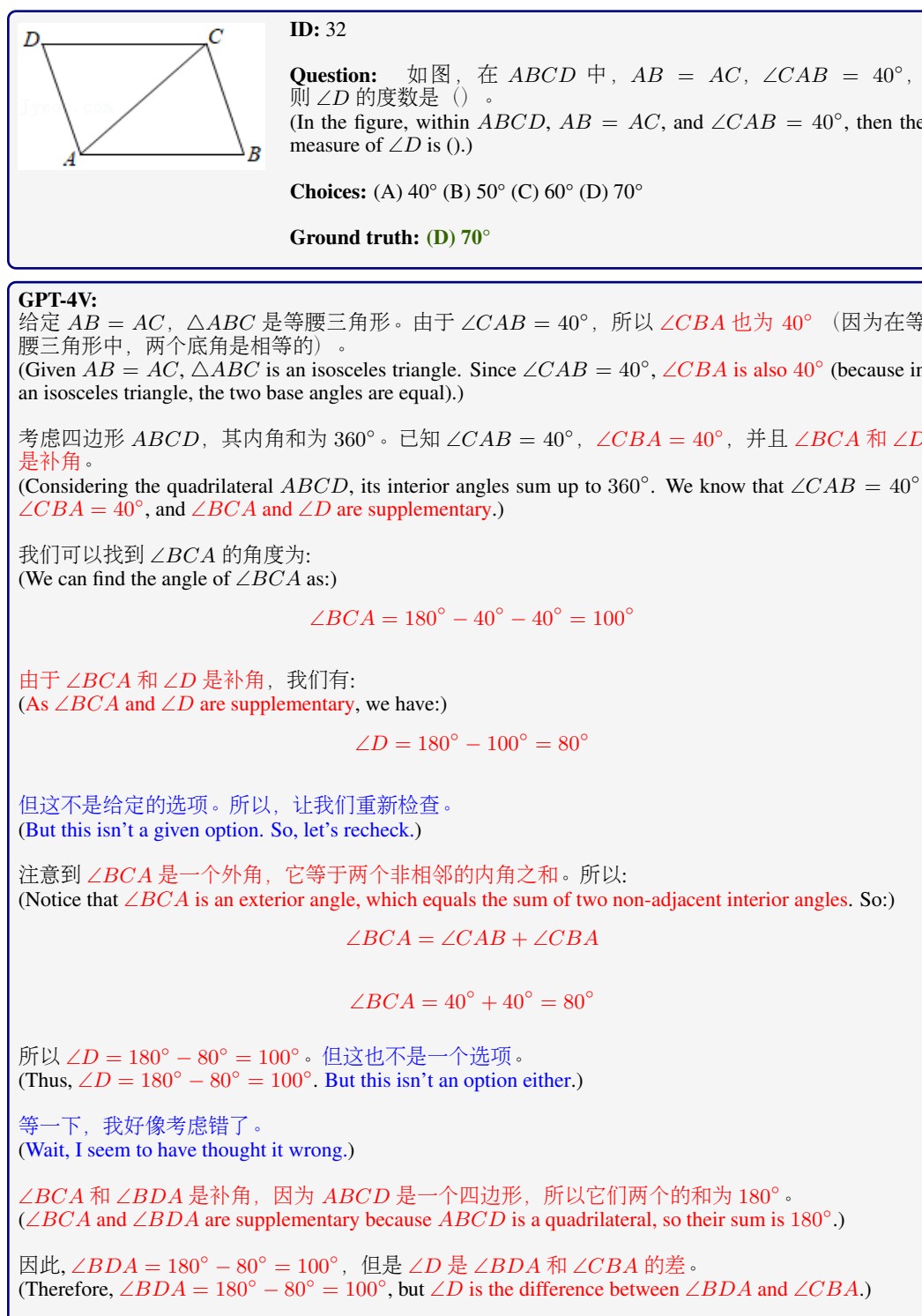

**ID:** 32

**Question:** 如图，在 $ABCD$ 中，$AB = AC$，$\angle CAB = 40°$，则 $\angle D$ 的度数是（）。
(In the figure, within $ABCD$, $AB = AC$, and $\angle CAB = 40°$, then the measure of $\angle D$ is ().)

**Choices:** (A) 40° (B) 50° (C) 60° (D) 70°

**Ground truth:** (D) 70°

**GPT-4V:**
给定 $AB = AC$，$\triangle ABC$ 是等腰三角形。由于 $\angle CAB = 40°$，所以 $\angle CBA$ 也为 40°（因为在等腰三角形中，两个底角是相等的）。
(Given $AB = AC$, $\triangle ABC$ is an isosceles triangle. Since $\angle CAB = 40°$, $\angle CBA$ is also 40° (because in an isosceles triangle, the two base angles are equal).)

考虑四边形 $ABCD$，其内角和为 360°。已知 $\angle CAB = 40°$，$\angle CBA = 40°$，并且 $\angle BCA$ 和 $\angle D$ 是补角。
(Considering the quadrilateral $ABCD$, its interior angles sum up to 360°. We know that $\angle CAB = 40°$, $\angle CBA = 40°$, and $\angle BCA$ and $\angle D$ are supplementary.)

我们可以找到 $\angle BCA$ 的角度为:
(We can find the angle of $\angle BCA$ as:)

$$\angle BCA = 180° - 40° - 40° = 100°$$

由于 $\angle BCA$ 和 $\angle D$ 是补角，我们有:
(As $\angle BCA$ and $\angle D$ are supplementary, we have:)

$$\angle D = 180° - 100° = 80°$$

但这不是给定的选项。所以，让我们重新检查。
(But this isn't a given option. So, let's recheck.)

注意到 $\angle BCA$ 是一个外角，它等于两个非相邻的内角之和。所以:
(Notice that $\angle BCA$ is an exterior angle, which equals the sum of two non-adjacent interior angles. So:)

$$\angle BCA = \angle CAB + \angle CBA$$

$$\angle BCA = 40° + 40° = 80°$$

所以 $\angle D = 180° - 80° = 100°$。但这也不是一个选项。
(Thus, $\angle D = 180° - 80° = 100°$. But this isn't an option either.)

等一下，我好像考虑错了。
(Wait, I seem to have thought it wrong.)

$\angle BCA$ 和 $\angle BDA$ 是补角，因为 $ABCD$ 是一个四边形，所以它们两个的和为 180°。
($\angle BCA$ and $\angle BDA$ are supplementary because $ABCD$ is a quadrilateral, so their sum is 180°.)

因此, $\angle BDA = 180° - 80° = 100°$，但是 $\angle D$ 是 $\angle BDA$ 和 $\angle CBA$ 的差。
(Therefore, $\angle BDA = 180° - 80° = 100°$, but $\angle D$ is the difference between $\angle BDA$ and $\angle CBA$.)

所以 (So)

$$\angle D = 100° - 40° = 60°$$

Figure 105: The question is presented in Chinese, with the translated English provided in parentheses after the Chinese text. GPT-4V is capable of performing self-verification until it produces a solution that matches one of the given options (highlighted in blue).

| | |
|---|---|
| oval beads | $2/kilogram |
| rectangular beads | $3/kilogram |
| star-shaped beads | $2/kilogram |
| spherical beads | $2/kilogram |
| heart-shaped beads | $3/kilogram |
| square beads | $2/kilogram |
| flower-shaped beads | $2/kilogram |

**ID:** 44

**Question:** Chase wants to buy 4 kilograms of oval beads and 5 kilograms of star-shaped beads. How much will he spend? (Unit: $)

**Ground truth: 18**

**Solution:** $8 + 4 - 2 = 10, 3 + 2 - 3 = 2, 12 + 1 - 5 = 8$

---

**Multimodal Bard:**

The cost of 4 kilograms of oval beads is $4 * 2 = \$8$.

The cost of 5 kilograms of star-shaped beads is $5 * 2 = \$10$.

So the total cost is $\$8 + \$10 = \$18$.

Therefore, Chase will spend $\$18$.

I also wrote a Python code to calculate the total cost and got the same answer of 35.

```python
def calculate_cost(oval_beads, star_shaped_beads):
    oval_bead_cost = 2
    star_shaped_bead_cost = 2
    total_cost = oval_beads * oval_bead_cost + star_shaped_beads *
        star_shaped_bead_cost
    return total_cost

print(calculate_cost(4, 5))

# 35
```

Figure 106: Multimodal Bard seems incapable of verifying its solutions and revisiting them if found invalid. For instance, in this example, after initially producing a natural language solution, Multimodal Bard subsequently generates Python code. However, the executed result of the code (35) differs from its initial solution (18).

## H.6    SELF-CONSISTENCY FOR GPT-4V

Self-consistency (Wang et al., 2022) is a decoding strategy for chain-of-thought prompting (Wei et al., 2022b). A diverse set of reasoning paths is sampled, and the most consistent answer is selected as the final prediction. Moving beyond vanilla greedy decoding, this method resorts to the inherent coherence and reliability of multiple reasoning trajectories to produce a more trustworthy conclusion. Self-consistency has been widely employed in LLMs for complex reasoning tasks, such as math word problems and commonsense reasoning.

In our experiments, we validated the effectiveness of using self-consistency for GPT-4V on MATH-VISTA. Given a question and context, we ran GPT-4V multiple times to obtain a set of different reasoning paths and then selected the most frequent answer as the final prediction. We found that self-consistency is instrumental in rectifying visual perception errors (Figure 107), correcting calculation mistakes (Figure 108), and mitigating hallucinations (Figure 109). In comparison, self-consistency is less effective when GPT-4V has difficulties in interpreting complex visual contexts (Figures 110, 111) or extracting salient information from images (Figure 112).

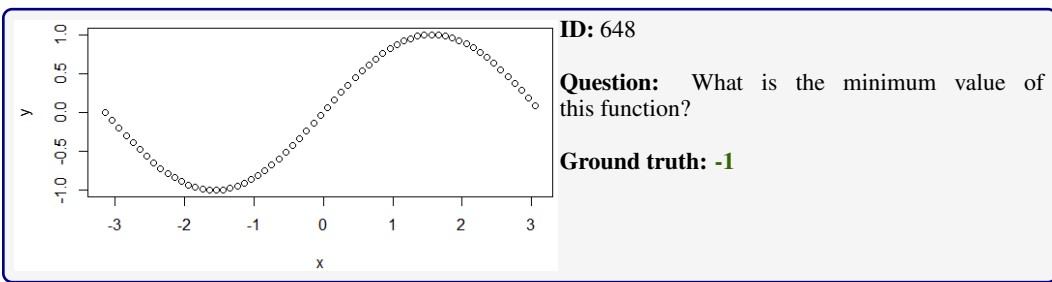

**ID:** 648

**Question:** What is the minimum value of this function?

**Ground truth: -1**

---

**GPT-4V (Reasoning Path 1):**
The minimum value of the function is observed where the graph is at its lowest point along the $y$-axis.

From the graph, the minimum value lies on the $y$-axis between -0.5 and 0. Based on the visual representation, it appears that the minimum value of $y$ is: -0.5.

So, the answer is -0.5.
**Wrong output: -0.5**

---

**GPT-4V (Reasoning Path 2):**
The graph shown represents a function with circular data points plotted across the $x$-axis. To determine the minimum value of the function, we need to locate the lowest point on the graph.

From the graph, it is evident that the lowest point (or the minimum value of the function) is at a $y$-coordinate of approximately -1.

Therefore, the minimum value of the function is -1.
**Correct output: -1**

---

**GPT-4V (Reasoning Path 3):**
The graph depicts the function using circular points. The minimum value of the function can be observed by identifying the lowest point on the graph.

From the graph, the lowest point seems to be at $y = $ -1.

Therefore, the minimum value of this function is -1.
**Correct output: -1**

---

Figure 107: In this algebraic reasoning task, GPT-4V displays inconsistency in recognizing the value range of the function. After reevaluating GPT-4V three times, three distinct reasoning paths emerge. Utilizing self-consistency minimizes variation in the generated results, converging on the most frequent prediction (**-1**) as the final answer.

| Web pages visited | |
|---|---|
| **Person** | **Number of Web pages** |
| Bonnie | 84 |
| Winston | 56 |
| Percy | 70 |
| Derek | 68 |
| Jake | 60 |
| Finn | 47 |
| Carter | 75 |
| Kurt | 52 |

**ID:** 331

**Question:** Several people compared how many Web pages they had visited. What is the mean of the numbers?

**Ground truth: 64**

**GPT-4V (Reasoning Path 1):**
To find the mean of the numbers, you add up all the numbers and then divide by the quantity of numbers.

$$\text{Mean} = (84 + 56 + 70 + 68 + 60 + 47 + 75 + 52) \div 8$$
$$= 462 \div 8$$
$$= 57.75$$

However, since the question requires an integer answer, the mean number of web pages visited is: 58.
**Wrong output: 58**

**GPT-4V (Reasoning Path 2):**
To find the mean number of web pages visited:

First, sum the number of web pages for all the individuals:

Bonnie: 84, Winston: 56, Percy: 70, Derek: 68, Jake: 60, Finn: 47, Carter: 75, Kurt: 52

$$\text{Total} = 84 + 56 + 70 + 68 + 60 + 47 + 75 + 52 = 512$$

Next, divide by the number of individuals (8):

$$\text{Mean} = 512 \div 8 = 64$$

The mean number of web pages visited is 64.
**Correct output: 64**

**GPT-4V (Reasoning Path 3):**
First, I'll list the number of web pages each person visited:

Bonnie: 84, Winston: 56, Percy: 70, Derek: 68, Jake: 60, Finn: 47, Carter: 75, Kurt: 52

Next, I'll group some numbers together for easier addition:

$$(84 + 56) = 140$$
$$(70 + 68) = 138$$
$$(60 + 47) = 107$$
$$(75 + 52) = 127$$

Now, I'll sum these results:

$$140 + 138 + 107 + 127 = 512$$

Now, I'll divide the sum by the number of individuals to get the mean:

$$512 \div 8 = 64$$

So, the mean number of web pages visited is 64.
**Correct output: 64**

Figure 108: For this arithmetic reasoning question, GPT-4V arrives at the answers of **58**, **64**, and **64** through three distinct reasoning paths. Applying self-consistency would confirm the correct answer to be **64**.

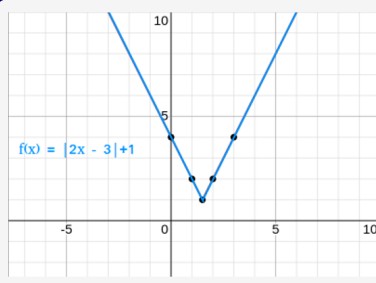

**ID:** 430

**Question:** The derivative of $f(x)$ at $x = 0$ is ______ that at $x = 5$

**Choices:** (A) larger than (B) equal to (C) smaller than

**Ground truth: (C) smaller than**

---

**GPT-4V (Reasoning Path 1):**
The graph represents the function $f(x) = |2x - 3| + 1$.

The slope (or the derivative) of $f(x)$ is represented by the inclination of the line.

At $x = 0$, the slope is negative (the line is declining), and at $x = 5$, the slope is positive (the line is inclining). The magnitude of the slopes appears to be the same; only their directions differ.

However, when considering the magnitude of the derivative, the value at $x = 0$ is negative, and the value at $x = 5$ is positive, but they have the same absolute value.
**Wrong output: (B) equal to**

---

**GPT-4V (Reasoning Path 2):**
The given function is $f(x) = |2x - 3| + 1$.

The absolute value function is piecewise defined, so its derivative will have different values depending on where you take it.

For $2x - 3 > 0$, which is $x > \frac{3}{2}$ or 1.5, $f(x) = 2x - 3$. The derivative is $f'(x) = 2$.

For $2x - 3 < 0$, which is $x < \frac{3}{2}$ or 1.5, $f(x) = -2x + 3$. The derivative is $f'(x) = -2$.

At $x = 0$, the derivative $f'(x) = -2$.

At $x = 5$, the derivative $f'(x) = 2$.

Therefore, the derivative of $f(x)$ at $x = 0$ is smaller than that at $x = 5$.
**Correct output: (C) smaller than**

---

**GPT-4V (Reasoning Path 3):**
The function provided is $f(x) = |2x - 3| + 1$.

The absolute value function has a V-shaped graph, as shown in the image. The slope (or derivative) of the line to the left of the vertex is negative, and the slope of the line to the right of the vertex is positive.

The derivative of $f(x)$ at $x = 0$ corresponds to the slope of the line on the left side of the vertex, which is negative (downward slope). The derivative of $f(x)$ at $x = 5$ corresponds to the slope of the line on the right side of the vertex, which is positive (upward slope).

Since a positive value is always larger than a negative value, the derivative of $f(x)$ at $x = 0$ is smaller than that at $x = 5$.
**Correct output: (C) smaller than**

Figure 109: This is another instance where the application of self-consistency across three reasoning paths results in a correct final prediction.

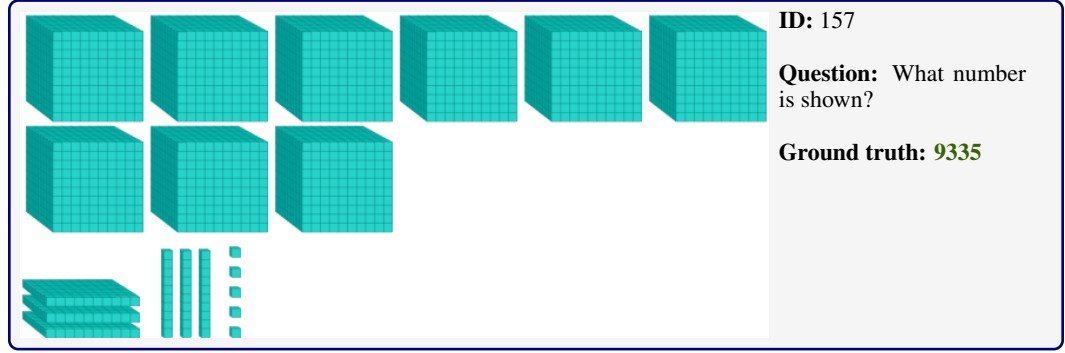

**ID:** 157

**Question:** What number is shown?

**Ground truth: 9335**

**GPT-4V (Reasoning Path 1):**
Let's break down the number shown using base-ten blocks:

- There are 6 large cubes, each representing $1,000$. So, $6 \times 1,000 = 6,000$.
- There are 7 flats, each representing 100. So, $7 \times 100 = 700$.
- There are 4 rods, each representing 10. So, $4 \times 10 = 40$.
- There are 3 units, each representing 1. So, $3 \times 1 = 3$.

Adding all of these values together:

$$6,000 + 700 + 40 + 3 = 6,743$$

The number shown is $6,743$.
**Wrong output: 6743**

**GPT-4V (Reasoning Path 2):**
The image displays a series of cubes that appear to represent numbers in base 10 using place value.

From left to right:

1. There are six large cubes, which represent a value of $6 \times 1000 = 6000$.
2. There are three medium-sized cubes, which represent a value of $3 \times 100 = 300$.
3. There are two small cubes, which represent a value of $2 \times 10 = 20$.
4. The vertical stack of single unit cubes shows nine cubes, which represent a value of $9 \times 1 = 9$.

Adding up these values:
$$6000 + 300 + 20 + 9 = 6329$$

The number shown is 6329.
**Wrong output: 6329**

**GPT-4V (Reasoning Path 3):**
The image displays a series of cubes representing numbers in base 10 using place value.

From left to right:

1. There are seven large cubes, which represent a value of $7 \times 1000 = 7000$.
2. There are two medium-sized cubes, which represent a value of $2 \times 100 = 200$.
3. There are nine small cubes, which represent a value of $9 \times 10 = 90$.
4. The vertical stack of single unit cubes shows eight cubes, which represent a value of $8 \times 1 = 8$.

Adding up these values:
$$7000 + 200 + 90 + 8 = 7298$$

The number shown is 7298.
**Wrong output: 7298**

Figure 110: In this arithmetic reasoning task, GPT-4V struggles with comprehending the abstract scene, leading to its inability to calculate the correct answer through multiplication. Multiple attempts yield varying incorrect solutions. Consequently, utilizing self-consistency does not verify the definitive right answer.

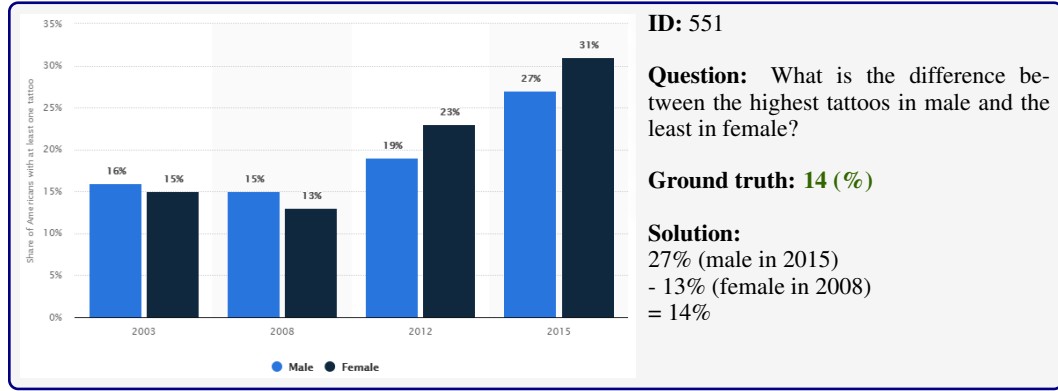

**ID:** 551

**Question:** What is the difference between the highest tattoos in male and the least in female?

**Ground truth:** 14 (%)

**Solution:**
27% (male in 2015)
- 13% (female in 2008)
= 14%

**GPT-4V (Reasoning Path 1):**
The highest percentage of tattoos in males is 31% (in 2015), and the least percentage of tattoos in females is 15% (in 2008).

Difference = 31% - 15% = 16%
**Wrong output: 16%**

**GPT-4V (Reasoning Path 2):**
The highest percentage of males with at least one tattoo is in 2015 at 31%.

The lowest percentage of females with at least one tattoo is in 2003, 2008, and 2012, all at 15%.

The difference between these values is: 31% - 15% = 16%
**Wrong output: 16%**

**GPT-4V (Reasoning Path 3):**
To determine the difference between the highest tattoos in males and the least in females:

Highest tattoos in males: 31% (in 2015)
Least tattoos in females: 15% (in 2003 and 2008)

Difference: 31% - 15% = 16%
**Wrong output: 16%**

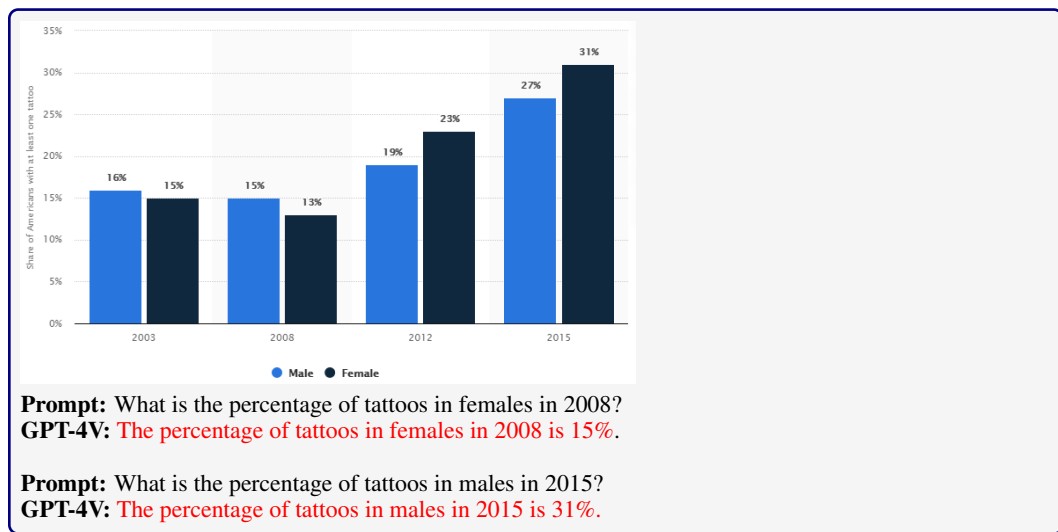

**Prompt:** What is the percentage of tattoos in females in 2008?
**GPT-4V:** The percentage of tattoos in females in 2008 is 15%.

**Prompt:** What is the percentage of tattoos in males in 2015?
**GPT-4V:** The percentage of tattoos in males in 2015 is 31%.

Figure 111: In this instance, GPT-4V persistently produces incorrect predictions across multiple evaluations. When prompted to detail the essential elements of the image, GPT-4V provides inaccurate descriptions. This suggests that if GPT-4V misinterprets vital visual cues, relying on self-consistency may not necessarily guide it towards the correct conclusion.

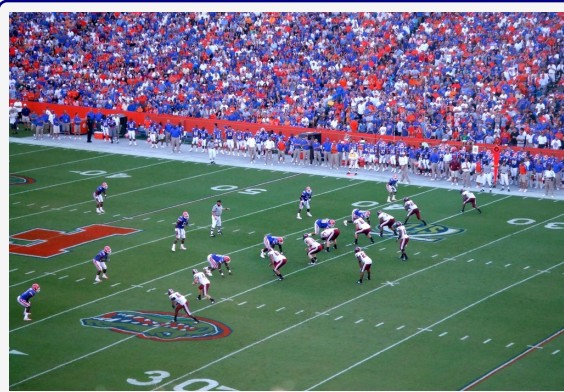

**ID:** 965

**Question:** What's the lowest number yard line that you can see?

**Ground truth: 30**

**GPT-4V (Reasoning Path 1):**
The lowest number yard line visible in the image is the 10-yard line.

**GPT-4V (Reasoning Path 2):**
From the image provided, the lowest number yard line that is clearly visible is the "10" yard line.

**GPT-4V (Reasoning Path 3):**
From the image provided, the lowest number yard line that can be seen is the 10-yard line.

**GPT-4V (Reasoning Path 4):**
From the image, the lowest number yard line that is visible is the "1" yard line.

**GPT-4V (Reasoning Path 5):**
From the image, the lowest number yard line that is visible is the 20-yard line.

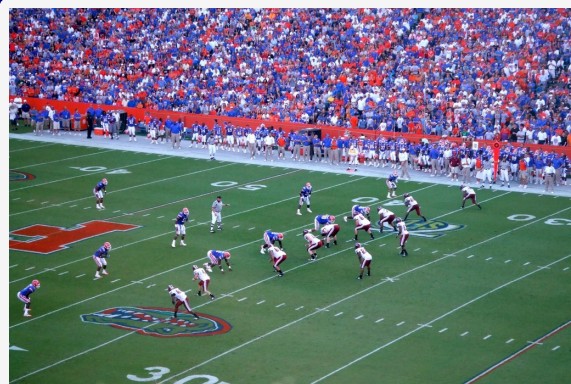

**Prompt:** List the yard line numbers that are visible in the image.
**GPT-4V:** The visible yard line numbers in the image are 10, 20, 30.

**Prompt:** Can you see the yard numbers 40 and 50 in the image?
**GPT-4V:** No, the yard numbers 40 and 50 are not visible in the image.

Figure 112: This example demonstrates that if GPT-4V misinterprets key information from an image, the application of self-consistency may not aid in producing the correct final prediction. We evaluated GPT-4V five times for the question "What's the lowest number yard line that you can see?" and received consistently incorrect responses. Further, when prompted to describe the yard line numbers present in the image, GPT-4V provided incorrect descriptions, indicating its difficulty in accurately recognizing them.

### H.7 GPT-4V FOR MULTI-TURN HUMAN-AI INTERACTION

This section investigates the use of GPT-4V for multi-turn human-AI interaction on MATHVISTA, as exemplified in the goal-directed dialog in Figure 113.

We found that GPT-4V is effective in engaging multi-turn goal-directed conversations with users. In particular, GPT-4V can make good use of *hints* (*e.g.,* user feedback or responses) to guide the conversion to generate desirable results. For instance, it can (1) rectify visual perception errors based on hints (Figure 114), (2) reassess reasoning steps and calculations (Figure 115), (3) correct misinformation using user-provided domain-specific knowledge (Figure 116), and (4) aggregate intricate contexts over multiple turns in a human-AI conversation (Figures 117 and 118).

We also observed failure cases in our evaluation. For instance, GPT-4V struggles to generate correct responses when questions and user hints are ambiguous (Figure 119), or when the model fails to understand abstract shapes and concepts visually (Figure 120). These failures motivate the development of more powerful, conversational foundation models.

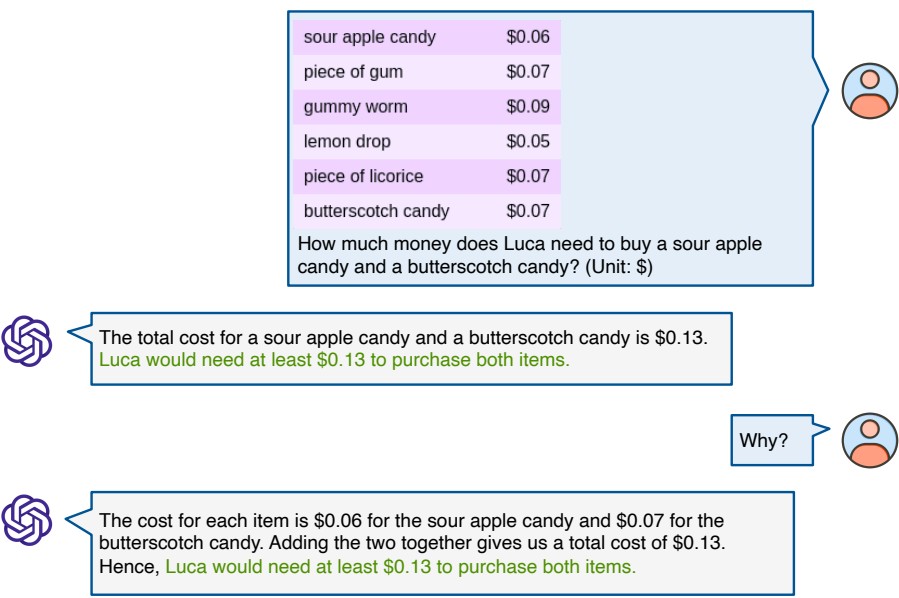

Figure 113: A multi-turn dialog between a human user and GPT-4V.

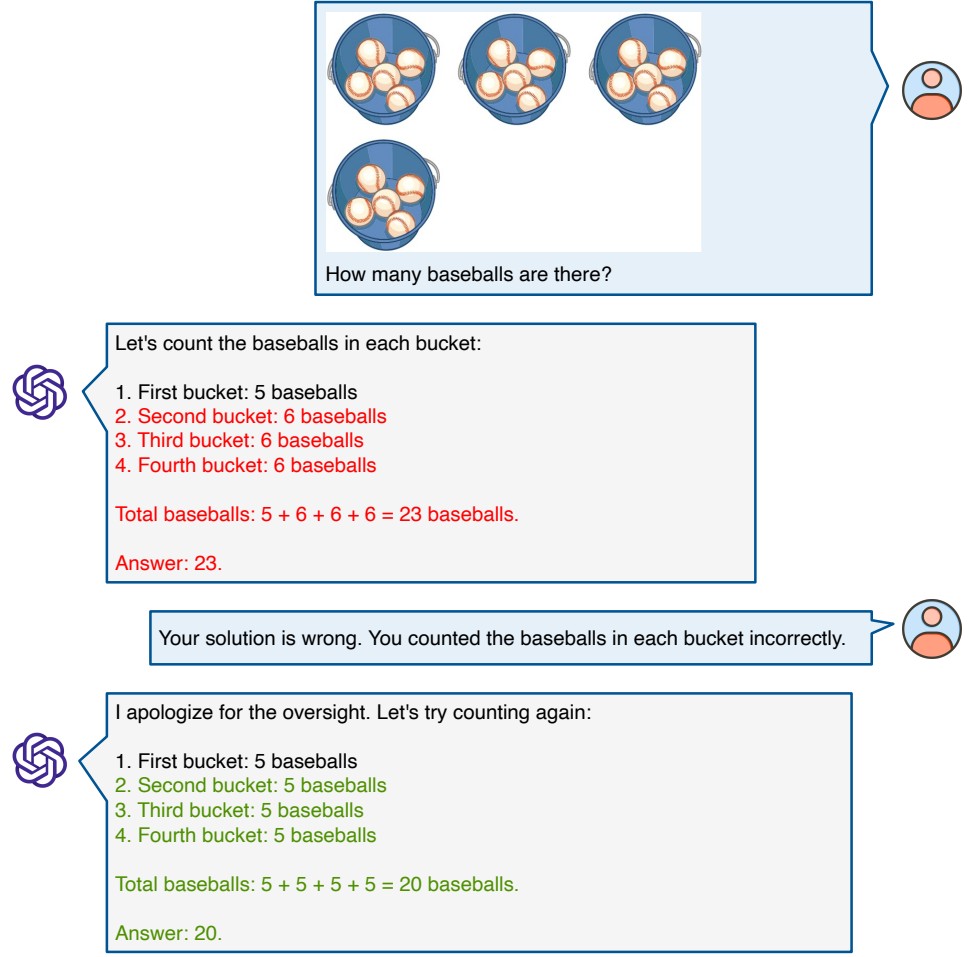

Figure 114: GPT-4V initially produces an incorrect solution due to errors in visually perceiving the number of baseballs. After receiving user feedback, it recalibrates and generates the accurate answer.

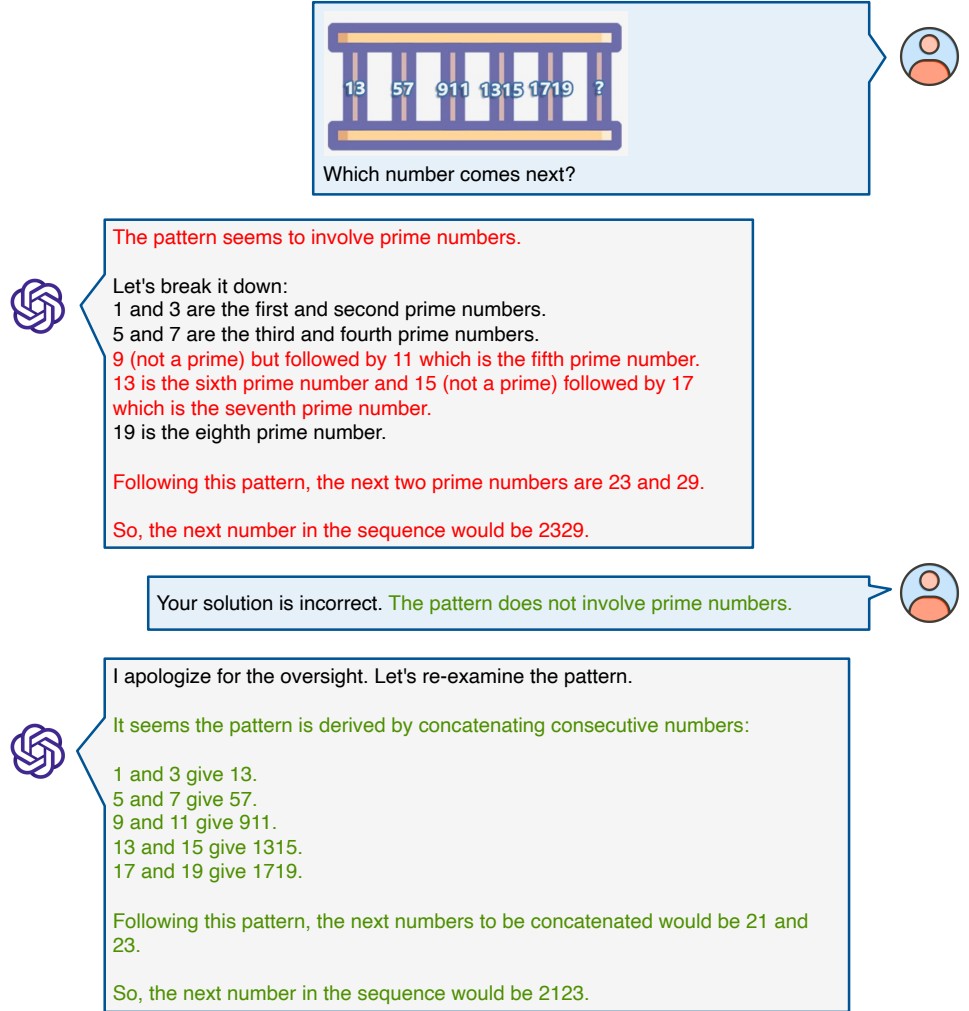

Figure 115: GPT-4V correctly identifies the sequence of numbers in the images but struggles to discern the underlying pattern. With feedback from the user, GPT-4V re-evaluates the pattern and produces the correct response to the logical reasoning question.

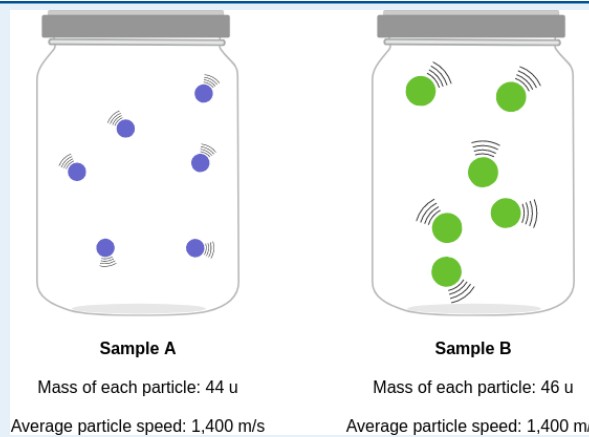

**Sample A**

Mass of each particle: 44 u

Average particle speed: 1,400 m/s

**Sample B**

Mass of each particle: 46 u

Average particle speed: 1,400 m/s

The diagrams below show two pure samples of gas in identical closed, rigid containers. Each colored ball represents one gas particle. Both samples have the same number of particles. Compare the average kinetic energies of the particles in each sample. Which sample has the higher temperature?
Choices: (A) sample A (B) sample B (C) neither; the samples have the same temperature

The average kinetic energy of an ideal gas particle is directly proportional to the temperature of the gas and is given by the equation:

$$KE_{avg} = \frac{3}{2} k\,T$$

Where:
- $KE_{avg}$ is the average kinetic energy per molecule.
- $k$ is the Boltzmann constant.
- $T$ is the absolute temperature.

For gases at the same temperature and under the same conditions, the average kinetic energy is the same, irrespective of the mass of the particles.

Given that the average particle speed in both Sample A and Sample B is 1,400 m/s, and since they are in identical closed, rigid containers, it implies that both samples have the same average kinetic energy. Therefore, they also have the same temperature.

The correct answer is: (C) neither; the samples have the same temperature.

Your solution is incorrect. I am providing retrieved knowledge to help you:

The temperature of a substance depends on the average kinetic energy of the particles in the substance. The higher the average kinetic energy of the particles, the higher the temperature of the substance. The kinetic energy of a particle is determined by its mass and speed. For a pure substance, the greater the mass of each particle in the substance and the higher the average speed of the particles, the higher their average kinetic energy.

I apologize for the oversight. Let's reconsider the question.

The kinetic energy ($KE$) of a particle is given by the equation:

$$KE = \frac{1}{2} m v^2$$

Where:
- $m$ is the mass of the particle.
- $v$ is the velocity (or speed) of the particle.

Given:
- Both Sample A and Sample B have particles with the same average speed (1,400 m/s).
- The mass of each particle in Sample A is 44 u and in Sample B is 46 u.

Given that the speeds are the same but the mass of the particles in Sample B is greater than that of Sample A, the particles in Sample B will have a higher kinetic energy than those in Sample A. As the temperature is directly related to the average kinetic energy of the particles, Sample B will have a higher temperature than Sample A.

Therefore, the correct answer is: (B) sample B.

Figure 116: Initially, GPT-4V applies an incorrect theorem to address the scientific reasoning query. However, after receiving the appropriate domain knowledge hint from the user, it utilizes the correct theorem, combined with key visual cues from the images, to produce the right answer.

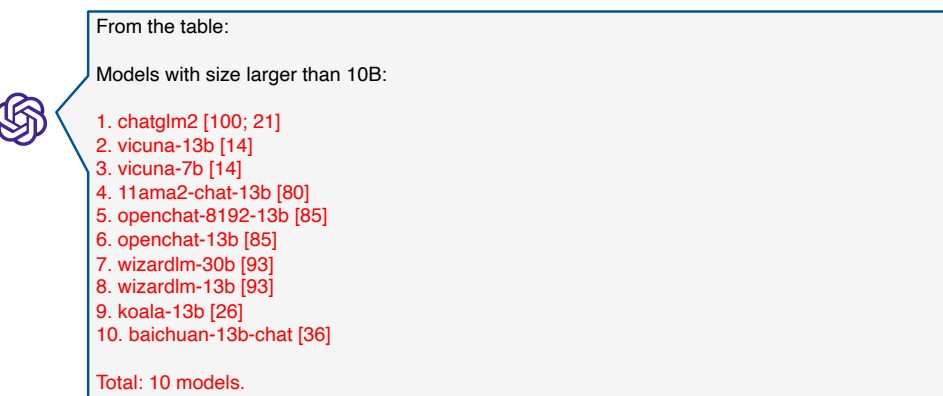

Table 1: AgentBench evaluates 25 API-based or open-sourced LLMs on LLM-as-Agent challenges.

| Model | Model Size | Form | Version | Creator |
|---|---|---|---|---|
| gpt-4 [56] | undisclosed | api | 0613 | |
| gpt-3.5-turbo [55] | undisclosed | api | 0613 | OpenAI |
| text-davinci-003 [58] | undisclosed | api | - | |
| text-davinci-002 [58] | undisclosed | api | - | |
| claude [6] | undisclosed | api | v1.3 | Anthropic |
| claude-instant [6] | undisclosed | api | v1.1 | |
| text-bison-001 [3] | undisclosed | api | - | Google |
| chatglm2 [100; 21] | undisclosed | api | v0.2 | |
| chatglm-6b [100; 21] | 6B | open | v1.1 | Tsinghua & Zhipu |
| codegeex2-6b-chat [102] | 6B | open | - | |
| vicuna-13b [14] | 13B | open | v1.1 | LMSYS |
| vicuna-7b [14] | 7B | open | v1.1 | |
| llama2-chat-13b [80] | 13B | open | - | Meta |
| llama2-chat-7b [80] | 7B | open | - | |
| openchat-8192-13b [85] | 13B | open | v1 | Tsinghua |
| openchat-13b [85] | 13B | open | v3.2 | |
| wizardlm-30b [93] | 30B | open | v1.0 | |
| wizardcoder-15b [48] | 15B | open | v1.0 | Microsoft |
| wizardlm-13b [93] | 13B | open | v1.0 | |
| koala-13b [26] | 13B | open | - | UC Berkeley |
| internlm-chat-7b [79] | 7B | open | - | InternLM Team |
| baichuan-13b-chat [36] | 13B | open | - | Baichuan Inc. |
| baichuan-7b [36] | 7B | open | - | |
| dolly-12b [16] | 12B | open | v2 | Databricks |
| oasst-sft-4-12b [42] | 12B | open | - | LAION |

How many models in the table have a model size larger than 10B?

From the table:

Models with size larger than 10B:

1. chatglm2 [100; 21]
2. vicuna-13b [14]
3. vicuna-7b [14]
4. 11ama2-chat-13b [80]
5. openchat-8192-13b [85]
6. openchat-13b [85]
7. wizardlm-30b [93]
8. wizardlm-13b [93]
9. koala-13b [26]
10. baichuan-13b-chat [36]

Total: 10 models.

Figure 117: Answering this question requires comprehending a complex table regarding AI models (such as identifying relevant rows and columns), and up-to-date knowledge of AI research, and the ability to perform advanced arithmetic reasoning. Without human in the loop, GPT-4V fails to produce the correct answer.

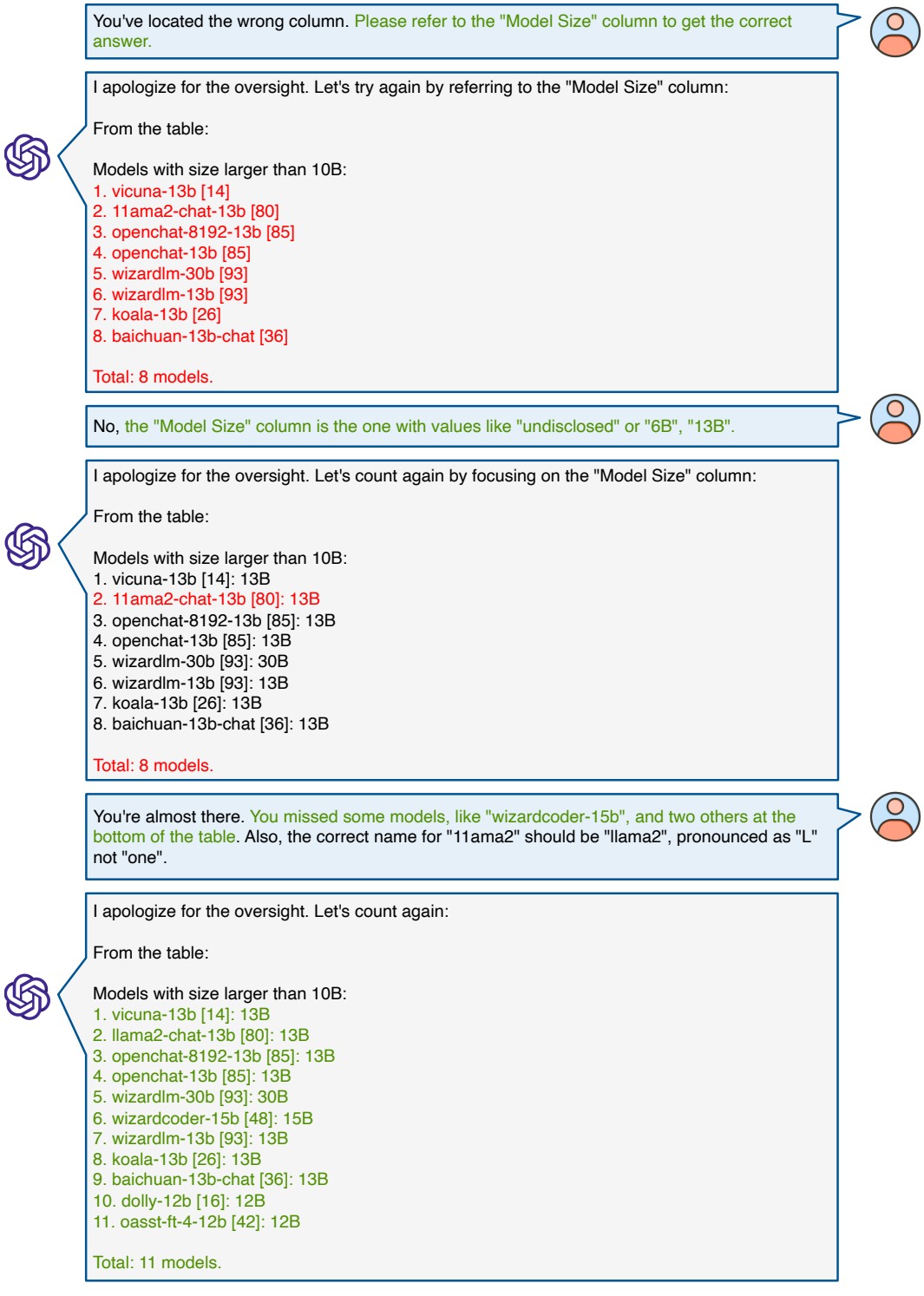

Figure 118: (Continued) In this multi-turn human-AI conversation, the user acts as a tutor to help GPT-4V generate the correct answer . (1) Initially, the user directs GPT-4V to the "Model Size" column after observing its misdirection. Despite the guidance, GPT-4V still struggles. (2) Recognizing the model's difficulty in identifying the "Model Size" column, the user provides more detailed instructions. This time, GPT-4V is on the right track but still makes some error in its response. (3) With further information from the user, GPT-4V corrects the remaining errors and generate the correct answer.

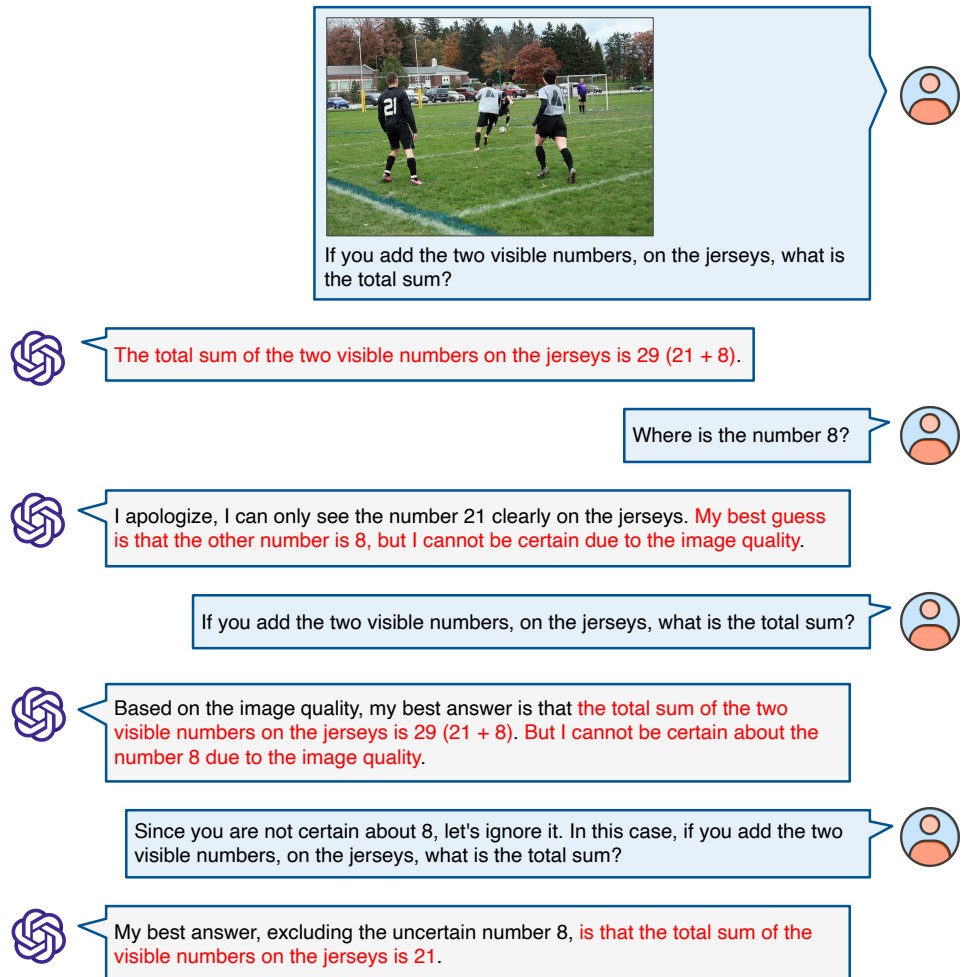

Figure 119: A failure example from GPT-4V in an interactive chatbot environment. Utilizing the visual cues from the image to resolve the question's ambiguity (distinguishing between two numbers, 2 and 1, instead of the single number 21), the correct answer is 3, obtained by summing 2 and 1.

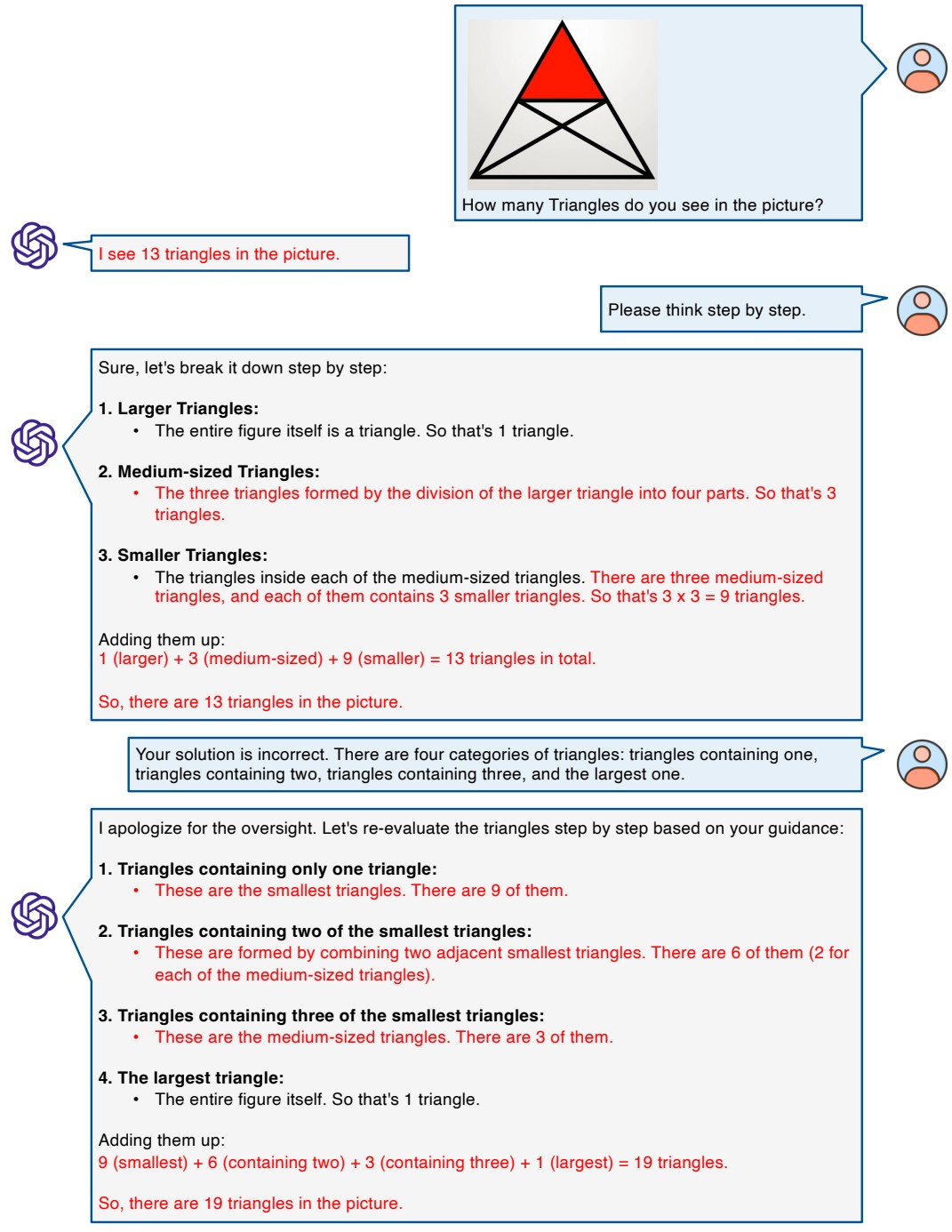

Figure 120: In this example, even after the user offers helpful hints, GPT-4V struggles to decipher the number of triangles in the given geometric shape. The task is challenging for AI models, including GPT-4V, with a limited ability to understand abstract shapes visually.

