# OpenReview forum: "MathVista: Evaluating Mathematical Reasoning of Foundation Models in Visual Contexts"
_ICLR.cc/2024/Conference — ICLR 2024 oral_

### Official Review · Reviewer_iJ4P · 2023-10-13

**Soundness:** 3 good
**Presentation:** 4 excellent
**Contribution:** 3 good
**Rating:** 8
**Confidence:** 4

**Summary:**

The paper focuses on the evaluation of mathematical reasoning based on visual inputs. This work presents a review of existing work on the topic, a new benchmark (MathVista) made of existing datasets plus three new ones, and a large evaluation of existing large pretrained models on this benchmark.

**Strengths:**

- Thorough review of the existing work on the topic, with a proposed taxonomy that clarifies and organizes the capabilities and settings relevant to visual/mathematical reasoning.

- Consolidation of existing datasets into a comprehensive benchmark.

- Large evaluation of existing models, under various settings (zero-shot, few-shot ICL, with various prompting strategies).

**Weaknesses:**

W1. A potential downside of the chosen tasks is that they "amalgamate" mathematical and visual reasoning (as stated in the abstract). This does not seem desirable since one would usually also wants to understand the capabilities of a model for these two steps (visual understanding and reasoning) independently. The argument that there exists other benchmarks that do look at these individual capabilities means that this is however not a critical issue.

------

W2. The low human performance on the benchmark (~60% accuracy) is concerning. Could this indicate an issue with data quality of annotation noise? (rather than intrinsic task difficulty)

**Questions:**

Please comment on W2 above.

Minor question regarding the sampling of data for "testmini", the text mentions the following:
"The KL Divergence and Total Variation (TV) distance between the testmini set and the entire set are 0.008 and 0.035"
What is being compared with KL and TV distances? Distributions of what?

---

> ### Author Response · Authors · 2023-11-17
> **Response to Reviewer iJ4P: Integrating Math and Visual Reasoning and Addressing Benchmark Challenges**
>
> Dear Reviewer,
>
> Thank you so much for your thoughtful review. We are encouraged by your recognition of our contributions in presenting a comprehensive benchmark and conducting a large evaluation of existing models under various settings.
>
> We hope to engage further with you to enhance our paper. Your valuable comments have been addressed as follows:
>
> ### W1: The dataset "amalgamates" mathematical and visual reasoning
>
> Thank you for your insightful comment. We would like to clarify that our objective in designing these tasks was to **mirror the intricacies of real-world situations**, where mathematical and visual reasoning are often interlinked and not easily separable. A prime example is K-12 math education, which notably emphasizes visual learning through the use of pictures, diagrams, and real-life scenarios. This approach underscores the significance of integrating visual elements with arithmetic tasks to accurately assess a child's math and reasoning skills. Moreover, our dataset reflects the practical applications found in data analysis, rich with figures, and scientific discovery that involves deducing theorems from multimodal raw results.
>
> Our MathVista benchmark aims to **fill a unique gap by providing a more holistic and comprehensive evaluation** of LLMs and LMMs. This approach is particularly relevant for advancing AI systems capable of tackling multifaceted and contextually rich tasks. We believe that the combined evaluation of mathematical and visual reasoning offers novel insights into the overall capabilities of these models, contributing significantly to our understanding of AI in complex problem-solving scenarios.
>
> We have conducted **detailed analyses to dissect model performances on individual aspects within these integrated tasks** on MathVista (as detailed in Sections 3.4 and 3.5, and Appendices F.2, F.3, and F.4). These analyses provide insights into how models navigate the challenges of visual understanding and mathematical reasoning both individually and in tandem.
>
> ### W2: The low human performance on the benchmark
>
> Thank you for your observation regarding the human performance on our benchmark, MathVista. The 60.3% accuracy achieved by humans indeed reflects the challenging nature of the tasks, which include advanced topics such as college-level mathematics and complex scientific reasoning. We would like to clarify that this outcome reflects **the intrinsic difficulty of the tasks rather than issues with data quality**.
>
> 1. **Task complexity:**
>
> Our benchmark is designed to be challenging, featuring tasks that require advanced mathematical reasoning within complex visual contexts. These tasks often involve high-level concepts drawn from **college curricula (10.8%)** and **scientific reasoning (10.7%)**, going beyond standard high school level problems. The tasks in MathVista are multidimensional, requiring not only mathematical skills (i.e., 41% of problems involve two math reasoning types), but also an in-depth understanding of visual elements. 44.8% of the questions are free-form, without option candidates. All these natures significantly raise the difficulty bar.
>
> 2. **Data quality assurance:**
>
> To ensure the integrity of our data, we **have implemented rigorous quality control measures**. Our team engaged with domain experts from STEM graduates and a professional data labeling company. We developed three professional tools for data labeling and collection (see Figures 22-24 in Appendix C.2, C.3, and C.4), ensuring accuracy and reliability. To label if the candidate examples involve mathematical reasoning, each is labeled with three expert annotators, and we did an inter-annotator agreement analysis. Our analysis yielded a Fleiss Kappa score of 0.775, indicating a substantial level of inter-consistency. We took multiple rounds of checks and validations by domain experts. We further checked 500 examples and all seemed doable in the sense that they had enough information to solve the question. For more data curation details, please refer to Section 2 and Appendix C.
>
> 3. **Benchmarking against human performance:**
>
> The benchmark's purpose is to push the boundaries of current AI capabilities in mathematical reasoning within visual contexts. The 60.3% human accuracy rate **sets a realistic and challenging target for AI models**, emphasizing the need for improvement in this area. It is also worth noting that the human baseline was established by engaging qualified annotators with at least a high school diploma, ensuring a reasonable level of competence.
>
> In conclusion, we believe it accurately reflects the challenging nature of the tasks in MathVista and the need for advanced AI capabilities in this domain.

---

> ### Author Response · Authors · 2023-11-17
> **Continued Response to Reviewer iJ4P: Clarifying KL and TV Distances**
>
> ### Q1: What is being compared with KL and TV distances?
>
> Thank you for your question regarding the sampling methodology for our 'testmini' dataset. In our analysis, the KL Divergence and Total Variation (TV) distances were employed to compare **the distributions of source datasets** between the 'testmini' set and the entire dataset. Since each source dataset studies a specific type of mathematical reasoning, visual context, task, and grade level, the similar distributions of source datasets lead to **similar distributions of fine-grained features**.
>
> This comparison was crucial to ensure that the 'testmini' set **accurately represents the overall dataset** in terms of task diversity, complexity, math reasoning, and visual contexts. By confirming the similarity of distributions through low KL Divergence (0.008) and TV distance (0.035), we aimed to validate that our 'testmini' set is a reliable subset for conducting representative evaluations of model performance. We compare the model accuracy performance on both 'testmini' and 'test' as follows:
>
> | **Model** | **Acc (%) on testmini (1,000 examples)** | **Acc (%) on test (5,141 examples)** | **Acc Diff.** |
> | - | - | - | - |
> | Frequent guess | 26.3 | 23.5 | -2.8 |
> | 2-shot CoT GPT-4 | 33.2 | 30.5 | -2.7 |
> | LLaVA-LLaMA-2-13B | 26.1 | 25.4 | -0.7 |
> | 2-shot PoT GPT-4 | 33.9 | 31.7 | -2.2 |
>
> The models **consistently have a marginal accuracy drop** of 0.7-2.8% on 'test' compared to on 'testmini', which guarantees that our findings on the 'testmini' set are **generalizable** to the larger 'test' set.
>
> We hope this explanation clarifies our methodology. We are happy to provide further details if needed and appreciate the opportunity to clarify this aspect of our methodology.

---

> > ### Comment · Reviewer_iJ4P · 2023-11-17
> > **Thanks for the response. Still rated as an "accept".**
> >
> > Thanks to the authors for the thorough response. I am satisfied with the additional information. With the other updates including the addition of GPT-4V results, I believe this is a strong paper that I would like to see presented at the conference.

---

> > > ### Author Response · Authors · 2023-11-18
> > > **Grateful Acknowledgment of the Positive Assessment**
> > >
> > > Dear Reviewer iJ4P,
> > >
> > > We greatly appreciate your recognition of the strength and updates of our paper. Your favorable evaluation and support for presenting our paper at the conference are highly motivating. We are grateful for your valuable feedback and guidance throughout this process.
> > >
> > > Best regards,
> > >
> > > The authors of Paper 6738

---

### Official Review · Reviewer_Vb5K · 2023-10-27

**Soundness:** 3 good
**Presentation:** 3 good
**Contribution:** 3 good
**Rating:** 8
**Confidence:** 4

**Summary:**

The paper proposes a benchmark to evaluate the ability of math reasoning with visual contexts. The test-only benchmark contains a relatively large number of examples (6k), featuring a wide coverage of math reasoning types (7), data sources (28), and task types (5). On the benchmark, representative multi-modal LLMs are evaluated. The results show that Bard is the best performing model among all the evaluated ones, getting 35% accuracy, which indicates that the current models still suffer from math reasoning.

**Strengths:**

1. The dataset is well-designed and relatively large. It contains 6k examples, coming from various (28) sources including existing ones and newly collected ones. The datasets also evaluate various types of reasoning abilities and tasks. The images contain synthetic, real and text-heavy images, which is a good coverage of different types.
2. The paper focuses on a specific topic and defines the problem well, which is important for LLM evaluation.
3. Several SoTA multimodal LLMs, including miniGPT-4, LLaVA(R), Bard, Instruct-BLIP, etc. are evaluated on the benchmark, and show reasonable results.

**Weaknesses:**

1. How to disentangle the visual understanding ability or text understanding ability with the math reasoning ability? For example, if a model incorrectly answers “how many cars are to the left of the tree”, it could be the error in spatial understanding (failing to find the cars on the left), or error in counting, or the model cannot understand this question. In the current form of the dataset, there is no disentangled evaluation of the visual/text understanding versus math reasoning. A possible way to address this questions could be having the annotations for **rationales** (results of intermediate reasoning steps or sub-questions), and evaluate the model’s performance against the intermediate steps.
2. Why is the human accuracy only 60.3% on the dataset? Does this suggest that the dataset is noisy, containing ambiguous/uncertain cases, or simply the task is very difficult thus humans are not good at it as well? An analysis on the 40% of the data where humans cannot answer correctly will be preferred.
3. While it is good that the paper defines 7 types of fine-grained reasoning ability, what are the take-away messages (of Tab-2) that can be derived with the differences in the 7 types? It would be nice if the expertise of different models can be reflected using the fine-grained types, besides that Bard is the best model
4. Results of GPT-4V? I understand that the model is not released by the submission deadline, but it would be good to have the results in later versions.

**Questions:**

See weakness.

---

> ### Author Response · Authors · 2023-11-17
> **Response to Reviewer Vb5K: Disentangling Skills on MathVista**
>
> Dear Reviewer,
>
> We sincerely appreciate your insightful and constructive review of our paper. We are gratified that you noted the novelty and wide coverage of our dataset. It is highly valued that you acknowledge the importance of our paper for LLM evaluation and highlight the reasonable results from our comprehensive evaluations with SoTA multimodal LLMs.
>
> We **have made substantial additions and revisions**, as detailed in [the updated submission](https://openreview.net/pdf?id=KUNzEQMWU7), to further enhance the depth of our work. Your feedback is instrumental and we look forward to **further engagement**.
>
> ### W1: How to disentangle the visual understanding ability or text understanding ability with the math reasoning ability?
>
> Thank you for highlighting the important challenge of disentangling visual and textual understanding from mathematical reasoning in AI model evaluation. We agree that this interplay of skills, while reflective of real-world applications, presents a complex evaluation scenario.
>
> In the current version of MathVisa, we have endeavored to address this issue by including annotations where possible. Notably, **85.6% of the questions** are accompanied by annotations sourced from their original datasets. **These annotations serve various purposes, ranging from specifying OCR texts and attributes within visual figures to providing textual solutions that elucidate the mathematical reasoning steps involved**. These annotations offer a level of disentanglement in the evaluation process. For instance, examples sourced from TabMWP (17) are annotated not only with ground truth tabular parsing results to verify models' understanding of the visual context but also with natural language solutions that illuminate the multiple intermediate steps leading to the final answers.
>
> Below is a snapshot of these annotations, which serve the purpose of evaluating visual or textual reasoning:
>
> | **ID** | **Source dataset** | **Annotation** | **Visual?** | **Textual?** |
> | - | - | - | - | - |
> | 1 | A-OKVQA | A list of rationales | Yes | Yes |
> | 2 | ChartQA | Text in the chart figure | Yes | / |
> | 3 | CLEVR-Math | The program that synthesizes the figure | Yes | / |
> | 4 | DocVQA | OCR text with the bounding boxes in the document image | Yes | / |
> | 5 | FigureQA | Structured attributes that generate the figures | Yes | / |
> | 6 | Geometry3K | Unified logic forms for the diagram and question text | Yes | / |
> | 7 | GeoQA+ | A Natural language solution | / | Yes |
> | 8 | IQTest | A Natural language solution | / | Yes |
> | 9 | KVQA | Named entities and the wiki captions | / | Yes |
> | 10 | MapQA | OCR text in the map graph | Yes | / |
> | 11 | PaperQA | The linked paper | / | Yes |
> | 12 | ParsVQA-Caps | The translation of the non-English question | / | Yes |
> | 13 | PlotQA | OCR text with the bounding boxes in the plot figure | Yes | / |
> | 14 | SciBench | The LaTeX format solution | / | Yes |
> | 15 | ScienceQA | A natural language solution with a related lecture | / | Yes |
> | 16 | Super-CLEVR | The program that synthesizes the figure | Yes | / |
> | 17 | TabMWP | A natural language solution and the diagram parsing results | Yes | Yes |
> | 18 | TextVQA | Detected objects from the image | Yes | / |
> | 19 | TheoremQA | A related theorem with a detailed definition | / | Yes |
> | 20 | TQA | A lecture that explains the question | / | Yes |
> | 21 | UniGeo | An expression that generates the answer | / | Yes |
> | 22 | VQA-AS | Fine-grained question and answer types | / | Yes |
> | 23 | VQA-RAD | Biological organ parts | Yes | / |
> | 24 | VQA2.0 | Fine-grained question and answer types | / | Yes |
>
> Furthermore, in response to your suggestion, we are exploring the possibility of **expanding our dataset to include more detailed annotations** for both visual (such as fine-grained captions) and textual aspects (including natural language and program solutions)**. This enhancement aims to facilitate a more nuanced evaluation of models' capabilities, specifically in terms of their ability to isolate and apply distinct cognitive skills such as visual perception, textual understanding, and mathematical reasoning.
>
> We hope these efforts and future enhancements will address the concerns raised and contribute to a more nuanced understanding of AI model performance in complex, multimodal scenarios.

---

> ### Author Response · Authors · 2023-11-17
> **Continued Response to Reviewer Vb5K: Analyzing Human Performance on the Challenging MathVista Benchmark**
>
> ### W2: Human accuracy achieves only 60.3% on the dataset
>
> Thank you for noting the human accuracy rate on our MathVista dataset. **The 60.3% accuracy indeed reflects the challenging nature of our benchmark**, which aims to test the limits of both human and AI capabilities in mathematical reasoning within visual contexts. The tasks in MathVista are designed to **require not just basic mathematical understanding but also deep, compositional reasoning combined with visual perception**, and they incorporate **advanced concepts such as college-level mathematics (10.8%) and complex scientific reasoning (10.7%)**, presenting significant challenges even for well-qualified individuals.
>
> As elaborated in Appendix E.6, we **conducted a meticulous study to gauge human performance** on MathVista. We utilized Amazon MT, where each question from the testmini subset was assigned to five annotators with a track record of completing over 5,000 HIT tasks and maintaining an acceptance score above 0.99, ensuring high-quality results. The study comprised five test questions and two qualification questions, all to be completed within 20 minutes. Only annotators who correctly answered the qualification questions were included in the final analysis. Furthermore, we selectively considered results from annotators with at least a high school diploma, aligning with the complexity of MathVista's questions, which are predominantly high-school (30.9%) and college-level (10.8%).
>
> Addressing your specific query, among the instances where humans did not answer correctly, only 1.9% of questions received empty responses, while 3.3% were somewhat ambiguous. Together, they account for just 5.2% of the evaluation set, as shown below:
>
> | | **Percentages** |
> | - | - |
> | Unanswered | 1.9% |
> | Ambiguous questions | 3.3% |
>
> In our further exploration to **identify which specific types of questions posed challenges for humans**, we reported the gains of human performance over random chances across **different tasks**, **math reasoning types**, and **grade levels** as follows:
>
> **(1) For different tasks:**
>
> | | **Random Chance** | **Human** | **Gain (%)** |
> | - | - | - | - |
> | MWP | 3.8 | 73.0 | +69.2 |
> | TQA | 19.6 | 63.2 | +43.6 |
> | FQA | 18.2 | 59.7 | +41.5 |
> | **VQA** | 26.3 | 55.9 | **+29.4** |
> | **GPS** | 21.6 | 48.4 | **+26.8** |
>
> Humans exhibited the lowest performance gain in Visual Question Answering (VQA) at 29.4% and Geometry Problem Solving (GPS) at 26.8%. The VQA tasks, being knowledge-intensive, demand a comprehensive understanding that extends beyond mere visual perception. For example, they often require the comprehension of detailed profile information of celebrities. Similarly, the GPS tasks necessitate accurate interpretation of diagrams, application of domain-specific theorems, and precise calculations.
>
> **(2) For different math reasoning types:**
>
> | | **Random Chance** | **Human** | **Gain (%)** |
> | - | - | - | - |
> | STA (statistical) | 20.9 | 63.9 | +43.0 |
> | ARI (arithmetic) | 18.7 | 59.2 | +40.5 |
> | NUM (numeric) | 19.4 | 53.8 | +34.4 |
> | SCI (scientific) | 32.0 | 64.9 | +32.9 |
> | **GEO (geometry)** | 31.4 | 51.4 | **+20.0** |
> | **ALG (algebraic)** | 33.1 | 50.9 | **+17.8** |
> | **LOG (logical)** | 24.3 | 40.7 | **+16.4** |
>
> The lowest gains were observed in geometry, algebraic, and logical reasoning, suggesting these areas are particularly challenging for human solvers.
>
> **(3) For different grade levels:**
>
> | | **Random Chance** | **Human** | **Gain (%)** |
> | - | - | - | - |
> | Elementary School | 20.9 | 63.9 | +43.0 |
> | High School | 18.7 | 59.2 | +40.5 |
> | **College** | 19.4 | 53.8 | **+34.4** |
>
> At the college level, the human performance showed a gain of only 34.4% over random chance, indicating the inherent difficulty of these advanced problems.
>
> In light of these insights, we recognize the importance of **extending our analysis to include the performance of human experts, i.e., graduate students**. This expansion will help us understand how individuals with advanced educational backgrounds tackle these complex problems. We anticipate that including grad students in our study will provide a more nuanced understanding of human capabilities in mathematical reasoning within visual contexts, especially for the more challenging tasks that currently exhibit lower human accuracy rates.

---

> ### Author Response · Authors · 2023-11-17
> **Continued Response to Reviewer Vb5K: Dissecting Math Reasoning and Evaluating GPT-4V**
>
> ### W3: The takeaways from Table 2 regarding the seven math reasoning types
>
> We appreciate your suggestion to delve deeper into the insights derived from the seven types of fine-grained reasoning abilities detailed in Table 2. We **have added this discussion regarding the seven math reasoning types in Appendix F.2**, and provided more details for the analysis regarding different visual contexts in Appendix F.3 and grade levels in Appendix F.4, in [the revised paper (click to see the pdf)](https://openreview.net/pdf?id=KUNzEQMWU7).
>
> Our analysis indeed provides several key takeaways regarding seven math reasoning types for different models:
>
> 1. **GPT-4V** outperforms other baseline models in most mathematical reasoning categories, except for logical reasoning and numeric commonsense reasoning.
>
> 2. **Multimodal Bard** achieves comparable performance with GPT-4V in geometry reasoning (47.8% vs. 51.0%) and algebraic reasoning (46.5% vs. 53.0%), highlighting its enhanced abilities in comprehending geometry diagrams and performing algebraic calculations.
>
> 3. Among open-source LMMs, **LLaVA** achieves the best overall accuracy on MathVista and the highest fine-grained scores for problems in geometry reasoning, logical reasoning, and statistical reasoning. However, these scores still substantially lag behind GPT-4V and Multimodal Bard, indicating a gap in the overall effectiveness of these open-source models compared to more advanced proprietary systems.
>
> 4. Despite this, **LLaMA-Adapter-V2**, tied with LLaVA, outperforms GPT-4V by 2.7% in logical reasoning, and InstructBLIP beats GPT-4V by 0.3% in numeric commonsense, suggesting that specific enhancements in open-source models can lead to superior performance in certain niches.
>
> 5. **LLaVAR**, being on par with Multimodal Bard, which is specifically designed to enhance capabilities in detecting OCR texts and symbols from various forms, including scientific domains, further illustrates the potential of targeted improvements in open-source LMMs to achieve competencies that rival or even exceed those of their proprietary counterparts in specialized areas.
>
> 6. **CoT GPT-4**, augmented with OCR texts and Bard captions, performs well in scientific reasoning, achieving a gain of 26.2% over random chance, showcasing its superiority in domain-specific knowledge. This performance suggests a significant trend where the integration of specialized functionalities, such as OCR text recognition and advanced captioning, into LLMs enhances their applicability and accuracy in specific domains.
>
> 7. **PoT GPT-4** outperforms Multimodal Bard in categories such as arithmetic reasoning, logical reasoning, numeric commonsense reasoning, and statistical reasoning. This superior performance is attributed to its ability to generate high-quality codes for precise mathematical reasoning, illustrating the effectiveness of integrating advanced coding capabilities into language models for complex problem-solving tasks.
>
> ### W4: Results of GPT-4V
>
> Thank you for your interest in the results of GPT-4V, which is indeed a crucial part of our study. We **have conducted a comprehensive evaluation of GPT-4V on MathVista with expanded 67 pages** in [the updated version (click to see the pdf)](https://openreview.net/pdf?id=KUNzEQMWU7).
>
> For a more detailed analysis of GPT-4V’s performance, including its strengths and limitations, please refer to Section 3 of our paper, as well as Appendix G. The key findings are summarized as follows:
>
> 1. Notably, GPT-4V achieves an overall accuracy of 49.9%, significantly outperforming other models such as Bard by 15.1%. This highlights GPT-4V’s advanced capabilities in visual perception and mathematical reasoning.
>
> 2. GPT-4V's superior performance is particularly evident in tasks **requiring complex visual understanding and compositional reasoning**, which are areas where other models typically struggle. However, it is important to note that GPT-4V still shows **a gap of 10.4% compared to human performance**, especially in tasks involving intricate figures and rigorous reasoning.
>
> 3. GPT-4 demonstrates **an ability of self-verification**, which is not present in concurrent models. Self-verification enables GPT-4V to inspect a set of candidate answers and identify the one that is valid and meets all the given constraints. It also allows GPT-4V to verify the validity of key reasoning steps, and explore alternative approaches if any invalid result is detected.
>
> 4. We found **the application of self-consistency** in GPT-4V is instrumental in correcting calculation mistakes, rectifying visual perception errors, and mitigating hallucinations
>
> 5. We found that GPT-4V is effective in **engaging in multi-turn goal-directed conversations with users**. In particular, GPT-4V can make good use of hints (e.g., user feedback or responses) to guide the conversion to generate desirable results.

---

> ### Author Response · Authors · 2023-11-20
> **Thank You for Your Acknowledgment, and We Look Forward to Your Further Feedback**
>
> Dear Reviewer Vb5K,
>
> We are grateful for your insightful and encouraging comments on our paper. We have addressed your comments in detail with [a updated version of the paper (click to see the PDF)](https://openreview.net/pdf?id=KUNzEQMWU7), including **the additional 67 pages that highlight the new results of GPT-4V**.
>
> We kindly draw your attention to these updates and **eagerly anticipate any further feedback**. Your insights are invaluable in refining our paper.
>
> Best regards,
>
> The Authors of Paper 6738

---

> ### Comment · Reviewer_Vb5K · 2023-11-22
> **Thanks for the author response**
>
> I thank the authors for addressing my questions regarding disentangled evaluation, low human performance and results for GPT-4V. If the low human performance is due to the difficulty of the dataset instead of noise, maybe it's helpful to first do an annotation in the human study to choose "not answerable/not valid" and "answerable", then answer the "answerable" questions.
>
> I believe the benchmark is solid work and I thank the authors for their efforts. I maintain my original score as 8.

---

> > ### Author Response · Authors · 2023-11-22
> > **Thank you for your encouraging acknowledgment of our work**
> >
> > Dear Reviewer Vb5K,
> >
> > Thank you for your continued engagement with our work. We appreciate your suggestion regarding the human study design. Incorporating an initial annotation phase to categorize questions as "not answerable/not valid" and "answerable" is indeed a valuable idea. This approach would allow us to more accurately assess human performance on genuinely answerable questions, providing a clearer benchmark for comparison with GPT-4V's performance.
> >
> > Your feedback has been instrumental in refining our evaluation methods, and we plan to implement this additional annotation step in our future studies.
> >
> > Thank you once again for your invaluable feedback and support for our benchmark as solid work.
> >
> > Best regards,
> >
> > The authors of Paper 6738

---

### Official Review · Reviewer_2daf · 2023-10-31

**Soundness:** 3 good
**Presentation:** 4 excellent
**Contribution:** 3 good
**Rating:** 8
**Confidence:** 3

**Summary:**

The paper proposes a mathematical and visual reasoning benchmark to evaluate various visual-based mathematical skills of large-language models (LLMs) and large-multimodal models (LMMs). The authors introduce a taxonomy for mathematical visual reasoning which involves seven different types of mathematical reasoning scenarios and five associated tasks including figure-based question answering, math world problems and geometry problem solving. The visual scenarios are targeted to be diverse involving real images, synthetic scenes, charts and plots, scientific figures, etc. While majority of the benchmark is formed through 28 existing publicly-available datasets, authors form three new datasets targeted to fill gap for mathematical scenarios not covered by existing datasets.

The benchmark is relatively small and meant to be a zero-or few-shot evaluation benchmark divided into a "testmini" (1000 examples) and "test" (5141 examples). Evaluation of several model types (including LLMs, LLMs with visual context augmentation and large m) under different setups (including chain-of-thought, program-of-thought, few-shot) is performed and results indicate that current models perform poorly in comparison to humans. A brief error/success analysis and qualitative examples are provided for multimodal bard and context-augmented GPT-4.

**Strengths:**

1. The construction of a visual-based mathematical reasoning benchmark to evaluate current LLMs and LMMs is well motivated and relatively novel. The method for construction and statistics of the final benchmark are adequately described with appropriate references to source datasets and prior works.

2. The identified taxonomy covers a broad range of mathematical reasoning scenarios and tasks, and diversity is also maintained in the visual contexts. Further, the 3 newly collected datasets consider new tasks not covered by past works.

3. Evaluation is performed on prominent LLMs (such as GPT-4, ChatGPT, Claude) and LMMs (mPLUG-Owl, InstructBLIP, LLaVa, Multimodal-Bard). LLMs are evaluated in zero-shot, few-shot, chain-of-thought and program-of-thought settings and also when they are augmented with visual contexts. Further, human performance is computed and qualitative analysis and fine-grained result comparisons are performed to better highlight capabilities and limitations of existing models.

4. Paper is generally well written with appropriate figures and details illustrating the dataset examples and analysis, performance breakdown, qualitative examples, model prompts/settings and annotation methods.

**Weaknesses:**

1. For data collection of the 3 new datasets, it is not clear if inter-annotation consistency checks were conducted and how the mentioned "rigorous review process" was conducted (details are missing).

2. Few-shot performance is computed only for LLMs and not LMMs. Given LMMs such as Multimodal-Bard, Flamingo/Open-Flamingo and mPLUG-Owl also support few-shot learning, these can also be evaluated few-shot to better evaluate the benchmark challenges.

3. Further, LLMs can also be evaluated on a broader range of K-shot settings (currently only 2-shot is evaluated). Evaluation over {2,4,8,16,32} could provide better evidence of whether mathematical reasoning capabilities can be learned in a few-shot manner.

Relatively minor:

4. Benchmark is relatively small (6141 examples) and meant as an evaluation benchmark primarily drawn from existing datasets (5405 examples) with no finetuning subset which could be useful for improving mathematical reasoning capabilities of current models.

**Questions:**

Please see the weaknesses section above (primarily points 1,2 and 3).

---

> ### Author Response · Authors · 2023-11-17
> **Response to Reviewer 2daf: Clarifying Inter-Annotation Consistency and Evaluating Few-Shot Performance in LMMs**
>
> Dear Reviewer,
>
> We express our sincere gratitude for your thorough and insightful comments. We appreciate your acknowledgment of the novelty and motivation of our benchmark, the comprehensive taxonomy and the diversity of our dataset, our methodological approach, along with the diverse evaluated models.
>
> In response to your constructive feedback, we **have enriched our paper with additional in-depth analyses and results with additional 67 pages** in [the updated submission (click to see the pdf)](https://openreview.net/pdf?id=KUNzEQMWU7). We hope these enhancements will address any remaining concerns. Your thoughtful comments have been addressed as follows:
>
> ### W1: Details regarding inter-annotation consistency and rigorous review for 3 new datasets
>
> In response to your query, we provide the following details:
>
> 1. **Initial review**: After collecting the raw examples, a group of four graduate students specializing in STEM fields conducted an initial manual review. This review focused on verifying the relevance, clarity, and accuracy of the examples.
>
> 2. **Inter-annotation consistency checks**: To ensure consistency in annotation, we employed a two-step process. Initially, each dataset was independently annotated by three reviewers, resulting in **a high inter-annotation consistency rate of 99.2%**. Specifically, **among the newly collected 736 questions, only 6 exhibited disagreements in the annotated answers**. Then, these discrepancies were resolved through discussion among the entire review team, ensuring a consensus was reached on each example.
>
> 3. **Expert validation and iterative refinement**: The datasets underwent scrutiny by two domain experts in math. These experts evaluated the datasets for complexity, relevance, and the appropriateness of the mathematical concepts involved. Finally, the datasets were iteratively refined to revise any remaining issues.
>
> This rigorous review process was designed to ensure the highest quality and consistency of the datasets, thereby providing a robust benchmark. We believe these measures address your concerns about the reliability of our data collection and annotation process.
>
> ### W2: Few-shot performance for LMMs
>
> Thank you for your valuable feedback regarding the evaluation of few-shot learning capabilities in Large Multimodal Models (LMMs). We acknowledge the importance of this aspect in benchmarking the performance of these models.
>
> In our initial submission, we did not include few-shot performance results for LMMs due to the following reasons:
>
> 1. **Initial experimental observations**: Our preliminary experiments indicated that models like OpenFlamingo performed poorly on our dataset. This was largely attributed to a lack of specific instruction tuning, which is crucial for few-shot learning.
>
> 2. **Recent developments in few-shot Learning**: The ability of LMMs to support few-shot learning is a recent advancement. Notably, top-tier models like Multimodal-Bard and GPT-4V only recently started supporting few-shot learning in a scalable manner.
>
> 3. **Resource limitations**: GPT-4V, which was released for API access on November 6, 2023, currently imposes a daily limit of 100 calls. This limitation makes it impractical to conduct a complete evaluation of our testmini set, which consists of 1000 examples.
>
> To address your suggestion, we **conducted an initial study on the few-shot learning ability of the LMM model IDEFICS** on MathVista. The results are as follows:
>
> | Model | 0-shot | 2-shot (new) | 3-shot (new) | 4-shot (new) |
> | - | - | - | - | - |
> | IDEFICS-9B-Instruct | 19.8% | 23.7% | 23.4% | 24.1% |
>
> The data shows a modest improvement with increased shot numbers, suggesting some potential benefits of few-shot learning for LMMs on MathVista.
>
> However, **recent studies highlight the instability and sensitivity of LMMs in few-shot settings [1]**. For example, a significant accuracy drop was observed in models like BLIP-2 and InstructBLIP (Flan) when applying 4-shot in-context learning, especially in tasks requiring common sense reasoning.
>
> Here are some specific performance statistics on VQA with Commonsense Knowledge:
>
> | Model | 0-shot Acc (%) | 4-shot Acc (%) | Gain (%) |
> | - | - | - | - |
> | MiniGPT-4 | 29.31 | 25.96 | **-3.35** |
> | BLIP-2 | 39.06 | 25.95 | **-13.11** |
> | InstructBLIP (Vicuna) | 41.02 | 48.90 | +7.88 |
> | InstructBLIP (Flan) | 47.96 | 41.67 | **-6.29** |
> | LLaVA | 61.93 | 58.32 | **-3.61** |
> | GPT-4V | 64.28 | 62.01 | **-2.27** |
>
> [1] https://arxiv.org/abs/2311.07536
>
> These variations may stem from the specific training techniques or the nature of few-shot examples used, affecting the in-context learning performance of LMMs. Given the rapidly evolving landscape of LMMs, **the consistent benefits of few-shot learning remain an open question**. We are committed to **expanding our evaluations to include advanced LMMs like Multimodal Bard and GPT-4V** as more resources become available.

---

> ### Author Response · Authors · 2023-11-17
> **Continued Response to Reviewer 2daf: Expanding LLM Few-Shot Evaluations and Contextualizing MathVista**
>
> ### W3: LLMs can be evaluated on a broader range of K-shot settings
>
> Thank you for your insightful suggestion regarding the evaluation of Large Language Models (LLMs) over a broader range of K-shot settings. Following your recommendation, **we have expanded our experiments to include 0, 1, 2, 3, and 4-shot settings**, thus providing a more comprehensive view of the models' few-shot learning capabilities in math reasoning.
>
> Here are the updated results:
>
> | LLM Models | Input | 0-shot | **1-shot** **(new)** | 2-shot | **3-shot** **(new)** | **4-shot** **(new)** | **Peak at 4-shot?** |
> | - | - | - | - | - | - | - | - |
> | Claude-2 | Q only | 26.4 | 26.8 | 24.4 | 21.5 | 23.1 | **No** |
> | ChatGPT | Q only | 23.5 | 28.7 | 26.8 | 25.7 | 25.9 | **No** |
> | GPT-4 | Q only | 26.1 | 26.5 | 29.2 | 27.5 | 30.6 | Yes |
> | CoT Claude-2 | Q + I | / | 32.4 | 33.2 | 33.4 | 29.8 | **No** |
> | CoT ChatGPT | Q + I | / | 33.0 | 33.2 | 33.5 | 32.7 | **No** |
> | CoT GPT-4 | Q + I | / | 32.6 | 33.2 | 34.2 | 35.3 | Yes |
> | PoT ChatGPT | Q + I | / | 28.7 | 26.8 | 28.3 | 30.2 | Yes |
> | PoT GPT-4 | Q + I | / | 29.6 | 33.9 | 37.0 | 35.3 | **No** |
>
> Our findings indicate that in 5 out of 8 baseline settings, the peak accuracy is reached at a shot number of 2 or 3, instead of a larger number like 4. **Increasing the number of shots beyond this point does not necessarily guarantee performance improvement and can even result in a substantial drop**. For instance, CoT Claude-2 experienced a significant drop of 3.4% when increasing from 2 to 4 shots.
>
> Overall, while there might be marginal improvements, larger numbers of few-shot examples do not consistently benefit the LLMs on MathVista. Some settings even exhibit unstable performance drops, suggesting that **the quality of the augmented information plays a more critical role for augmented LLMs**. A detailed analysis of these findings has been added to Appendix F.6 of [our revised paper](https://openreview.net/pdf?id=KUNzEQMWU7).
>
> ### W4: Benchmark is relatively small with no finetuning subset
>
> Thank you for your feedback regarding the size of our benchmark and the absence of a finetuning subset. We appreciate this opportunity to clarify these aspects in the context of current trends in multimodal benchmarking:
>
> 1. **Benchmark size in context**: MathVista is an evaluation benchmark designed to assess the capabilities of existing LMMs. In the future, we plan to develop strategies to enhance it further. It's important to note that recent multimodal benchmarks for evaluating LMMs tend to be smaller in size, especially when compared to earlier datasets. This trend can be attributed to:
>
>  - **Model scale**: Modern LMMs are extremely large in terms of parameter count, often ranging from 10 to 100 billion. Evaluating such large-scale models on extensive datasets (e.g., 10K examples) is not practically feasible due to computational and resource constraints.
>
>  - **Quality over quantity**: In the realm of LMM evaluation, the diversity and quality of benchmarks are prioritized over sheer size. A smaller, well-curated dataset can often provide more insightful evaluations than a larger, less diverse one.
>
> 2. **The Role of finetuning data**: The exclusion of finetuning data in recent benchmarks, including ours, aligns with current development trends in LMMs, which favor instruction tuning over traditional finetuning.
>
> 3. **Contextualizing MathVista within recent benchmarks**: To provide context, we compare MathVista with other recent multimodal benchmarks:
>
>  | **Benchmark** | **Release date** | **Size** | **Finetuning data** |
>  | - | - | - | - |
>  | MMBench [1] | 2023-07-12 | 2,974 | No |
>  | MM-Vet [2] | 2023-08-02 | 205 | No |
>  | VisIT-Bench [3] | 2023-08-12 | 592 | No |
>  | Bingo [4] | 2023-11-06 | 321 | No |
>  | K-I VQA [5] | 2023-11-13 | 4,290 | No |
>  | **MathVista (Ours)** | 2023-09-28 | 6,141 | No |
>
> [1] https://arxiv.org/abs/2307.10635
>
> [2] https://arxiv.org/abs/2308.02490
>
> [3] https://arxiv.org/abs/2308.06595
>
> [4] https://arxiv.org/abs/2311.03287
>
> [5] https://arxiv.org/abs/2311.07536
>
> MathVista is a comprehensive benchmark, notable for its specific focus on evaluating mathematical reasoning in visual contexts. It integrates content from 31 diverse datasets, making it a significant and diverse collection for this area of research. The inclusion of 6,141 examples ensures a low margin of error (±1%) at a 95% confidence level, thereby providing accurate and reliable results for the models evaluated.

---

> ### Author Response · Authors · 2023-11-20
> **Kind Request for Your Continued Feedback**
>
> Dear Reviewer 2daf,
>
> Thank you so much for your insightful comments and effort in reviewing our paper. We have addressed your feedback in our detailed responses, adding **new experiments for few-shot LMMs**, and expanding **the experiments to cover a wider range of few-shots for LLMs**. Furthermore, we have updated [the submission (click to see the PDF)](https://openreview.net/pdf?id=KUNzEQMWU7) with an additional 67 pages that provide **a comprehensive evaluation of the latest GPT-4V model**.
>
> We hope that these updates have addressed your questions. We're hopeful that our responses may prompt a reconsideration of the paper's rating. Your continued feedback is crucial to us. Thank you!
>
> Best regards,
>
> The Authors of Paper 6738

---

> > ### Comment · Reviewer_2daf · 2023-11-21
> > **Thanks for the informative response and updated evaluations.**
> >
> > Thanks to authors for the comprehensive response. The expanded few-shot results over {2,4,6,8..} and for VLMs are quite interesting suggesting the requisite mathematical reasoning abilities cannot be simply learned in-context. I would suggest including these results in appendix. The details about inter-annotation review and collection process for new datasets should also be included in appendix if not already done so. The further evaluation of GPT-4 is also an interesting inclusion.
> >
> > Overall, I find this to be a strong and interesting paper, and have increased my score to 8 accordingly.

---

> > > ### Author Response · Authors · 2023-11-21
> > > **Thank You for Your Acknowledgment and the Increased Rating**
> > >
> > > Dear Reviewer 2daf,
> > >
> > > We sincerely appreciate your prompt reply and further insightful comments. We are highly encouraged by your acknowledgment of our comprehensive response with new experiments and the revised paper. Your recognition of our paper's strength and the upgrade of its score to a clear accept 8 is greatly valued.
> > >
> > > Thank you for your suggestions. We have incorporated expanded few-shot results for LLMs in Appendix F.6 on page 39. We have also included additional details on the inter-annotation review and collection process on page 14 and few-shot results for LMMs in Appendix F.7 on the same page. These additions are highlighted in blue in the [the latest submission (click to see the PDF)](https://openreview.net/pdf?id=KUNzEQMWU7).
> > >
> > > Thank you again for your invaluable efforts in reviewing and refining our paper.
> > >
> > > Sincerely,
> > >
> > > The Authors of Paper 6738

---

### Official Review · Reviewer_juci · 2023-11-01

**Soundness:** 3 good
**Presentation:** 3 good
**Contribution:** 2 fair
**Rating:** 5
**Confidence:** 4

**Summary:**

This work introduces MathVista, a benchmark for evaluating the mathematical reasoning abilities of large language models (LLMs) and large multimodal models (LMMs) within visual contexts. The work uses data from a broad range of existing math and visual question-answering datasets and constructs three novel datasets: IQTest, FunctionQA, and PaperQA. The work evaluates nearly a dozen models with MathVista and finds that Multimodal Bard, the best-performing model, achieves 58% of human performance.

**Strengths:**

1. Curating the datasets from existing sources and developing three new datasets contribute to comprehensive and diverse testing.

2. Evaluating nearly a dozen models and comparing their performances with a human benchmark provides insight into current performance.

3. Mathematical reasoning within visual contexts is an important field.

**Weaknesses:**

1. The results have been completely superseded by GPT-4 Vision, which together with GPT-4 are SotA in vision-language models.

2. The methods have been superseded by recent prompting methods.

3. The related work and references are lacking:

a. Prompting:
+ Phenomenal yet puzzling: Testing inductive reasoning capabilities of language models with hypothesis refinement
L. Qiu, L. Jiang, X. Lu, M. Sclar, V. Pyatkin, C. Bhagavatula, B. Wang, Y. Kim, Y. Choi, N. Dziri, X. Ren, 2023.
+ Hypothesis search: Inductive reasoning with language models, R. Wang, E. Zelikman, G. Poesia, Y. Pu, N. Haber, N. D. Goodman, 2023.
+ Large language model (LLM) as a system of multiple expert agents: An approach to solve the abstraction and reasoning corpus (ARC) Challenge, J. T. Min, M. Motani, 2023.

b. GPT-4V:
+ Lost in translation: When GPT-4V(ision) can’t see eye to eye with text, a vision-language-consistency analysis of VLLMs and beyond,
X. Zhang, S. Li, Z. Wu, N. Shi, 2023.

c. PoT:
Solving Linear Algebra by program synthesis, I. Drori, N. Verma, November 2021.
Solving Probability and Statistics problems by probabilistic program synthesis at human level and predicting solvability
L. Tang, E. Ke, N. Singh, B. Feng, D. Austin, N. Verma, I. Drori, AIED, 2022.
A neural network solves, explains, and generates university math problems by program synthesis and few-shot learning at human level
I. Drori, S. Zhang, R. Shuttleworth, L. Tang, A. Lu, E. Ke, K. Liu, L. Chen, S. Tran, N. Cheng, R. Wang, N. Singh, T. L. Patti, J. Lynch, A. Shporer, N. Verma, E. Wu, G. Strang, PNAS, 2022.

4. The paper lacks a comprehensive discussion of the limitations of the benchmark.

5. The work is missing an analysis of why specific models perform better than others.

I expect the author response to include GPT-4 Vision results, missing related work and references, and updated methods.

**Questions:**

While the paper covers a broad range of models,

it is missing an analysis of why specific models perform better than others?

and what features of each model contribute to performance?

This would be a valuable contribution.

---

> ### Author Response · Authors · 2023-11-17
> **Response to Reviewer juci: Incorporating GPT-4V Results and Enhanced Revisions**
>
> Dear Reviewer,
>
> Thank you for your constructive comments. We are pleased that you recognize the motivation and diversity of our dataset, the novelty of the three new datasets, and the comprehensive evaluation covering a large number of prominent LLMs and LMMs across diverse settings.
>
> To address your comments, we have added **67 pages that include new results of GPT-4V and revisions** in [the updated submission (click to see the pdf)](https://openreview.net/pdf?id=KUNzEQMWU7). We hope these updates address your concerns, possibly prompting you to adjust your rating.
>
> ### W1: Adding GPT-4V(ision) results
>
> Thank you for your comments. We have included a comprehensive study of GPT-4V in the revision, showcasing **both quantitative and qualitative results**. The main findings are highlighted as follows:
>
> - GPT-4V outperforms other models like Bard, achieving 49.9% accuracy, notably higher by 15.1%.
> - Its superiority is due to improved visual perception and mathematical reasoning abilities.
> - However, it still lags behind human performance by 10.4%, struggling with complex figures and rigorous reasoning.
> - GPT-4V introduces an ability of self-verification to verify and refine the intermediate reasoning steps.
> - The model's use of self-consistency helps correct visual and calculation errors, though it's less - effective in complex visual contexts.
> - In multi-turn human-AI interactions on MathVista, GPT-4V effectively uses hints to guide conversations, correct errors, and handle complex contexts.
>
> For more detailed information, please refer to the updates highlighted in blue on the main pages and Appendix G in [the updated submission (click to see the pdf)](https://openreview.net/pdf?id=KUNzEQMWU7).
>
> ### W2: The methods have been superseded by recent prompting methods
>
> In this paper, our aim is to evaluate the capabilities of **mathematical reasoning within visual contexts**. The **multimodal evaluation** framework led us to choose 9 large multimodal models (LLMs) as our primary models. We focused on evaluating the latest LLMs, including GPT-4V, released on September 25, 2023, and Multimodal Bard, released in late July 2023.
>
>
> We also evaluated three large language models as **question-only baselines** in both zero-shot and few-shot settings, employing chain-of-thought and program-of-thought prompting strategies. The best-performing LLM baseline achieved an accuracy of only 26.0%, highlighting that the **main limitation of current LLMs is their inability to understand visual contexts**, as evidenced in our MathVista benchmark.
>
>
> Consequently, we explored augmented LLMs enhanced with external tools, representing a cutting-edge research area in LLMs. The top-performing augmented LLM (PoT GPT-4) attained only 33.9% accuracy, showing a significant 16.0% gap compared to GPT-4V. Our detailed study in Section 3.5 indicates that for current augmented LLMs, **the primary bottleneck is the incorrect augmentation information from external visual models**.
>
>
> Given the multimodal nature of our benchmark and the notable performance gap between augmented LLMs and GPT-4V, our evaluation predominantly focused on multimodal models on MathVista. This approach is due to the fact that **recent prompting methods for LLMs are not suitable in this context**. We sincerely hope this response adequately addresses your concerns.
>
> ### W3: Lacking related work and references
>
> Thank you for your comments regarding the related work and references. As per the ICLR 2023 Reviewer Guidelines, authors are not required to compare their work to papers published on or after May 28, 2023 (source: [ICLR 2023 Reviewer Guide](https://iclr.cc/Conferences/2023/ReviewerGuide)). We wish to clarify that some recent publications were not included in our previous submission as they were **published around or even after the ICLR paper submission deadline (September 28, 2023)**:
>
> 1. Lost in translation: When GPT-4V(ision) can’t see eye to eye with text, a vision-language-consistency analysis of VLLMs and beyond, https://arxiv.org/abs/2310.12520, submitted on 19 Oct, 2023.
>
> 2. Phenomenal yet puzzling: Testing inductive reasoning capabilities of language models with hypothesis refinement, https://arxiv.org/abs/2310.08559, submitted on 12 Oct, 2023.
>
> 3. Large language model (LLM) as a system of multiple expert agents, https://arxiv.org/abs/2310.05146, submitted on 8 Oct, 2023.
>
> 4. Hypothesis search: Inductive reasoning with language models, https://arxiv.org/abs/2309.05660, submitted on 11 Sep, 2023.
>
> We appreciate your suggestion to discuss these papers. Accordingly, we **have included them on Page 21** in [the updated version (click to see the pdf)](https://openreview.net/pdf?id=KUNzEQMWU7) of our paper.
>
> We will **cite these very recent works in the final version of the paper**. But given that the ICLR rules state there is no requirement to do so, we respectfully contend that **this comment shouldn't be considered a weakness**.

---

> ### Author Response · Authors · 2023-11-17
> **Continued Response to Reviewer juci: Adding Limitation Discussion and Enhancing Model Comparative Analysis**
>
> ### W4: A comprehensive discussion of the limitations of the benchmark
>
> Thank you for your valuable feedback. In response, **we have included a detailed discussion of the limitations of our benchmark, MathVista, in Section 4** of the [the updated version (click to see the pdf)](https://openreview.net/pdf?id=KUNzEQMWU7), which we iterate below.
>
> MathVista represents a significant advancement in combining mathematical and visual tasks, challenging even for models like GPT-4V, particularly in understanding complex figures and performing rigorous reasoning. Despite our progress in evaluating model performance, we recognize several limitations:
>
> 1. A primary limitation is **dataset coverage**. While MathVista covers a wide range of tasks and visual contexts, gaps may exist in the representation of certain mathematical problem types and visual scenarios.
>
> 2. Additionally, the dataset's emphasis on mathematical reasoning within visual contexts, especially in domains like science and advanced mathematics, requires a more **labor-intensive data collection** process than what is typical for textual-only or general-purpose datasets. Consequently, the **scalability and generalizability** of our benchmark to other domains present challenges.
>
> 3. The heterogeneity of data sources, from which we sourced **annotations** for 85.6% of examples (as indicated in Table 1), leads to **a lack of uniformity in format and structure**. For example, the annotations could be logic forms of the problem parsing from Geometry3K, natural language solutions from TabMWP, and theorems from TheoremQA.
>
> In future iterations, we hope to **broaden the scope of MathVista** to include a more diverse range of problems and visual contexts and to **provide unified and comprehensive annotations**. As part of our ongoing research process, we are committed to continually **updating and refining the benchmark** based on community feedback. This includes **improving data quality and adapting the leaderboard** to reflect advancements and new models in the field.
>
> In conclusion, while there are limitations to our current approach, MathVista represents a significant step forward in the field. We are dedicated to continuously improving our benchmark to better understand and enhance the capabilities of AI in mathematical and visual reasoning.
>
> ### W5: An analysis of why specific models perform better than others (and what features of each model contribute to performance)
>
> Thank you for emphasizing the importance of a detailed comparative analysis of different models. In our original submission, we have conducted a comprehensive analysis of representative models:
>
> - Section 3.2 provides an analysis of the **overall performance of various models**, including the best LLM (GPT-4), the best Augmented LLM (PoT GPT-4), open-source LMMs, Bard, and the best LMM (GPT-4V).
>
> - In Section 3.5, we presented **a detailed success and failure analysis of Multimodal Bard** with human annotations. We discovered that 44.6% of its predictions were incorrect with flawed explanations, and hallucinations were present in 49.6% of these incorrect explanations.
>
> - A quantitative study comparing the performance of **Multimodal Bard and Augmented GPT-4** was also conducted in Section 3.5.
>
> - We conducted a qualitative analysis examining model performance across **different math reasoning types** (Appendix F.2), **visual contexts** (Appendix F.3), and **grade levels** (Appendix F.4).
>
> - An ablation study in Appendix F.5 explored **how LLMs can benefit from augmented visual information**, such as OCR and captions.
>
> In [the updated version of our paper (click to see the pdf)](https://openreview.net/pdf?id=KUNzEQMWU7), we have included an additional 67 pages offering **an in-depth study of the strengths and weaknesses of GPT-4V, Bard, LLaVA, and LLaMA-Adapter-V2**, primarily detailed in Appendix G.

---

> ### Author Response · Authors · 2023-11-20
> **Sincere Request for Review of Our Responses, New Experiments, and Revised Paper**
>
> Dear Reviewer juci,
>
> We greatly appreciate your time and efforts in reviewing our paper and offering insightful comments. During the rebuttal period, **we have added 67 pages that include new results of GPT-4V and revisions** in [the updated submission (click to see the PDF)](https://openreview.net/pdf?id=KUNzEQMWU7).
>
> **We hope that our thorough responses, along with the new results and revisions, will further underscore the value of our work and encourage you to consider adjusting and elevating your rating**. Your insights are invaluable in refining our paper!
>
> Best regards,
>
> The authors of Paper 6738

---

> ### Author Response · Authors · 2023-11-22
> **Kind Request for Acknowledgment of Our Responses, New Experiments, and Revisions**
>
> Dear Reviewer juci,
>
> Thank you so much for your time and efforts in reviewing our paper. Your initial comments are very insightful and helpful in refining our paper. We have addressed your comments by (1) posting detailed responses; (2) adding comprehensive quantitative and qualitative results of GPT-4V; and (3) updating the [submission (click to see the PDF)](https://openreview.net/pdf?id=KUNzEQMWU7) with an additional 67 pages.
>
> **We sincerely hope that these revisions have adequately addressed your feedback and will encourage you to reconsider the rating**. If you have any further questions, please do not hesitate to discuss them with us before the discussion deadline. Your insights are invaluable in refining our paper!
>
> Best regards,
>
> The authors of Paper 6738

---

### Author Response · Authors · 2023-11-17
**General Responses with GPT-4V Results and Additional 67 Pages of New Content in the Revision**

We greatly appreciate the efforts of all reviewers in reviewing our paper and providing thoughtful suggestions for its improvement.

We are pleased that the reviewers have acknowledged the **diversity** of our proposed MathVista benchmark. This diversity spans across source datasets, mathematical reasoning types, visual contexts, and task types. Our dataset is described as **comprehensive** (R4-iJ4P), **well-designed**, and **relatively large** (R3-Vb5K), structured under an identified taxonomy (R2-2daf & R4-iJ4P). We are particularly pleased to see R1-juci and R2-2daf highlight the **novelty** of the three newly curated datasets that address gaps in existing datasets.

The **comprehensive evaluations** conducted under various settings with a large number of existing LLMs and multimodal LLMs, including SoTA models such as miniGPT-4, LLaVA(R), Bard, and Instruct-BLIP, have also been appreciated by all reviewers.

Furthermore, we are delighted that all reviewers have emphasized the significant **contributions of our work**. R1-juci notes that our paper studies **an important field**, and the curated datasets contribute to comprehensive and diverse testing. R2-2daf remarks on the **well-motivated** and relatively novel construction of MathVista. R3-Vb5K points out the paper focuses on a specific topic and defines the problem well, which is **important** for LLM evaluation. Lastly, R4-iJ4P values the consolidation of existing datasets into a **comprehensive** benchmark.

### Added GPT-4V Results

Following the suggestions of R1-juci and R3-Vb5K, we have included **a comprehensive quantitative and qualitative study of GPT-4V(ision)** on our MathVista benchmark in [the updated submission (click to see the pdf)](https://openreview.net/pdf?id=KUNzEQMWU7). The new findings are highlighted as follows:

- GPT-4V emerges as the best-performing model, achieving **an overall accuracy of 49.9%**, substantially outperforming Bard, the second-best model, by 15.1% (see Table 2 on Page 7).

- Our in-depth analysis indicates that the superiority of GPT-4V is primarily due to its **enhanced visual perception and mathematical reasoning** capabilities (see Figure 1 on Page 2, Appendix G.3 and G.4 on Pages 56-96).

- Despite its achievements, GPT-4V still **falls short of human performance by 10.4%**, often struggling with complex figures and rigorous reasoning (Table 2 on Page 7).

- We highlight **the new ability of self-verification** in GPT-4V, absent in existing LMMs. This feature allows GPT-4V to verify the validity of primary reasoning steps and explore alternative approaches if any invalid results are detected, demonstrating its potential in solving rigorous reasoning and theorem-proving tasks (Appendix G.5 on Pages 97-102)

- Our research into **the application of self-consistency** with GPT-4V shows that it is instrumental in rectifying visual perception errors, correcting calculation mistakes, and mitigating hallucinations. However, self-consistency is less effective when GPT-4V faces challenges in interpreting complex visual contexts or extracting salient information from images. (Appendix G.6 on Pages 103-108)

- In examining GPT-4V’s application for **multi-turn human-AI interaction** in MathVista, we found that GPT-4V effectively uses hints to guide conversations and generate desirable results. It can rectify visual perception errors, reassess reasoning steps and calculations, correct misinformation with user-provided domain-specific knowledge, and aggregate intricate contexts over multiple turns in human-AI dialogues. (Appendix G.7 on Pages 109-116)

For more details, please refer to the updates highlighted in blue in [the updated submission (click to see the pdf)](https://openreview.net/pdf?id=KUNzEQMWU7).

### Other New Contents in the Updated PDF Submission

[The updated submission (click to see the pdf)](https://openreview.net/pdf?id=KUNzEQMWU7) also includes the following enhancements:

- We have updated the abstract, introduction, experiment, and conclusion sections to reflect the new results and findings pertaining to GPT-4V.

- Additional discussions of more recent work have been included on Page 21.

- For easier navigation, a detailed Contents section has been added on Pages 18-19.

- A detailed comparison of different models in their fine-grained scores across different math reasoning types, visual contexts, and grade levels in Appendices F.2, F.3, and F.4 on Pages 36-38.

- A study of LLMs with different shot numbers is discussed in Appendix F.6 on Page 39.

- A comprehensive study of GPT-4V, Bard, and other models is now featured in Appendix G.

These updates will be consistently incorporated into future versions of the paper.

---

### Meta-Review · Area_Chair_Fxbk · 2023-12-11

**Metareview:**

The authors introduced a new benchmark for evaluating the mathematical reasoning capabilities of large visual-language models on problems with visual context. The submission also made comprehensive experiments to compare the performance of state-of-the-art methods, including Bard and GPT-4V (in the rebuttal). The dataset and the results are both insightful for future researchers. While one of the reviewers raised concerns about citing existing works, these works are published close to or after the ICLR submission deadline. Therefore, the area chair recommends to accept the submission.

**Justification For Why Not Higher Score:**

N/A

**Justification For Why Not Lower Score:**

The work seems to be impactful for the community.

---

### Decision · Program_Chairs · 2024-01-16

Accept (oral)